# Regret in Online Recommendation Systems

**Kaito Ariu**
KTH
Stockholm, Sweden
ariu@kth.se

**Narae Ryu**
KAIST
Daejeon, South Korea
nrryu@kaist.ac.kr

**Se-Young Yun**
KAIST
Daejeon, South Korea
yunseyoung@kaist.ac.kr

**Alexandre Proutière**
KTH
Stockholm, Sweden
alepro@kth.se

## Abstract

This paper proposes a theoretical analysis of recommendation systems in an online setting, where items are sequentially recommended to users over time. In each round, a user, randomly picked from a population of $m$ users, requests a recommendation. The decision-maker observes the user and selects an item from a catalogue of $n$ items. Importantly, an item cannot be recommended twice to the same user. The probabilities that a user likes each item are unknown. The performance of the recommendation algorithm is captured through its regret, considering as a reference an Oracle algorithm aware of these probabilities. We investigate various structural assumptions on these probabilities: we derive for each structure regret lower bounds, and devise algorithms achieving these limits. Interestingly, our analysis reveals the relative weights of the different components of regret: the component due to the constraint of not presenting the same item twice to the same user, that due to learning the chances users like items, and finally that arising when learning the underlying structure.

## 1 Introduction

Recommendation systems [28] have over the last two decades triggered important research efforts (see, e.g., [9, 10, 21, 3] for recent works and references therein), mainly focused towards the design and analysis of algorithms with improved efficiency. These algorithms are, to some extent, all based on the principle of *collaborative filtering*: similar items should yield similar user responses, and similar users have similar probabilities of liking or disliking a given item. In turn, efficient recommendation algorithms need to learn and exploit the underlying structure tying the responses of the users to the various items together.

Most recommendation systems operate in an online setting, where items are sequentially recommended to users over time. We investigate recommendation algorithms in this setting. More precisely, we consider a system of $n$ items and $m$ users, where $m \geq n$ (as this is typically the case in practice). In each round, the algorithm needs to recommend an item to a *known* user, picked randomly among the $m$ users. The response of the user is noisy: the user likes the recommended item with an a priori *unknown* probability depending on the (item, user) pair. In practice, it does not make sense to recommend an item twice to the same user (why should we recommend an item to a user who already considered or even bought the item?). We restrict our attention to algorithms that do not recommend an item twice to the same user, a constraint referred to as the *no-repetition* constraint. The objective is to devise algorithms maximizing the expected number of successful recommendations over a time horizon of $T$ rounds.

We investigate different system structures. Specifically, we first consider the case of clustered items and statistically identical users – the probability that a user likes an item depends on the item cluster only. We then study the case of unclustered items and statistically identical users – the probability that a user likes an item depends on the item only. The third investigated structure exhibits clustered items and clustered users – the probability that a user likes an item depends on the item and user clusters only. In all cases, the structure (e.g., the clusters) is initially unknown and has to be learnt to some extent. This paper aims at answering the question: How can the structure be optimally learnt and exploited?

To this aim, we study the regret of online recommendation algorithms, defined as the difference between their expected number of successful recommendations to that obtained under an Oracle algorithm aware of the structure and of the success rates of each (item, user) pair. We are interested in regimes where $n, m$, and $T$ grow large simultaneously, and $T = o(mn)$ (see §3 for details). For the aforementioned structures, we first derive non-asymptotic and problem-specific regret lower bounds satisfied by any algorithm.

(i) For clustered items and statistically identical users, as $T$ (and hence $m$) grows large, the minimal regret scales as $K \max\{\frac{\log(m)}{\log(\frac{m \log(m)}{T})}, \frac{T}{\Delta m}\}$, where $K$ is the number of item clusters, and $\Delta$ denotes the minimum difference between the success rates of items from the optimal and sub-optimal clusters.

(ii) For unclustered items and statistically identical users, the minimal *satisficing* regret[1] scales as $\max\{\frac{\log(m)}{\log(\frac{m \log(m)}{T})}, \frac{T}{m\varepsilon}\}$, where $\varepsilon$ denotes the threshold defining the satisficing regret (recommending an item in the top $\varepsilon$ percents of the items is assumed to generate no regret).

(iii) For clustered items and users, the minimal regret scales as $\frac{m}{\Delta}$ or as $\frac{m}{\Delta} \log(T/m)$, depending on the values of the success rate probabilities.

We also devise algorithms that provably achieve these limits (up to logarithmic factors), and whose regret exhibits the right scaling in $\Delta$ or $\varepsilon$. We illustrate the performance of our algorithms through experiments presented in the appendix.

Our analysis reveals the relative weights of the different components of regret. For example, we can explicitly identify the regret induced by the no-repetition constraint (this constraint imposes us to select unrecommended items and induces an important learning price). We may also characterize the regret generated by the fact that the item or user clusters are initially unknown. Specifically, fully exploiting the item clusters induces a regret scaling as $K\frac{T}{\Delta m}$. Whereas exploiting user clusters has a much higher regret cost scaling as least as $\frac{m}{\Delta}$.

In our setting, deriving regret lower bounds and devising optimal algorithms cannot be tackled using existing techniques from the abundant bandit literature. This is mainly due to the no-repetition constraint, to the hidden structure, and to the specificities introduced by the random arrivals of users. Getting tight lower bounds is particularly challenging because of the non-asymptotic nature of the problem (items cannot be recommended infinitely often, and new items have to be assessed continuously). To derive these bounds, we introduce novel techniques that could be useful in other online optimization problems. The design and analysis of efficient algorithms also present many challenges. Indeed, such algorithms must include both clustering and bandit techniques, that should be jointly tuned.

Due to space constraints, we present the pseudo-codes of our algorithms, all proofs, numerical experiments, as well as some insightful discussions in the appendix.

## 2   Related Work

The design of recommendation systems has been framed into structured bandit problems in the past. Most of the work there consider a linear reward structure (in the spirit of the matrix factorization approach), see e.g. [9], [10], [22], [20], [21], [11]. These papers ignore the no-repetition constraint (a usual assumption there is that when a user arrives, a set of fresh items can be recommended). In [24], the authors try to include this constraint but do not present any analytical result. Furthermore, notice that the structures we impose in our models are different than that considered in the low-rank matrix factorization approach.

Our work also relates to the literature on clustered bandits. Again the no-repetition constraint is not modeled. In addition, most often, only the user clusters [6], [23] or only the item clusters are considered [18], [14]. Low-rank bandits extend clustered bandits by modeling the (item, user) success rates as a low-rank matrix, see [15], [25], still without accounting for the no-repetition constraint, and without a complete analysis (no precise regret lower bounds).

One may think of other types of bandits to model recommendation systems. However, none of them captures the essential features of our problem. For example, if we think of contextual bandits (see, e.g., [12] and references therein), where the context would be the user, it is hard to model the fact that when the same context appears several times, the set of available arms (here items) changes depending on the previous arms selected for this context. Budgeted and sleeping bandits [7], [17] model scenarios where the set of available arms changes over time, but in our problem, this set changes in a very specific way not covered by these papers. In addition, studies on budgeted and sleeping bandits do not account for any structure.

The closest related work can be found in [4] and [13]. There, the authors explicitly model the no-repetition constraint but consider user clusters only, and do not provide regret lower bounds. [3] extends the analysis to account for item clusters as well but studies a model where users in the same cluster deterministically give the same answers to items in the same cluster.

## 3  Models and Preliminaries

We consider a system consisting of a set $\mathcal{I} = [n] := \{1, \ldots, n\}$ of items and a set $\mathcal{U} = [m]$ of users. In each round, a user, chosen uniformly at random from $\mathcal{U}$, needs a recommendation. The decision-maker observes the user id and selects an item to be presented to the user. Importantly an item cannot be recommended twice to a user. The user immediately rates the recommended item +1 if she likes it or 0 otherwise. This rating is observed by the decision-maker, which helps subsequent item selections.

Formally, in round $t$, the user $u_t \sim \mathrm{unif}(\mathcal{U})$ requires a recommendation. If item $i$ is recommended, the user $u_t = u$ likes the item with probability $\rho_{iu}$. We introduce the binary r.v. $X_{iu}$ to indicate whether the user likes the item, $X_{iu} \sim \mathrm{Ber}(\rho_{iu})$. Let $\pi$ denote a sequential item selection strategy or algorithm. Under $\pi$, the item $i_t^\pi$ is presented to the $t$-th user. The choice $i_t^\pi$ depends on the past observations and on the identity of the $t$-th user, namely, $i_t^\pi$ is $\mathcal{F}_{t-1}^\pi$-measurable, with $\mathcal{F}_{t-1}^\pi = \sigma(u_t, (u_s, i_s^\pi, X_{i_s^\pi u_s}), s \leq t-1)$ ($\sigma(Z)$ denotes the $\sigma$-algebra generated by the r.v. $Z$). Denote by $\Pi$ the set of such possible algorithms. The reward of an algorithm $\pi$ is defined as the expected number of positive ratings received over $T$ rounds: $\mathbb{E}[\sum_{t=1}^T \rho_{i_t^\pi u_t}]$. We aim at devising an algorithm with maximum reward.

We are mostly interested in scenarios where $(m, n, T)$ grow large under the constraints (i) $m \geq n$ (this is typically the case in recommendation systems), (ii) $T = o(mn)$, and (iii) $\log(m) = o(n)$. Condition (ii) complies with the no-repetition constraint and allows some freedom in the item selection process. (iii) is w.l.o.g. as explained in [4], and is just imposed to simplify our definitions of regret (refer to Appendix C for a detailed discussion).

### 3.1  Problem structures and regrets

We investigate three types of systems depending on the structural assumptions made on the success rates $\rho = (\rho_{iu})_{i \in \mathcal{I}, u \in \mathcal{U}}$.

**Model A. Clustered items and statistically identical users.** In this case, $\rho_{iu}$ depends on the item $i$ only. Items are classified into $K$ clusters $\mathcal{I}_1, \ldots \mathcal{I}_K$. When the algorithm recommends an item $i$ for the first time, $i$ is assigned to cluster $\mathcal{I}_k$ with probability $\alpha_k$, independently of the cluster assignments of the other items. When $i \in \mathcal{I}_k$, then $\rho_i = p_k$. We assume that both $\alpha = (\alpha_k)_{k \in [K]}$ and $p = (p_k)_{k \in [K]}$ do not depend on $(n, m, T)$, but are initially unknown. W.l.o.g. assume that $p_1 > p_2 \geq p_3 \geq \ldots \geq p_K$. To define the regret of an algorithm $\pi \in \Pi$, we compare its reward to that of an Oracle algorithm aware of the item clusters and of the parameters $p$. The latter would *mostly* recommend items from cluster $\mathcal{I}_1$. Due to the randomness in the user arrivals and the cluster sizes, recommending items not in $\mathcal{I}_1$ may be necessary. However, we define regret as if recommending items from $\mathcal{I}_1$ was always possible. Using our assumptions $T = o(mn)$ and $\log(m) = o(n)$, we can show that the difference between our regret and the true regret (accounting for the possible need to

recommend items outside $\mathcal{I}_1$) is always negligible. Refer to Appendix C for a formal justification. In summary, the regret of $\pi \in \Pi$ is defined as: $R^\pi(T) = Tp_1 - \sum_{t=1}^T \mathbb{E}\left[\sum_{k=1}^K \mathbb{1}_{\{i_t^\pi \in \mathcal{I}_k\}} p_k\right].$

**Model B. Unclustered items and statistically identical users.** Again here, $\rho_{iu}$ depends on the item $i$ only. when a new item $i$ is recommended for the first time, its success rate $\rho_i$ is drawn according to some distribution $\zeta$ over $[0,1]$, independently of the success rates of the other items. $\zeta$ is arbitrary and initially unknown, but for simplicity assumed to be absolutely continuous w.r.t. Lebesgue measure. To represent $\zeta$, we also use its inverse distribution function: for any $x \in [0,1]$, $\mu_x := \inf\{\gamma \in [0,1] : \mathbb{P}[\rho_i \leq \gamma] \geq x\}$. We say that an item $i$ is within the $\varepsilon$-best items if $\rho_i \geq \mu_{1-\varepsilon}$. We adopt the following notion of regret: for a given $\varepsilon > 0$, $R_\varepsilon^\pi(T) = \sum_{t=1}^T \mathbb{E}\left[\max\{0, \mu_{1-\varepsilon} - \rho_{i_t^\pi}\}\right].$ Hence, we assume that recommending items within the $\varepsilon$-best items does not generate any regret. We also assume, as in Model A, that an Oracle policy can always recommend such items (refer to Appendix C). This notion of *satisficing* regret [29] has been used in the bandit literature to study problems with a very large number of arms (we have a large number of items). For such problems, identifying the best arm is very unlikely, and relaxing the regret definition is a necessity. Satisficing regret is all the more relevant in our problem that even if one would be able to identify the best item, we cannot recommend it (play it) more than $m$ times (due to the no-repetition constraint), and we are actually forced to recommend sub-optimal items. A similar notion of regret is used in [4] to study recommendation systems in a setting similar to our Model B.

**Model C. Clustered items and clustered users.** We consider the case where both items and users are clustered. Specifically, users are classified into $L$ clusters $\mathcal{U}_1, \dots, \mathcal{U}_L$, and when a user arrives to the system the first time, she is assigned to cluster $\mathcal{U}_\ell$ with probability $\beta_\ell$, independently of the other users. There are $K$ item clusters $\mathcal{I}_1, \dots \mathcal{I}_K$. When the algorithm recommends an item $i$ for the first time, it is assigned to cluster $\mathcal{I}_k$ with probability $\alpha_k$ as in Model A. Now $\rho_{iu} = p_{k\ell}$ when $i \in \mathcal{I}_k$ and $u \in \mathcal{U}_\ell$. Again, we assume that $p = (p_{k\ell})_{k,\ell}$, $\alpha = (\alpha_k)_{k \in [K]}$ and $\beta = (\beta_\ell)_{\ell \in [L]}$ do not depend on $(n, m, T)$. For any $\ell$, let $k_\ell^* = \arg\max_k p_{k\ell}$ be the best item cluster for users in $\mathcal{U}_\ell$. We assume that $k_\ell^*$ is unique. In this scenario, we assume that an Oracle algorithm, aware of the item and user clusters and of the parameters $p$, would only recommend items from cluster $k_\ell^*$ to a user in $\mathcal{U}_\ell$ (refer to Appendix C). The regret of an algorithm $\pi \in \Pi$ is hence defined as: $R^\pi(T) = T\sum_\ell \beta_\ell p_{k_\ell^* \ell} - \sum_{t=1}^T \mathbb{E}\left[\sum_{k,\ell} \mathbb{1}_{\{u_t \in \mathcal{U}_\ell, i_t^\pi \in \mathcal{I}_k\}} p_{k\ell}\right].$

## 3.2 Preliminaries – User arrival process

The user arrival process is out of the decision maker's control and strongly impacts the performance of the recommendation algorithms. To analyze the regret of our algorithms, we will leverage the following results. Let $N_u(T)$ denote the number of requests of user $u$ up to round $T$. From the literature on "Balls and Bins process", see e.g. [27], we know that if $\overline{n} := \mathbb{E}[\max_{u \in \mathcal{U}} N_u(T)]$, then

$$\overline{n} = \begin{cases} \frac{\log(m)}{\log(\frac{m\log(m)}{T})}(1 + o(1)) & \text{if } T = o(m\log(m)), \\ \log(m)(d_c + o(1)) & \text{if } T = cm\log(m), \\ \frac{T}{m}(1 + o(1)) & \text{if } T = \omega(m\log(m)), \end{cases}$$

where $d_c$ is a constant depending on $c$ only. We also establish the following concentration result controlling the tail of the distribution of $N_u(T)$ (refer to Appendix B):

**Lemma 1.** *Define* $\overline{N} = \frac{4\log(m)}{\log(\frac{m\log(m)}{T}+e)} + \frac{e^2 T}{m}$. *Then,* $\forall u \in \mathcal{U}$, $\mathbb{E}[\max\{0, N_u(T) - \overline{N}\}] \leq \frac{1}{(e-1)m}$.

The quantities $\overline{n}$ and $\overline{N}$ play an important role in our regret analysis.

## 4 Regret Lower Bounds

In this section, we derive regret lower bounds for the three envisioned structures. Interestingly, we are able to quantify the minimal regret induced by the specific features of the problem: (i) the no-repetition constraint, (ii) the unknown success probabilities, (iii) the unknown item clusters, (iv) the unknown user clusters. The proofs of the lower bounds are presented in Appendices D-E-F.

## 4.1 Clustered items and statistically identical users

We denote by $\Delta_k = p_1 - p_k$ the gap between the success rates of items from the best cluster and of items from cluster $\mathcal{I}_k$, and introduce the function: $\phi(k, m, p) = \frac{1 - e^{-m\gamma(p_1, p_k)}}{8(1 - e^{-\gamma(p_1, p_k)})}$, where $\gamma(p, q) = \mathrm{kl}(p, q) + \mathrm{kl}(q, p)$ and $\mathrm{kl}(p, q) = p \log \frac{p}{q} + (1 - p) \log \frac{1-p}{1-q}$. Using the fact that $\mathrm{kl}(p, q) \leq (p - q)^2 / q(1 - q)$, we can easily show that as $m$ grows large, $\phi(k, m, p)$ scales as $\eta/(16\Delta_k^2)$ when $\Delta_k$ is small, where $\eta := \min_k p_k(1 - p_k)$.

We derive problem-specific regret lower bounds, and as in the classical stochastic bandit literature, we introduce the notion of *uniformly* good algorithm. $\pi$ is uniformly good if its expected regret $R^\pi(T)$ is $O(\max\{\sqrt{T}, \frac{\log(m)}{\log(\frac{m \log(m)}{T} + e)}\})$ for all possible system parameters $(p, \alpha)$ when $T, m, n$ grow large with $T = o(nm)$ and $m \geq n$. As shown in the next section, uniformly good algorithms exist.

**Theorem 1.** *Let $\pi \in \Pi$ be an arbitrary algorithm. The regret of $\pi$ satisfies: for all $T \geq 1$ such that $m \geq c/\Delta_2^2$ (for some constant $c$ large enough), $R^\pi(T) \geq \max\{R_{\mathrm{nr}}(T), R_{\mathrm{ic}}(T)\}$, where $R_{\mathrm{nr}}(T)$ and $R_{\mathrm{ic}}(T)$, the regrets due to the no-repetition constraint and to the unknown item clusters, respectively, are defined by $R_{\mathrm{nr}}(T) := \overline{n} \sum_{k \neq 1} \alpha_k \Delta_k$ and $R_{\mathrm{ic}}(T) := \frac{T}{m} \sum_{k \neq 1} \alpha_k \phi(k, m, p) \Delta_k$.*
*Assume that $\pi$ is uniformly good, then we have[2]: $R^\pi(T) \gtrsim R_{\mathrm{sp}}(T) := \log(T) \sum_{k \neq 1} \frac{\Delta_k}{2\mathrm{kl}(p_k, p_1)}$, where $R_{\mathrm{sp}}(T)$ refers to the regret due to the unknown success probabilities.*

From the above theorem, analyzing the way $R_{\mathrm{nr}}(T)$, $R_{\mathrm{ic}}(T)$, and $R_{\mathrm{sp}}(T)$ scale, we can deduce that:
(i) When $T = o(m \log(m))$, the regret arises mainly due to either the no-repetition constraint or the need to learn the success probabilities, and it scales at least as $\max\{\frac{\log(m)}{\log(\frac{m \log(m)}{T})}, \log(T)\}$.
(ii) When $T = cm \log(m)$, the three components of the regret lower bound scales in the same way, and the regret scales at least as $\log(T)$.
(iii) When $T = \omega(m \log(m))$, the regret arises mainly due to either the no-repetition constraint or the need to learn the item clusters, and it scales at least as $\frac{T}{m}$.

## 4.2 Unclustered items and statistically identical users

In this scenario, the regret is induced by the no-repetition constraint, and by the fact the success rate of an item when it is first selected and the distribution $\zeta$ are unknown. These two sources of regret lead to the terms $R_{\mathrm{nr}}(T)$ and $R_{\mathrm{i}}(T)$, respectively, in our regret lower bound.

**Theorem 2.** *Assume that the density of $\zeta$ satisfies, for some $C > 0$, $\zeta(\mu) \leq C$ for all $\mu \in [0, 1]$. Let $\pi \in \Pi$ be an arbitrary algorithm. Then its satisficing regret satisfies: for all $T \geq 1$ such that $m \geq c/\varepsilon^2$ (for some constant $c \geq 1$ large enough), $R_\varepsilon^\pi(T) \geq \max\{R_{\mathrm{nr}}(T), R_{\mathrm{i}}(T)\}$, where $R_{\mathrm{nr}}(T) := \overline{n} \int_0^{\mu_{1-\varepsilon}} (\mu_{1-\varepsilon} - \mu) \zeta(\mu) d\mu$ and $R_{\mathrm{i}}(T) := \frac{T}{m} \frac{\frac{(1-\varepsilon)^2}{2C}\left(1 - \frac{\varepsilon C}{1-\varepsilon}\right)^2}{\min\{1, (1+C)\varepsilon\} + 1/m}$.*

## 4.3 Clustered items and clustered users

To state regret lower bounds in this scenario, we introduce the following notations. For any $\ell \in [L]$, let $\Delta_{k\ell} = p_{k_\ell^* \ell} - p_{k\ell}$ be the gap between the success rates of items from the best cluster $\mathcal{I}_{k_\ell^*}$ and of items from cluster $\mathcal{I}_k$. We also denote by $\mathcal{R}_\ell = \{r \in [L] : k_\ell^* \neq k_r^*\}$. We further introduce the functions:

$$\phi(k, \ell, m, p) = \frac{1 - e^{-m\gamma(p_{k_\ell^* \ell}, p_{k\ell})}}{8\left(1 - e^{-\gamma(p_{k_\ell^* \ell}, p_{k\ell})}\right)} \quad \text{and} \quad \psi(\ell, k, T, m, p) = \frac{1 - e^{-\frac{T}{m}\gamma(p_{k_\ell^* \ell}, p_{k\ell})}}{8\left(1 - e^{-\gamma(p_{k_\ell^* \ell}, p_{k\ell})}\right)}.$$

Compared to the case of clustered items and statistically identical users, this scenario requires the algorithm to actually learn the user clusters. To discuss how this induces additional regret, assume that the success probabilities $p$ are known. Define $\mathcal{L}_\perp = \{(\ell, \ell') \in [L]^2 : p_{k_\ell^* \ell} \neq p_{k_{\ell'}^* \ell'}\}$, the set of pairs of user clusters whose best item clusters differ. If $\mathcal{L}_\perp \neq \emptyset$, then there isn't a single optimal item cluster for all users, and when a user $u$ first arrives, we need to learn its cluster. If $p$ is known, this classification generates at least a constant regret (per user) – corresponding to the term $R_{\mathrm{uc}}(T)$ in the

theorem below. For specific values of $p$, we show that this classification can even generate a regret scaling as $\log(T/m)$ (per user). This happens when $\mathcal{L}^\perp(\ell) = \{\ell' \neq \ell : k_\ell^* \neq k_{\ell'}^*, p_{k_\ell^* \ell} = p_{k_\ell^* \ell'}\}$ is not empty – refer to Appendix F for examples. In this case, we cannot distinguish users from $\mathcal{U}_\ell$ and $\mathcal{U}_{\ell'}$ by just presenting items from $\mathcal{I}_{k_\ell^*}$ (the greedy choice for users in $\mathcal{U}_\ell$). The corresponding regret term in the theorem below is $R'_{\mathrm{uc}}(T)$. To formalize this last regret component, we define uniformly good algorithms as follows. An algorithm is uniformly good if for any user $u$, $R_u^\pi(N) = o(N^\alpha)$ as $N$ grows large for all $\alpha > 0$, where $R_u^\pi(N)$ denotes the accumulated expected regret under $\pi$ for user $u$ when the latter has arrived $N$ times.

**Theorem 3.** *Let $\pi \in \Pi$ be an arbitrary algorithm. Then its regret satisfies: for all $T \geq 2m$ such that $m \geq c/\min_{k,\ell} \Delta_{k\ell}^2$ (for some constant $c$ large enough), $R^\pi(T) \geq \max\{R_{\mathrm{nr}}(T), R_{\mathrm{ic}}(T), R_{\mathrm{uc}}(T)\}$, where $R_{\mathrm{nr}}(T)$, $R_{\mathrm{ic}}(T)$, and $R_{\mathrm{uc}}(T)$ are regrets due to the no-repetition constraint, to the unknown item clusters, and to the unknown user clusters respectively, defined by:*

$$\begin{cases} R_{\mathrm{nr}}(T) := \overline{n} \sum_\ell \beta_\ell \sum_{k \neq k_\ell^*} \alpha_k \Delta_{k\ell}, \\ R_{\mathrm{ic}}(T) := \frac{T}{m} \sum_\ell \beta_\ell \sum_{k \neq k_\ell^*} \alpha_k \phi(k,\ell,m,p) \Delta_{k\ell}, \\ R_{\mathrm{uc}}(T) := m \sum_{\ell \in [L]} \beta_\ell \frac{\sum_{k \in \mathcal{R}_\ell} \Delta_{k\ell} \psi(\ell,k,T,m,p)}{K}. \end{cases}$$

*In addition, when $T = \omega(m)$, if $\pi$ is uniformly good, $R^\pi(T) \gtrsim R'_{\mathrm{uc}}(T) := c(\beta,p) m \log(T/m)$ where $c(\beta,p) = \inf_{n \in \mathcal{F}} \sum_\ell \beta_\ell \sum_{k \neq k_\ell^*} \Delta_{k\ell} n_{k\ell}$ with*

$$\mathcal{F} = \{n \geq 0 : \forall \ell, \ \forall \ell' \in \mathcal{L}^\perp(\ell), \sum_{k \neq k_\ell^*} \mathrm{kl}(p_{k\ell}, p_{k\ell'}) n_{k\ell} \geq 1\}.$$

Note that we do not include in the lower bound the term $R_{\mathrm{sp}}(T)$ corresponding to the regret induced by the lack of knowledge of the success probabilities. Indeed, it would scale as $\log(T)$, and this regret would be negligible compared to $R_{\mathrm{uc}}(T)$ (remember that $T = o(m^2)$), should $\mathcal{L}_\perp \neq \emptyset$. Under the latter condition, the main component of regret is for any time horizon is due to the unknown user clusters. When $\mathcal{L}_\perp \neq \emptyset$, the regret scales at least as $m$ if for all $\ell$, $\mathcal{L}^\perp(\ell) = \emptyset$, and $m \log(T/m)$ otherwise.

# 5 Algorithms

This section presents algorithms for our three structures and an analysis of their regret. The detailed pseudo-codes of our algorithms and numerical experiments are presented in Appendix A. The proofs of the regret upper bounds are postponed to Appendices G-H-I.

## 5.1 Clustered items and statistically identical users

To achieve a regret scaling as in our lower bounds, the structure needs to be exploited. Even without accounting for the no-repetition constraint, the KL-UCB algorithm would, for example, yield a regret scaling as $\frac{n}{\Delta_2} \log(T)$. Now we could first sample $T/m$ items and run KL-UCB on this restricted set of items – this would yield a regret scaling as $\frac{T}{m\Delta_2} \log(T)$, without accounting for the no-repetition constraint. Our proposed algorithm, Explore-Cluster-and-Test (ECT), achieves a better regret scaling and complies with the no-repetition constraint. Refer to Appendix A for numerical experiments illustrating the superiority of ECT.

**The Explore-Cluster-and-Test algorithm.** ECT proceeds in the following phases:

(a) **Exploration phase.** This first phase consists in gathering samples for a subset $\mathcal{S}$ of randomly selected items so that the success probabilities and the clusters of these items are learnt accurately. Specifically, we pick $|\mathcal{S}| = \lfloor \log(T)^2 \rfloor$ items, and for each of these items, gather roughly $\log(T)$ samples.

(b) **Clustering phase.** We leverage the information gathered in the exploration phase to derive an estimate $\hat{\rho}_i$ of the success probability $\rho_i$ for item $i \in \mathcal{S}$. These estimates are used to cluster items, using an appropriate version of the K-means algorithm. In turn, we extract from this phase, accurate estimates $\hat{p}_1$ and $\hat{p}_2$ of the success rates of items in the two best item clusters, and a set $\mathcal{V} \subset \mathcal{S}$ of items believed to be in the best cluster: $\mathcal{V} := \{i \in \mathcal{S} : \hat{\rho}_i > (\hat{p}_1 + \hat{p}_2)/2\}$.

(c) **Test phase.** The test phase corresponds to an exploitation phase. Whenever this is possible (the no-repetition constraint is not violated), items from $\mathcal{V}$ are recommended. When an item outside $\mathcal{V}$

has to be selected due to the no-repetition constraint, we randomly sample and recommend an item outside $\mathcal{V}$. This item is appended to $\mathcal{V}$. To ensure that any item $i$ in the (evolving) set $\mathcal{V}$ is from the best cluster with high confidence, we keep updating its empirical success rate $\hat{\rho}_i$, and periodically test whether $\hat{\rho}_i$ is close enough from $\hat{p}_1$. If this is not the case, $i$ is removed from $\mathcal{V}$.

In all phases, ECT is designed to comply with the no-repetition constraint: for example, in the exploration phase, when the user arrives, if we cannot recommend an item from $\mathcal{S}$ due to the constraint, we randomly select an item not violating the constraint. In the analysis of ECT regret, we upper bound the regret generated in rounds where a random item selection is imposed. Observe that ECT does not depend on any parameter (except for the choice of the number of items initially explored in the first phase).

**Theorem 4.** *We have:* $R^{\mathrm{ECT}}(T) = \mathcal{O}\left( \frac{2\overline{N}}{\alpha_1} \sum_{k=2}^{K} \frac{\alpha_k(p_1-p_k)}{(p_1-p_2)^2} + (\log T)^3 \right)$.

The regret lower bound of Theorem 1 states that for any algorithm $\pi$, $R^\pi(T) = \Omega(\overline{N})$, and if $\pi$ is uniformly good $R^\pi(T) = \Omega(\max\{\overline{N}, \log(T)\})$. Thus, in view of the above theorem, ECT is order-optimal if $\overline{N} = \Omega((\log T)^3)$, and order-optimal up to an $(\log T)^2$ factor otherwise. Furthermore, note that when $R_{\mathrm{ic}}(T) = \Omega(\frac{T}{\Delta_2 m})$ is the leading term in our regret lower bound, ECT regret has also the right scaling in $\Delta_2$: $R^{\mathrm{ECT}}(T) = \mathcal{O}(\frac{T}{\Delta_2 m})$.

## 5.2 Unclustered items and statistically identical users

When items are not clustered, we propose ET (Explore-and-Test), an algorithm that consists of two phases: an exploration phase that aims at estimating the threshold level $\mu_{1-\varepsilon}$, and a test phase where we apply to each item a sequential test to determine whether the item if above the threshold.

**The Explore-and-Test algorithm.** The ET algorithm proceeds as follows.

(a) **Exploration phase.** In this phase, we randomly select of set $\mathcal{S}$ consisting of $\lfloor \frac{8^2}{\varepsilon^2} \log T \rfloor$ items and recommend each selected item to $\lfloor 4^2 \log T \rfloor$ users. For each item $i \in \mathcal{S}$, we compute its empirical success rate $\hat{\rho}_i$. We then estimate $\mu_{1-\frac{\varepsilon}{2}}$ by $\hat{\mu}_{1-\frac{\varepsilon}{2}}$ defined so that: $\frac{\varepsilon}{2}|\mathcal{S}| = \left| \{ i \in \mathcal{S} : \hat{\rho}_i \geq \hat{\mu}_{1-\frac{\varepsilon}{2}} \} \right|$. We also initialize the set $\mathcal{V}$ of candidate items to exploit as $\mathcal{V} = \{ i \in \mathcal{S} : \hat{\rho}_i \geq \hat{\mu}_{1-\frac{\varepsilon}{2}} \}$.

(b) **Test phase.** In this phase, we recommend items in $\mathcal{V}$, and update the set $\mathcal{V}$. Specifically, when a user $u$ arrives, we recommend the item $i \in \mathcal{V}$ that has been recommended the least recently among items that would not break the no-repetition constraint. If no such items exist in $\mathcal{V}$, we randomly recommend an item outside $\mathcal{V}$ and add it to $\mathcal{V}$.
Now to ensure that items in $\mathcal{V}$ are above the threshold, we perform the following sequential test, which is reminiscent of sequential tests used in optimal algorithms for infinite bandit problems [2]. For each item, the test is applied when the item has been recommended for the $\lfloor 2^\ell \log \log_2(2^e m^2) \rfloor$ times for any positive integer $\ell$. For the $\ell$-th test, we denote by $\bar{\rho}^{(\ell)}$ the real number such that $\mathrm{kl}(\bar{\rho}^{(\ell)}, \hat{\mu}_{1-\frac{\varepsilon}{2}}) = 2^{-\ell}$. If $\bar{\rho}^{(\ell)} \leq \hat{\mu}_{1-\frac{\varepsilon}{2}}$, the item is removed from $\mathcal{V}$.

**Theorem 5.** *Assume that the density of $\zeta$ satisfies $\zeta(\mu) \leq C$ for all $\mu \in [0, 1]$.*
*For any $\varepsilon \geq C\sqrt{\frac{\pi}{2 \log T}}$, we have:* $R_\varepsilon^{\mathrm{ET}}(T) = \mathcal{O}\left( \overline{N} \frac{\log(1/\varepsilon) \log \log(m)}{\varepsilon} + \frac{(\log T)^2}{\varepsilon^2} \right)$.

In view of Theorem 2, the regret of any algorithm scales at least as $\Omega(\frac{\overline{N}}{\varepsilon})$. Hence, the above theorem states that ET is order-optimal at least when $\overline{N} = \Omega((\log T)^2)$.

## 5.3 Clustered items and clustered users

The main challenge in devising an algorithm in this setting stems from the fact that we do not control the user arrival process. In turn, clustering users with low regret is delicate. We present Explore-Cluster with Upper Confidence Sets (EC-UCS), an algorithm that essentially exhibits the same regret scaling as our lower bound. The idea behind the design of EC-UCS is as follows. We estimate the success rates $(p_{k\ell})_{k,\ell}$ using small subsets of items and users. Then based on these estimates, each user is optimistically associated with a UCS, *Upper Confidence Set*, a set of clusters the user may likely belong to. The UCS of a user then shrinks as the number of requests made by this user increases (just as the UCB index of an arm in bandit problems gets closer to its average reward).

The design of our estimation procedure and of the various UCS is made so as to get an order-optimal algorithm. In what follows, we assume that $m^2 \geq T(\log T)^3$ and $T \geq m \log(T)$.

**The Explore-Cluster-with-Upper Confidence Sets algorithm.**

(a) **Exploration and item clustering phase.** The algorithm starts by collecting data to infer the item clusters. It randomly selects a set $\mathcal{S}$ consisting of $\min\{n, \lfloor \frac{m}{(\log T)^2} \rfloor\}$ items. For the $10m$ first user arrivals, it recommends items from $\mathcal{S}$ uniformly at random. These $10m$ recommendations and the corresponding user responses are recorded in the dataset $\mathcal{D}$. From the dataset $\mathcal{D}$, the item clusters are extracted using a spectral algorithm (see Algorithm 4 in the appendix). This algorithm is taken from [33], and considers the *indirect edges* between items created by users. Specifically, when a user appears more than twice in $\mathcal{D}$, she creates an indirect edge between the items recommended to her for which she provided the same answer (1 or 0). Items with indirect edges are more likely to belong to the same cluster. The output of this phase is a partition of $\mathcal{S}$ into item clusters $\hat{I}_1, \ldots, \hat{I}_K$. We can show that with an exploration budget of $10m$, w.h.p. at least $m/2$ indirect edges are created and that in turn, the spectral algorithm does not make any clustering errors w.p. at least $1 - \frac{1}{T}$.

(b) **Exploration and user clustering phase.** To the $(10 + \log(T))m$ next user arrivals, EC-UCS clusters a subset of users using a Nearest-Neighbor algorithm. The algorithm selects a subset $\mathcal{U}^*$ of users to cluster, and recommendations to the remaining users will be made depending some distance to the inferred clusters in $\mathcal{U}^*$. Users from all clusters must be present in $\mathcal{U}^*$. To this aim, EC-UCS first randomly selects a subset $\mathcal{U}_0$ of $\lfloor m/\log(T) \rfloor$ users from which it extracts the set $\mathcal{U}^*$ of $\lfloor \log(T)^2 \rfloor$ users who have been observed the most. The extraction and the clustering of $\mathcal{U}^*$ is made several times until the $\lfloor (10 + \log(T))m \rfloor$-th user arrives so as to update and improve the user clusters. From these clusters, we deduce estimates $\hat{p}_{k\ell}$ of the success probabilities.

(c) **Recommendations based on Optimistic Assignments.** After the $10m$-th arrivals, recommendations are made based on the estimated $\hat{p}_{k\ell}$'s. For user $u_t \notin \mathcal{U}_0$, the item selection further depends on the $\hat{\rho}_{ku_t}$'s, the empirical success rates of user $u_t$ for items in the various clusters. A greedy recommendation for $u_t$ would consist in assigning $u_t$ to cluster $\ell$ minimizing $\|\hat{p}_{\cdot\ell} - \hat{\rho}_{\cdot u_t}\|$ over $\ell$, and then in picking an item from cluster $\hat{I}_k$ with maximal $\hat{p}_{k\ell}$. Such a greedy recommendation would not work as when $u_t$ has not been observed many times, the cluster she belongs to remains uncertain. To address this issue, we apply the Optimism in Front of Uncertainty principle often used in bandit algorithms to foster exploration. Specifically, we build a set $\mathcal{L}(u_t)$ of clusters $u_t$ is likely to belong to. $\mathcal{L}(u_t)$ is referred to as the Upper Confidence Set of $u_t$. As we get more observations of $u_t$, this set shrinks. Specifically, we let $x_{k\ell} = \max\{|\hat{p}_{k\ell} - \hat{\rho}_{ku_t}| - \epsilon, 0\}$, for some well defined $\epsilon > 0$ (essentially scaling as $\sqrt{\log\log(T)/\log(T)}$, see Appendix A for details), and define $\mathcal{L}(u_t) = \{\ell \in [L] : \sum_k x_{k\ell}^2 n_{ku_t} < 2K \log(n_{u_t})\}$ ($n_{u_t}$ is the number of time $u_t$ has arrived, and $n_{ku_t}$ is the number of times $u_t$ has been recommended an item from cluster $\hat{I}_k$). After optimistically composing the set $\mathcal{L}(u_t)$, $u_t$ is assigned to cluster $\ell$ chosen uniformly at random in $\mathcal{L}(u_t)$, and recommended an item from cluster $\hat{I}_k$ with maximal $\hat{p}_{k\ell}$.

**Theorem 6.** *For any $\ell$, let $\sigma_\ell$ be the permutation of $[K]$ such that $p_{\sigma_\ell(1)\ell} > p_{\sigma_\ell(2)\ell} \geq \cdots \geq p_{\sigma_\ell(K)\ell}$. Let $\mathcal{R}_\ell = \{r \in [L] : k_\ell^* \neq k_r^*\}$, $\mathcal{S}_{\ell r} = \{k \in [K] : p_{k\ell} \neq p_{kr}\}$, $y_{\ell r} = \min_{k \in \mathcal{S}_{\ell r}} |p_{k\ell} - p_{kr}|$, $\delta = \min_\ell(p_{\sigma_\ell(1)\ell} - p_{\sigma_\ell(2)\ell})$, and $\phi(x) := x/\log(1/x)$. Then, we have:*

$$
R^{\text{EC-UCS}}(T) = \mathcal{O}\left( m \sum_\ell \beta_\ell(p_{\sigma_\ell(1)\ell} - p_{\sigma_\ell(K)\ell}) \left( \max\left( \frac{K^3 \log K}{\phi(\min(y_{\ell r}, \delta)^2)}, \frac{\sqrt{K}}{\min_\ell \beta_\ell} \right) \right. \right.
$$

$$
\left. \left. + \sum_{r \in \mathcal{R}_\ell \setminus \mathcal{L}^\perp(\ell)} \frac{K^2 \log K}{\phi(|p_{k_\ell^* r} - p_{k_\ell^* \ell}|^2)} + \sum_{k \in \mathcal{S}_{\ell r}} \sum_{r \in \mathcal{L}^\perp(\ell)} \frac{K \log \overline{N}}{|\mathcal{S}_{\ell r}| |p_{k\ell} - p_{kr}|^2} \right) \right).
$$

EC-UCS blends clustering and bandit algorithms, and its regret analysis is rather intricate. The above theorem states that remarkably, the regret of the EC-UCS algorithm matches our lower bound order-wise. In particular, the algorithm manages to get a regret (i) scaling as $m$ whenever it is possible, i.e., when $\mathcal{L}^\perp(\ell) = \emptyset$ for all $\ell$, (ii) scaling as $m \log(\overline{N})$ otherwise.

In Appendix A.3, we present ECB, a much simpler algorithm than EC-UCS, but whose regret upper bound, derived in Appendix J, always scales as $m \log(\overline{N})$.

# 6 Conclusion

This paper proposes and analyzes several models for online recommendation systems. These models capture both the fact that items cannot repeatedly be recommended to the same users and some underlying user and item structure. We provide regret lower bounds and algorithms approaching these limits for all models. Many interesting and challenging questions remain open. We may, for example, investigate other structural assumptions for the success probabilities (e.g. soft clusters), and adapt our algorithms. We may also try to extend our analysis to the very popular linear reward structure, but accounting for no-repetition constraint.

## Broader Impact

This work, although mostly theoretical, may provide guidelines and insights towards an improved design of recommendation systems. The benefits of such improved design could be to increase user experience with these systems, and to help companies to improve their sales strategies through differentiated recommendations. The massive use of recommendation systems and its potential side effects have recently triggered a lot of interest. We must remain aware of and investigate such effects. These include: opinion polarization, a potential negative impact on users' behavior and their willingness to pay, privacy issues.

## Acknowledgements

K. Ariu was supported by the Nakajima Foundation Scholarship. S. Yun and N. Ryu were supported by Institute of Information & communications Technology Planning & Evaluation (IITP) grant funded by the Korea government(MSIT)(No.2019-0-00075, Artificial Intelligence Graduate School Program(KAIST)). A. Proutiere's research is supported by the Wallenberg AI, Autonomous Systems and Software Program (WASP) funded by the Knut and Alice Wallenberg Foundation.

## Footnotes

[1]For this unstructured scenario, we will justify why considering the satisficing regret is needed.

[2]We write $a \gtrsim b$ if $\liminf_{T \to \infty} a/b \geq 1$.

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
