[Supplementary Material]

# Supplementary Material:

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

# 7 Table of Notations

| Notations common to all models | |
|---|---|
| $n$ | Number of items |
| $m$ | Number of users |
| $\mathcal{I}$ | Set of items |
| $\mathcal{U}$ | Set of users |
| $u_t$ | User requesting recommendation at round $t$ |
| $T$ | Time horizon |
| $X_{iu}$ | Binary random variable to indicate whether user $u$ likes the item $i$ |
| $\rho = (\rho_{iu})_{i \in \mathcal{I}, u \in \mathcal{U}}$ | Probability that the user $u$ likes the item $i$ |
| $\pi$ | Algorithm for sequential item selection |
| $\Pi$ | Set of all algorithms for sequential item selection |
| $i_t^\pi$ | Item selected at round $t$ under $\pi$ |
| $\mathcal{F}_{t-1}^\pi$ | $\sigma$-algebra generated by $(u_t, (u_s, i_s^\pi, X_{i_s^\pi u_s}), s \leq t-1)$ |
| $\overline{n}$ | Term $\mathbb{E}[\max_{u \in \mathcal{U}} N_u(T)]$ |
| $\overline{N}$ | Term $\frac{4 \log(m)}{\log\left(\frac{m \log(m)}{T} + e\right)} + \frac{e^2 T}{m}$ |

| Generic notations | |
|---|---|
| $\hat{a}$ | Estimated value of $a$ |
| $\sigma(A)$ | $\sigma$-algebra generated by $A$ |
| $\mathrm{kl}(p, q)$ | Kullback–Leibler divergence from Bernoulli random variable with parameter $p$ to that with parameter $q$ |
| $\gtrsim$ | We write $a \gtrsim b$ if $\liminf_{T \to \infty} a/b \geq 1$ |

Table 1: Table of notations common to all models

| Model A: Clustered items and statistically identical users | |
|---|---|
| $\mathcal{I}_k$ | Set of items in the item cluster $k$ |
| $\alpha = (\alpha_k)_{k \in [K]}$ | Probability that an item is assigned to the item cluster $k$ |
| $K$ | Number of item clusters |
| $\Delta$ | Minimum difference between the success rates of items in optimal cluster and of items in sub-optimal cluster |
| $p = (p_k)_{k \in [K]}$ | Probability that the user likes the item $i \in \mathcal{I}_k$ |
| $R^\pi(T)$ | Regret of an algorithm $\pi$ |
| $\Delta_k$ | Term $p_1 - p_k$ |
| $\phi(k, m, p)$ | Term $\frac{1 - e^{-m\gamma(p_1, p_k)}}{8(1 - e^{-\gamma(p_1, p_k)})}$ |
| $\gamma(p, q)$ | Term $\mathrm{kl}(p, q) + \mathrm{kl}(q, p)$ |
| $\eta$ | Term $\min_k p_k(1 - p_k)$ |
| $\mathcal{S}$ | Set of initially sampled items |
| $\mathcal{V}$ | Set of items believed to be in the best cluster |

Table 2: Table of notations: Model A

| Model B: Unclustered items and statistically identical users | |
| --- | --- |
| $\zeta$ | Distribution over $[0, 1]$ |
| $\mu_x$ | Term $\inf\{\gamma \in [0, 1] : \mathbb{P}[\rho_i \leq \gamma] \geq x\}$ |
| $\varepsilon$ | Constant that specifies the $\varepsilon$-best items |
| $R_\varepsilon^\pi(T)$ | Satisficing regret of algorithm $\pi$ with a given $\varepsilon > 0$ |
| $C$ | Constant that regularizes the distribution $\zeta(\mu)$ |
| $\mathcal{S}$ | Set of initially sampled items |
| $\mathcal{V}$ | Set of items believed to be in the best cluster |

Table 3: Table of notations: Model B

| Model C: Clustered items and clustered users | |
| --- | --- |
| $\mathcal{I}_k$ | Set of items in the item cluster $k$ |
| $\alpha = (\alpha_k)_{k \in [K]}$ | Probability that an item is assigned to the item cluster $k$ |
| $\mathcal{U}_\ell$ | Set of users in the user cluster $\ell$ |
| $\beta = (\beta_\ell)_{\ell \in [L]}$ | Probability that a user is assigned to the user cluster $\ell$ |
| $K$ | Number of item clusters |
| $L$ | Number of user clusters |
| $p = (p_{k\ell})_{k \in [K], \ell \in [L]}$ | Probability that the user $u$ likes the item $i$ such that $i \in \mathcal{I}_k$ and $u \in \mathcal{U}_\ell$ |
| $k_\ell^*$ | Term $\arg\max_k p_{k\ell}$ |
| $\Delta_{k\ell}$ | Term $p_{k_\ell^* \ell} - p_{k\ell}$ |
| $\Delta$ | Minimum difference between the success rates of items in optimal cluster and of items in sub-optimal cluster |
| $\delta_\ell$ | Term $\min_{k:\Delta_{k\ell}>0} \Delta_{k\ell}$ |
| $\phi(k, \ell, m, p)$ | Term $\frac{1-e^{-m\gamma(p_{k_\ell^* \ell}, p_{k\ell})}}{8\left(1-e^{-\gamma(p_{k_\ell^* \ell}, p_{k\ell})}\right)}$ |
| $\psi(\ell, k, T, m, p)$ | Term $\frac{1-e^{-\frac{T}{m}\gamma(p_{k_\ell^*}, p_k)}}{8\left(1-e^{-\gamma(p_{k_\ell^*}, p_k)}\right)}$ |
| $\mathcal{L}_\perp$ | Set $\{(\ell, \ell') \in [L]^2 : p_{k_\ell^* \ell} \neq p_{k_{\ell'}^* \ell'}\}$ |
| $\mathcal{L}^\perp(\ell)$ | Set $\{\ell' \neq \ell : k_\ell^* \neq k_{\ell'}^*, p_{k_\ell^* \ell} = p_{k_\ell^* \ell'}\}$ |
| $R_u^\pi(N)$ | Accumulated expected regret under $\pi$ for user $u$ when the user has arrived $N$ times |
| $R^\pi(T)$ | Regret of an algorithm $\pi$ |
| $\mathcal{S}$ | Set of initially sampled items |
| $\mathcal{U}_0$ | Set of initially sampled users |
| $\mathcal{U}^*$ | Set of $(\log T)^2$ users in $\mathcal{U}_0$ who have been arrived the most |
| $\mathcal{L}(u_t)$ | Upper Condifence Set of the user $u_t$ |
| $\sigma_\ell$ | Permutation of $[K]$ such that $p_{\sigma_\ell(1)\ell} > p_{\sigma_\ell(2)\ell} \geq \cdots \geq p_{\sigma_\ell(K)\ell}$ |
| $\mathcal{R}_\ell$ | Set $\{r \in [L] : k_\ell^* \neq k_r^*\}$ |
| $\mathcal{S}_{\ell r}$ | Set $\{k \in [K] : p_{k\ell} \neq p_{kr}\}$ |
| $y_{\ell r}$ | Term $\min_{k \in \mathcal{S}_{\ell r}} |p_{k\ell} - p_{kr}|$ |
| $\delta$ | Term $\min_\ell (p_{\sigma_\ell(1)\ell} - p_{\sigma_\ell(2)\ell})$ |
| $\epsilon$ | Term $K\sqrt{\frac{8Km}{t}\log\frac{t}{m}}$ (Updated only when the user clustering is executed) |

Table 4: Table of notations: Model C

# A Algorithms and experiments

In this section, we present the detailed pseudo-codes of our algorithms. We also illustrate the performance of these algorithms numerically.

## A.1 Clustered items and statistically identical users

---
**Algorithm 1** Explore-Cluster-and-Test

---
**Input:** $T, K$
**1. Exploration**
Sample a set $\mathcal{S}$ of $\lfloor (\log T)^2 \rfloor$ items (uniformly at random)
Recommend each item in $\mathcal{S}$ to $\lfloor \log T \rfloor$ users
(when this is not possible due to the no-repetition constraint) recommend a random feasible item.
$T_0 \leftarrow$ round where the exploration phase ends
**2. Clustering**
$\hat{\rho}_i \leftarrow$ the empirical average of $i$ for all $i \in \mathcal{S}$
$Q_i \leftarrow \{j \in \mathcal{S} : |\hat{\rho}_i - \hat{\rho}_j| \leq (\log T)^{-\frac{1}{4}}\}$ for all $i \in \mathcal{S}$
$M \leftarrow \emptyset$
**for** $k = 1$ **to** $K$ **do**
   $i_k \leftarrow \arg\max_{j \in \mathcal{S}} |Q_j \setminus \cup_{\ell=1}^{k-1} Q_{i_\ell}|$
   **if** $|Q_{i_k}| < \log T$ **then**
     **break**
   **end if**
   $M \leftarrow M \cup \{i_k\}$
**end for**
$i_1 \leftarrow \arg\max_{i \in M} \hat{\rho}_i$ and $\hat{p}_1 \leftarrow \hat{\rho}_{i_1}$
$i_2 \leftarrow \arg\max_{i \in M \setminus \{i_1\}} \hat{\rho}_i$ and $\hat{p}_2 \leftarrow \hat{\rho}_{i_2}$
**3. Test**
$\Delta_0 \leftarrow \hat{p}_1 - \hat{p}_2$
$\mathcal{V}, \mathcal{V}_0 \leftarrow \{i \in \mathcal{S} : \hat{\rho}_i > (\hat{p}_1 + \hat{p}_2)/2\}$
**for** $t = (T_0 + 1)$ **to** $T$ **do**
   Recommend item from $\mathcal{V}$ with the highest empirical average if possible, otherwise randomly
   recommend item $i$ from $\mathcal{I} \setminus \mathcal{V}_0$ and add $i$ to $\mathcal{V}$ and $\mathcal{V}_0$
   **if** the number of times $i$ has been recommended is a multiple of $\lfloor \frac{2 \log 3}{\Delta_0^2} \rfloor$ and $\hat{\rho}_i < (\hat{p}_1 + \hat{p}_2)/2$
   **then**
     Remove $i$ from $\mathcal{V}$
   **end if**
**end for**

---

**Numerical experiments.** We illustrate the performance of ECT in the following scenario: $K = 2$ item clusters, $n = 3000$ items, $m = 5000$ users, $p_1 = 0.7$, $p_2 = 0.2$, $\alpha_1 = \alpha_2 = 0.5$.

We compare the performance of ECT to two naive algorithms:

(i) B-KLUCB [7]: This algorithm was proposed for budgeted bandits. Here the budget per arm is $m$. The algorithm ranks the arms (the items) according to their KL-UCB indexes, and selects the available item (accounting for the no-repetition constraint) with the highest index.

(ii) B-KLUCB with sampling: The algorithm first samples $\lfloor (T/m) \log T \rfloor$ items randomly, and play B-KLUCB only for these items. When none of these items can be played, the algorithm plays a randomly selected item (as in ECT).

Figure 1 plots the regret vs time for the 3 algorithms, for a time horizon $T = 20000$. The regret is averaged over 200 runs for ECT and B-KLUCB with sampling, and 20 runs for B-KL-UCB (we do not need more runs since there is no randomness induced by the initial item sampling procedure). For ECT, the number of items initially sampled is $|\mathcal{S}| = \lfloor 0.225 \log(T)^2 \rfloor$, whereas for B-KLUCB, it is 39 (this number is optimized so as to get the best performance – refer to Figure 3 for a sensitivity analysis of the regret depending on the number of items initially sampled).

Figure 1: Regret vs time averaged over 200 instances for ECT and B-KLUCB with sampling, and over 20 instances for B-KLUCB. $T = 20000$, $n = 3000$, $m = 5000$, $p_1 = 0.7$, $p_2 = 0.2$, $\alpha_1 = \alpha_2 = 0.5$. Shaded areas correspond to one standard deviations.

Figure 2: Regret vs time averaged over 100 instances for each algorithm. $T = 100000$, $n = 3000$, $m = 5000$, $p_1 = 0.7$, $p_2 = 0.2$, $\alpha_1 = \alpha_2 = 0.5$. Shaded areas correspond to one standard deviations.

Figure 2 compares the regret of ECT to that obtained under B-KLUCB with sampling when $T = 100000$. ECT initially samples 29 items. For B-KLUCB with sampling, we have tested two different numbers of items initially sampled, namely 29 and 230. After round 20000, ECT starts playing items that have not being used in the exploration phase. To keep regret low, ECT hence relies on sequential tests. The regret curve of ECT shows that these tests perform very well. This contrasts with B-KLUCB with sampling: when the number of initially sampled items is 29, as for ECT, after round 20000, new items must be selected, and B-KLUCB performs very poorly (the regret rapidly grows).

Finally, we assess the sensitivity of ECT and B-KLUCB with sampling w.r.t. the number of initially sampled items. Figure 3 plots the regret after $T = 20000$ rounds depending on this number. Again

Figure 3: Regret at $T = 20000$ vs number of items initially sampled $|\mathcal{S}|$ for ECT and B-KLUCB with sampling, averaged over 200 instances. $T = 20000$, $n = 3000$, $m = 5000$, $p_1 = 0.7$, $p_2 = 0.2$, $\alpha_1 = \alpha_2 = 0.5$. One standard deviations are shown in the error bars.

we average over 200 runs. ECT is not very sensitive to the number of sampled items; B-KLUCB is, on the other hand, very sensitive. For ECT, this provides further evidence that the sequential tests applied to items not used in the exploration phase are very efficient.

## A.2 Unclustered items and statistically identical users

---

**Algorithm 2** Explore-and-Test

---

**Input:** $T, K$

**1. Exploration**

Sample a set $\mathcal{S}$ of $\lfloor \frac{8^2}{\varepsilon^2} \log T \rfloor$ items (uniformly at random)

Recommend each item in $\mathcal{S}$ to $\lfloor 4^2 \log T \rfloor$ users

(when this is not possible due to the no-repetition constraint) recommend a random feasible item

$\hat{\rho}_i \leftarrow$ the empirical average of $i$ for all $i \in \mathcal{S}$

$T_0 \leftarrow$ round where the exploration phase ends

**2. Test**

Compute $\hat{\mu}_{1-\frac{\varepsilon}{2}}$ s.t. $\frac{\varepsilon}{2}|\mathcal{S}| = \left| \{ i \in \mathcal{S} : \hat{\rho}_i \geq \hat{\mu}_{1-\frac{\varepsilon}{2}} \} \right|$

$\mathcal{V} \leftarrow \{ i \in \mathcal{S} : \hat{\rho}_i \geq \hat{\mu}_{1-\frac{\varepsilon}{2}} \}$

Reset the reward observation history

**for** $t = (T_0 + 1)$ **to** $T$ **do**

    Recommend an item $i$ that was recommended the least recently among items in $\mathcal{V}$. If items in $\mathcal{V}$ cannot be selected, recommend an item $i$, randomly selected from the set of unrecommended items, and add $i$ to $\mathcal{V}$

    **if** the number of times $i$ has been recommended is exactly $\lfloor 2^\ell \log \log_2(2^e m^2) \rfloor$ for some positive integer $\ell$ **then**

        Compute $\bar{\rho}^{(\ell)} (\leq \hat{\mu}_{1-\frac{\varepsilon}{2}})$ s.t. $\mathrm{kl}(\bar{\rho}^{(\ell)}, \hat{\mu}_{1-\frac{\varepsilon}{2}}) = 2^{-\ell}$

        **if** $\hat{\rho}_i \leq \bar{\rho}^{(\ell)}$ **then**

            $\mathcal{V} \leftarrow \mathcal{V} \setminus \{i\}$

        **end if**

    **end if**

**end for**

---

**Numerical experiments.** Consider a system with $n = 700$ items, and $m = 1300$ users. The time horizon is $T = 100000$. Assume that the distribution $\zeta$ is uniform over the interval $[0.1, 0.9]$, and let us target items within the 30% best items, i.e., $\varepsilon = 0.3$. In Figure 4, we compare the satisficing regret averaged over 100 runs of the ET algorithm with $|\mathcal{S}| = 65$ items used in the exploration phase, to that achieved under B-KLUCB with sampling. Since under any algorithm, one needs to use at least $T/m$ items, the number of items sampled under B-KLUCB with sampling is chosen as $\min((T/m) \log T, n)$, which is equal to $n = 700$ in our setting. Figure 4 illustrates the efficiency of the sequential tests used under ET.

Next, we assess the sensitivity of ET and B-KLUCB w.r.t. the number of initially sampled items. Figure 5 compares the satisficing regret after $T = 100000$ depending on this number. The values are averaged over 20 runs. ET seems robust to the number of sampled items. B-KLUCB is, however, very sensitive to the number of sampled items and shows larger regret than that of ET. This result presents further evidence that the sequential testing procedures used in ET are efficient.

Figure 4: Satisficing regret vs time averaged over 100 runs of ET and B-KLUCB with sampling. $n = 700, m = 1300, T = 100000, \varepsilon = 0.3$. $\zeta$ is uniform on $[0.1, 0.9]$. Shaded areas correspond to one standard deviations.

Figure 5: Satisficing regret at $T = 100000$ vs number of items initially sampled $|\mathcal{S}|$ for ET and B-KLUCB with sampling, averaged over 20 instances. $n = 700, m = 1300, T = 100000, \varepsilon = 0.3$. $\zeta$ is uniform on $[0.1, 0.9]$. One standard deviations are shown in the error bars.

### A.3 Clustered items and users

Here we start by providing the description of our order-optimal algorithm, EC-UCS. We then present ECB (Explore-Cluster-Bandit), a much simpler algorithm but with lower performance guarantees.

#### A.3.1 The EC-UCS algorithm

We present the pseudo-code of the EC-UCS algorithm in Algorithm 3. The algorithm calls spectral clustering algorithms whose pseudo-codes are provided in Algorithms 4-5-6.

---

**Algorithm 3** Explore-and-Cluster-with-Upper-Confidence-Sets (EC-UCS)

---

Input: $T, K, L$
**1. Exploration for Item Clustering**
$n_0 \leftarrow \min\{n, \lfloor m/(\log T)^2 \rfloor\}$
Sample a set $\mathcal{S}$ of $n_0$ items.
**for** $1 \leq t \leq 10m$ **do**
    Recommend items from $\mathcal{S}$ randomly (when this is not possible due to the no-repetition constraint, recommend a random feasible item). Record the user responses in the dataset $\mathcal{D}$.
**end for**
**2. Item Clustering**
Run Algorithm 4 with input $\mathcal{S}, \mathcal{D}, K$ and output $\hat{I}_1, \ldots, \hat{I}_K$.
**3. Exploitation**
$\mathcal{U}_0 \leftarrow$ a set of randomly chosen $\lfloor \frac{m}{\log T} \rfloor$ users

$\hat{\rho}_u = (\hat{\rho}_{uk})_{k \in [K]} \leftarrow$ the empirical average of $u$ for each item cluster $\hat{I}_k$ for all $u \in \mathcal{U}$
**for** $10m < t \leq T$ **do**
    **if** $t \leq \lfloor (10 + \log T)m \rfloor$ and $t = (9 + 2^i)m + 1$ for some non-negative integer $i$ **then**
        $\mathcal{U}^* \leftarrow$ a set of $\lfloor (\log T)^2 \rfloor$ users in $\mathcal{U}_0$ who have been observed the most
        $\epsilon \leftarrow K\sqrt{\frac{8Km}{t} \log \frac{t}{m}}$
        $Q_u \leftarrow \{v \in \mathcal{U}^* : \|\hat{\rho}_u - \hat{\rho}_v\| \leq \epsilon\}$ for all $u \in \mathcal{U}^*$
        **for** $\ell = 1$ **to** $L$ **do**
            $u_\ell \leftarrow \arg\max_{v \in U^*} |Q_v \setminus \cup_{r=1}^{\ell-1} Q_{u_r}|$ and $\hat{p}_\ell \leftarrow \hat{\rho}_{u_\ell}$
            $L_0 \leftarrow \ell$
            **if** $\cup_{r=1}^{\ell} Q_{i_r} = \mathcal{U}^*$ **then**
                **break**
            **end if**
        **end for**
    **end if**
    **if** $u_t \in \mathcal{U}_0$ and $t \leq \lfloor (10 + \log T)m \rfloor$ **then**
        Recommend an item in a round-robin fashion from $\hat{I}_1, \ldots, \hat{I}_K$ for each user
    **else**
        $x_{k\ell} \leftarrow \max\{|\hat{p}_{k\ell} - \hat{\rho}_{ku_t}| - \epsilon, 0\}$
        $\mathcal{L}(u_t) \leftarrow \{\ell \in [L_0] : \sum_{k=1}^K n_{ku_t} x_{k\ell}^2 < 2K \log n_{u_t}\}$
        **if** $|\mathcal{L}(u_t)| \geq 1$ **then**
            Recommend an item uniformly at random from $(\hat{I}_{k_\ell^*})_{\ell \in \mathcal{L}(u_t)}$
        **else**
            Recommend an item uniformly at random from $\hat{I}_1, \ldots, \hat{I}_K$
        **end if**
    **end if**
    Update $\hat{\rho}_{u_t}$
**end for**

---

---
**Algorithm 4** Spectral Item Clustering by indirect edges (inspired by Algorithm 1 in [33])
---
Input: $\mathcal{S}, \mathcal{D}, K$
$A \leftarrow \mathbf{0} \in \mathbb{R}^{\mathcal{S} \times \mathcal{S}}, s \leftarrow 0$
**for** $u \in \mathcal{U}$ **do**
    **if** $u$ has received recommendations at least two times **then**
        $s \leftarrow s + 1$
        **if** user $u$ gives positive responses to both $i$ and $j$ that are the first two recommended items to user $u$ **then**
            $A_{ij} \leftarrow A_{ij} + 1, A_{ji} \leftarrow A_{ji} + 1$
        **end if**
    **end if**
**end for**
Run Algorithm 5 with input $A$, $s$, $K$ and output $\hat{I}_1, \ldots, \hat{I}_K$.
Output: $\hat{I}_1, \ldots, \hat{I}_K$
---

---
**Algorithm 5** Spectral Partitioning+ (an improved version of Algorithm 2 in [26]
---
Input: Observation matrix $A$, $s$, $K$
**1. Spectral Decomposition**
Run Algorithm 6, with input $A$, $K$ and output $(S_k)_{k=1,\ldots,K}$.
**2. Improvement**
$\hat{p}(i,j) \leftarrow \frac{\sum_{v \in S_i} \sum_{v' \in S_j} A_{v,v'}}{|S_i|s}$ for all $1 \leq i, j \leq K$
$S_k^{(0)} \leftarrow S_k$ for all $1 \leq k \leq K$
**for** $t = 1$ **to** $\lfloor \log n_0 \rfloor$ **do**
    $S_k^{(t)} \leftarrow \emptyset$ for all $1 \leq k \leq K$
    **for** $v \in \mathcal{S}$ **do**
        $i^* \leftarrow \arg\max_{1 \leq i \leq K}$
        $\left\{ \sum_{k=0}^{K} (\sum_{w \in S_k^{(t-1)}} A_{vw}) \log \hat{p}(i,k) \right\}$
            where $\sum_{w \in S_0^{(t-1)}} A_{vw} := s - \sum_{k=1}^{K} \sum_{w \in S_k^{(t-1)}} A_{vw}$
            and $\hat{p}(i,0) := 1 - \sum_{k=1}^{K} \hat{p}(i,k)$ (ties are broken uniformly at random)
            $S_{i^*}^{(t)} \leftarrow S_{i^*}^{(t)} \cup \{v\}$
    **end for**
**end for**
$\hat{I}_k \leftarrow S_k^{(\log n)}$ for all $1 \leq k \leq K$
Output: $\hat{I}_1, \ldots, \hat{I}_K$
---

**Algorithm 6** Spectral Decomposition (Algorithm 3 in [26])

---

Input: Observation matrix $A$, $K$

$\hat{A} \leftarrow$ rank-$K$ approximation of $A$

$\tilde{p} \leftarrow \frac{\sum_{v,w \in \mathcal{I}} A_{vw}}{|\mathcal{S}|(|\mathcal{S}|-1)}$

**for** $x = 1$ to $\lfloor \log n_0 \rfloor$ **do**

$\quad Q_v^{(x)} \leftarrow \left\{ w \in \mathcal{S} : \|\hat{A}_w - \hat{A}_v\|^2 \leq x \frac{\tilde{p}}{100} \right\}$ for all $v \in \mathcal{S}$

$\quad T_l^{(x)} \leftarrow \emptyset$ for all $k \in [K]$

$\quad$**for** $k = 1$ **to** $K$ **do**

$\quad\quad v_k^* \leftarrow \arg\max_{v \in \mathcal{I}} \left| Q_v^{(x)} \setminus \cup_{i=1}^{k-1} T_i^{(x)} \right|$

$\quad\quad T_k^{(x)} \leftarrow Q_{v_k^*}^{(x)} \setminus \cup_{i=1}^{k-1} T_i^{(x)}$

$\quad\quad \xi_k^{(x)} \leftarrow \sum_{v \in T_k^{(x)}} \frac{\hat{A}_v}{|T_k^{(x)}|}$

$\quad$**end for**

$\quad$**for** $v \in \mathcal{S} \setminus \cup_{k=1}^{K} T_k^{(x)}$ **do**

$\quad\quad k^* \leftarrow \arg\min_{1 \leq k \leq K} \|\hat{A}_v - \xi_k^{(x)}\|^2$

$\quad\quad T_{k^*}^{(x)} \leftarrow T_{k^*}^{(x)} \cup \{v\}$

$\quad$**end for**

$\quad r_x \leftarrow \sum_{k=1}^{K} \sum_{v \in T_k^{(x)}} \|\hat{A}_v - \xi_k^{(x)}\|^2$

**end for**

$x^* \leftarrow \arg\min_x r_x$

$S_k \leftarrow T_k^{(x^*)}$ for all $k \in [K]$

Output: $S_1, \ldots, S_K$

---

### A.3.2 The ECB algorithm

The ECB algorithm presented in Algorithm 7. ECB achieves a regret scaling as $O(m \log(\overline{N}))$ for all $p$ (ECB treats each user independently, and does not transfer the information gathered across users). The algorithm proceeds as follows.

(a)-(b) **Exploration and clustering phases.** (b) These phases are identical to those of EC-UCS. The algorithm outputs item cluster estimates $\hat{I}_1, \ldots, \hat{I}_K$. We can show that with an exploration budget of $10m$, the spectral algorithm does not make any clustering errors w.p. at least $1 - \frac{1}{T}$.

(c) **Bandit phase.** The last phase consists in just applying $m$ (one for each user) UCB1 algorithms [1] with the set of arms $1, \ldots, K$. There, selecting arm $k$ means recommending an item from $\hat{I}_k$, accounting for the no-repetition constraint (which is possible w.h.p. since $m^2 \geq T(\log T)^3$).

ECB calls the clustering algorithm presented in Algorithm 4, that first constructs an item adjacency matrix (using indirect edges from users), and then applies the spectral clustering algorithm, Algorithm 5, to output $K$ item clusters. Note that Algorithm 5 further calls the spectral decomposition algorithm, shown in Algorithm 6.

We have the following performance guarantee on ECB (the proof is presented in Appendix J.1):

**Theorem 7.** *When $m^2 \geq T(\log T)^3$, the regret of ECB satisfies:*

$$R^{\text{ECB}}(T) = \mathcal{O}\left( \sum_{\ell=1}^{L} \beta_\ell m \sum_{k \neq k_\ell^*} \frac{\log(\overline{N})}{p_{k_\ell^* \ell} - p_{k\ell}} \right).$$

---

**Algorithm 7** Explore-Cluster-Bandit

---

Input: $T, K$

**1. Exploration for Item Clustering**

$n_0 \leftarrow \min\{n, \lfloor m/(\log T)^2 \rfloor\}$

Sample a set $\mathcal{S}$ of $n_0$ items.

**for** $1 \leq t \leq 10m$ **do**

    Recommend items from $\mathcal{S}$ randomly (when this is not possible due to the no-repetition constraint, recommend a random feasible item). Record the user responses in the datatset $\mathcal{D}$.

**end for**

**2. Item Clustering**

Run Algorithm 4 with input $\mathcal{S}, \mathcal{D}, K$ and output $\hat{I}_1, \ldots, \hat{I}_K$.

**3. Run UCB1 [1] for each user**

**for** $10m < t \leq T$ **do**

    $k^* \leftarrow \arg\max_{k\in[K]} \hat{\rho}_k^{u_t} + \sqrt{\frac{2\ln(t)}{N_k^{u_t}(t)}}$, where $\hat{\rho}_k^{u_t}$ is the empirical average reward of user $u_t$ for items in $\hat{I}_k$ and $N_k^{u_t}(t)$ is the total number of samples of user $u_t$ for items in $\hat{I}_k$.

    Recommend an item from $\hat{I}_{k^*}$ randomly (when this is not possible due to the no-repetition constraint) recommend a random feasible item.

**end for**

---

### A.3.3 Numerical Experiment

Consider a system with $n = 2000$ items and $m = 5000$ users. The time horizon is $T = 800000$. The statistical parameters of $p_{k\ell}$ are given in Table 5. $\alpha_1 = \alpha_2 = \frac{1}{2}$ and $\beta_1 = \beta_2 = \frac{1}{2}$. With this parameter setting, for each $\ell = 1, 2$, $\mathcal{L}^\perp(\ell) = \emptyset$. From Theorems 6 and 7, we know that the regret of EC-UCS is $R^{\text{EC-UCS}}(T) = \mathcal{O}(m)$ whereas that of ECB is $R^{\text{ECB}}(T) = \mathcal{O}(m\log(\overline{N}))$. Hence, we expect that EC-UCS to outperform ECB. Figure 6 shows the regret evolution over time of EC-UCS algorithm and ECB algorithm after the item clustering phase. The curves are averaged over 10 instances. The rate at which the regret of EC-UCS increases is rapidly decreasing. This is not the case for that of the regret of ECB.

Next, we assess the regret of the two algorithms after $T$ rounds as a function of $T$. We consider a system with $n = 3000$ items and $m = 5000$ users. $\alpha_1 = \alpha_2 = \frac{1}{2}$ and $\beta_1 = \beta_2 = \frac{1}{2}$. The statistical parameters of $p_{k\ell}$ are the same as in the previous system. Figure 7 presents the results. Here, the regrets are averages over 20 runs. The regret of ECB clearly increases with $T$, while the regret of EC-UCS does not seem to be sensitive to $T$. Overall, our results confirm our theoretical results, at least on simple examples.

|          | $k = 1$ | $k = 2$ |
|----------|---------|---------|
| $\ell = 1$ | 0.2     | 0.8     |
| $\ell = 2$ | 0.8     | 0.2     |

Table 5: The values of $(p_{k\ell})$.

Figure 6: Regret vs time after the item clustering phase, averaged over 10 runs of EC-UCS and ECB. $T = 800000$, $n = 2000$, $m = 5000$, $\alpha_1 = \alpha_2 = 0.5$, $\beta_1 = \beta_2 = 0.5$. One standard deviations are shown as the shaded areas.

Figure 7: Regret at round $T$ vs budget $T$ after the item clustering phase, averaged over 20 runs of EC-UCS and ECB. $n = 3000$, $m = 5000$, $\alpha_1 = \alpha_2 = 0.5$, $\beta_1 = \beta_2 = 0.5$. One standard deviations are shown as the shaded areas.

## A.4 Experimental set-up

The simulations were performed on a desktop computer with Intel Core i7-8700B 3.2 GHz CPU and 32 GB RAM.

# B   Preliminaries: Properties of the user arrival process

This section presents several preliminary results on the user arrival process, extensively used throughout the proofs of the main theorems. Here we also provide the proof of Lemma 1.

**Lemma 2** (Chernoff-Hoeffding theorem). *Let $X_1, \ldots, X_n$ be i.i.d. Bernoulli random variables with mean $\nu$. Then, for any $\delta > 0$,*

$$\mathbb{P}\left(\frac{1}{n}\sum_{i=1}^{n} X_i \geq \nu + \delta\right) \leq \exp\left(-n\mathrm{kl}(\nu + \delta, \nu)\right)$$

$$\mathbb{P}\left(\frac{1}{n}\sum_{i=1}^{n} X_i \leq \nu - \delta\right) \leq \exp\left(-n\mathrm{kl}(\nu - \delta, \nu)\right)$$

**Lemma 3** (Pinsker's inequality [31]). *For any $0 \leq p, q \leq 1$, $2(p - q)^2 \leq \mathrm{kl}(p, q)$.*

**Lemma 4.** *For any $0 \leq p, q \leq 1$, $\mathrm{kl}(p, q) \geq p\log\frac{p}{q} + (q - p)$.*

**Proof.** This follows from $(1 - p)\log\frac{1-p}{1-q} \geq q - p$. $\qquad\square$

**Proof of Lemma 1.** This lemma is quoted below for convenience:

**Lemma 1 (restated).** *For every user $u \in \mathcal{U}$, we have*

$$\mathbb{E}\left[\max\left\{0, N_u(T) - \overline{N}\right\}\right] \leq \frac{1}{m(e - 1)}$$

*where*

$$\overline{N} = \frac{4\log m}{\log\left(\frac{m\log m}{T} + e\right)} + \frac{e^2 T}{m}.$$

*Proof.* Since $u$ arrives with probability $\frac{1}{m}$ in each round, the probability that $u$ arrives for more than $\overline{N} + x$ times in the $T$ first round is,

$$
\begin{aligned}
\mathbb{P}\left(N_u(T) \geq \overline{N} + x\right) &\overset{(a)}{\leq} \exp\left(-T\mathrm{kl}\left(\frac{\overline{N} + x}{T}, \frac{1}{m}\right)\right) \\
&\overset{(b)}{\leq} \exp\left(-(\overline{N} + x)\log\left(\frac{m(\overline{N} + x)}{T}\right) - \frac{T}{m} + (\overline{N} + x)\right) \\
&\overset{(c)}{\leq} \exp\left(-\frac{\overline{N} + x}{2}\log\left(\frac{m(\overline{N} + x)}{T}\right)\right) \\
&\overset{(d)}{\leq} \exp\left(-\frac{\overline{N}}{2}\log\left(\frac{m\overline{N}}{T}\right)\right)\exp(-x)
\end{aligned}
$$

where (a) follows from Lemma 2, (b) from Lemma 4, (c) is obtained from the fact that $\frac{m\overline{N}}{T} \geq e^2$, and (d) holds since $x \geq 0$ and $\log\left(\frac{m\overline{N}}{T}\right) \geq 2$. We deduce that:

$$
\begin{aligned}
\mathbb{E}\left[\max\left\{0, N_u(T) - \overline{N}\right\}\right] &= \sum_{x=1}^{\infty} \mathbb{P}\left(N_u(T) \geq \overline{N} + x\right) \\
&\leq \sum_{x=1}^{\infty} \exp\left(-\frac{\overline{N}}{2}\log\left(\frac{m\overline{N}}{T}\right)\right)\exp(-x) \\
&= \frac{\exp\left(-\frac{\overline{N}}{2}\log\left(\frac{m\overline{N}}{T}\right)\right)}{e - 1}.
\end{aligned}
\tag{1}
$$

To conclude the proof, we compute an upper bound of (1) for two cases: $\frac{T}{m} \geq \frac{\log m}{e}$ and $\frac{T}{m} \leq \frac{\log m}{e}$.

If $\frac{T}{m} \geq \frac{\log m}{e}$, then $\frac{\overline{N}}{2} > \log m$ and $\log\left(\frac{m\overline{N}}{T}\right) \geq 2$. Thus,

$$\exp\left(-\frac{\overline{N}}{2}\log\left(\frac{m\overline{N}}{T}\right)\right) < \frac{1}{m^2}. \tag{2}$$

We now consider the case where $\frac{T}{m} \leq \frac{\log m}{e}$. When we define $f(x) = \log\left(\frac{4x}{\log(x+e)}\right) - \frac{\log(x+e)}{2}$, one can easily check that $f(e) \geq 0$ and $f'(x) \geq 0$ for all $x \geq e$. Therefore, since $\frac{m\log m}{T} \geq e$, we can deduce that

$$\frac{2\log m}{\log\left(\frac{m\log m}{T} + e\right)}\log\left(\frac{4m\log m}{T\log\left(\frac{m\log m}{T} + e\right)}\right) \geq \log m$$

which directly implies that

$$\exp\left(-\frac{\overline{N}}{2}\log\left(\frac{m\overline{N}}{T}\right)\right) < \frac{1}{m}. \tag{3}$$

Lemma 1 is obtained by combining (1), (2) and (3). $\qquad \square$

The following Lemma characterizes the lower tail of the number of user arrivals.

**Lemma 5.** *For every user $u \in \mathcal{U}$, we have*

$$\mathbb{P}\left(N_u(T) \leq \frac{T}{2m}\right) \leq \exp\left(-\frac{T}{2m}(1 - \log(2))\right).$$

**Proof.**

$$\mathbb{P}\left(N_u(T) \leq \frac{T}{2m}\right) \overset{(a)}{\leq} \exp\left(-T\mathrm{kl}\left(\frac{1}{2m}, \frac{1}{m}\right)\right)$$

$$\overset{(b)}{\leq} \exp\left(-T\left(\frac{1}{2m}\log(1/2) + \left(\frac{1}{m} - \frac{1}{2m}\right)\right)\right)$$

$$= \exp\left(-\frac{T}{2m}(1 - \log(2))\right),$$

where $(a)$ is from Lemma 2 and $(b)$ is from Lemma 4. $\qquad \square$

The next lemma is instrumental in the performance analysis of the EC-UCS and ECB algorithms (for systems with clustered items and users).

**Lemma 6.** *With probability $1 - \frac{1}{T}$, at least $\frac{m}{2}$ users arrive at least two times among the first $10m$ arrivals.*

**Proof.** We denote by $N_u(10m)$ the number of times user $u$ arrives in the $10m$ first arrivals. For any set $A \subset \mathcal{U}$, let $N(A, 10m)$ denote the total number of arrivals of users in $A$ among the first $10m$ arrivals.

We write the probability that less than $m/2$ users arrive twice in the $10m$ first arrivals as:

$$\mathbb{P}[\sum_u \mathbb{1}_{\{N_u(10m)\geq 2\}} < m/2] = \mathbb{P}[\sum_u \mathbb{1}_{\{N_u(10m)<2\}} \geq m/2]$$

$$\leq \mathbb{P}[\exists A : |A| = \frac{m}{2}, \forall u \in A, N_u(10m) < 2]$$

$$\leq \mathbb{P}[\exists A : |A| = \frac{m}{2}, N(A, 10m) \leq \frac{m}{2}]$$

$$\leq \binom{m}{m/2}\exp\left(-20m(\frac{1}{2} - \frac{1}{10})^2\right)$$

$$\leq (2e)^{m/2}\exp\left(-\frac{9}{5}m\right)$$

$$\leq e^{-0.8m} \leq \frac{1}{T^2}. \tag{4}$$

$\square$

## C   Justifying the regret definitions

In this section, we justify our definitions of regret for Models A, B and C. In these models, we define regret as if an Oracle policy would always be able to select for any user $u$ an item from the best cluster for Models A and C for this user, or an $\varepsilon$-best item for Model B, even under the no-repetition constraint. In fact, the definition of the true regret should account for the no-repetition constraint and in turn for the fact that an Oracle policy may be obliged to select items that do not belong to the best cluster for the user $u$, because user $u$ may arrive too often before the time horizon $T$ or because the size of this cluster is too small (remember that when an item, selected for the first time, is randomly assigned to a cluster). We prove here that the difference between our notion of regret and the true regret is actually negligible when compared to any of the terms involved in our regret lower bounds, namely $T/m$ and $\bar{n}$.

To establish this claim, recall the assumptions made on $(n, m, T)$. $T = o(nm)$, $m \geq n$, and $\log(m) = o(n)$ (and as a consequence, for any $\beta > 0$ independent of $(n, m, T)$, $m = o(e^{\beta n})$).

For illustrative purposes, we prove our claim for Model A (the same result holds for Models B and C). Let $Z_u$ denote the (random) number of items in the best cluster for user $u$. It is easy to show that the difference between our notion of regret $R^\pi(T)$ and the true regret $R^\pi_{true}(T)$ satisfies:

$$|R^\pi(T) - R^\pi_{true}(T)| \leq \sum_{u \in \mathcal{U}} \mathbb{E}[\max\{N_u(T) - Z_u, 0\}].$$

In Model A, $Z_u$ is the size of cluster $\mathcal{I}_1$. The average size of $\mathcal{I}_1$ is $\mathbb{E}[Z_u] = \alpha_1 n$. Let $\epsilon > 0$ be such that $\epsilon < \alpha_1$ and $\frac{\epsilon nm}{T} \geq e^2$. Using the same arguments as those leading to Lemma 1, we obtain the following concentration result:

$$\mathbb{E}\Big[ \max\Big\{0, N_u(T) - \epsilon n\Big\}\Big] \leq \frac{1}{(e-1)} \exp(-\frac{\epsilon n}{2} \log(\frac{\epsilon nm}{T})).$$

In addition, we also have as a direct application of Chernoff-Hoeffding inequality presented in Lemma 2:

$$\mathbb{P}[Z_u \leq \epsilon n] \leq \exp(-n\mathrm{kl}(\epsilon, \alpha_1)).$$

From the two above inequalities, we deduce that:

$$\mathbb{E}[\max\{N_u(T) - Z_u, 0\}] \leq \frac{T}{m}\mathbb{P}[Z_u \leq \epsilon n] + \mathbb{E}\Big[ \max\Big\{0, N_u(T) - \epsilon n\Big\}\Big]$$
$$\leq \frac{T}{m} \exp(-n\mathrm{kl}(\epsilon, \alpha_1)) + \exp(-\frac{\epsilon n}{2} \log(\frac{\epsilon nm}{T})).$$

We conclude that:

$$|R^\pi(T) - R^\pi_{true}(T)| \leq \underbrace{T \exp(-n\mathrm{kl}(\epsilon, \alpha_1))}_{=A(n,m,T)} + \underbrace{m \exp(-\frac{\epsilon n}{2} \log(\frac{\epsilon nm}{T}))}_{=B(n,m,T)}.$$

Next we verify that $\max\{A(n, m, T), B(n, m, T)\} = o(\min\{\frac{T}{m}, \bar{n}\})$. This will be enough to justify our definition of regret, since our regret lower bounds are all larger than $\min\{\frac{T}{m}, \bar{n}\}$.

(i) Let us check that $\max\{A(n, m, T), B(n, m, T)\} = o(\frac{T}{m})$. Indeed, for $A(n, m, T)$, we have $m = o(e^{n\mathrm{kl}(\epsilon, \alpha_1)})$; as for $B(n, m, T)$, we have $m^2/T = o(e^{\epsilon n})$ which results from $m = o(e^{\epsilon n/2})$.

(ii) Let us check that $\max\{A(n, m, T), B(n, m, T)\} = o(\bar{n})$. We consider the three regimes for $\bar{n}$:

1. When $T = \omega(m \log(m))$. Then $\bar{n} = T/m(1 + o(1))$ and we conclude as in (i).

2. When $T = \Theta(m \log(m))$. To simplify, we just prove the statement for $T = cm \log(m)$. Then $\bar{n} = \log(m)(d_c + o(1))$ and $A(n, m, T) = m \log(m) \exp(-n\mathrm{kl}(\epsilon, \alpha_1))$. We hence conclude that $A(n, m, T) = \bar{n}$ since $m = o(e^{n\mathrm{kl}(\epsilon, \alpha_1)})$. Now $B(n, m, T) = m \exp(-\frac{\epsilon n}{2} \log(\frac{\epsilon n}{c \log(m)}))$, and $B(n, m, T) = o(\bar{n})$ is a consequence of $m = o(e^{\frac{\epsilon n}{2}})$.

3. When $T = o(m \log(m))$. Then $\bar{n} = \frac{\log(m)}{\log(m \log(m)/T)}(1 + o(1))$. $A(n, m, T) = o(\bar{n})$ is equivalent to $X = o(e^{n\mathrm{kl}(\epsilon, \alpha_1)})$ where $X = \frac{T}{\log(m)} \log(\frac{m \log(m)}{T})$. Now $X = o(m)$ since

$T = o(m \log(m))$, and thus $A(n, m, T) = o(\bar{n})$. Finally, $B(n, m, T) = o(\bar{n})$ is equivalent to $Y = o(\exp(\frac{\epsilon n}{2} \log(\epsilon n m/T)))$ where $Y = \frac{m \log(m \log(m)/T)}{\log(m)}$. Since $T = o(nm)$, this would be implied by $Y = o(\exp(\frac{\epsilon n}{2}))$. However, since $T \geq 1$, $Y \leq \frac{m \log(m \log(m))}{\log(m)} = m(1 + o(1))$. We conclude that $B(n, m, T) = o(\bar{n})$ since $m = o(e^{\frac{\epsilon n}{2}})$.

# D   Fundamental limits for Model A: Proof of Theorem 1

Proof of $R^\pi(T) \geq R_{\mathrm{ic}}(T)$: Let $\pi \in \Pi$. Assume that the success probabilities $p_k$'s are known. Further simplify the problem by relaxing the no-repetition constraint: instead, we just impose that any item cannot be selected more than $m$ times. The algorithm $\pi$ has an expected regret larger than an optimal algorithm for the problem where $p_k$'s are known, and the no-repetition constraint is relaxed as explained above. Denote by $\tau$ this optimal algorithm. Next we establish a regret lower bound for $\tau$. To this aim, we first consider that the cluster ids of the items are drawn before the first round. That way, all the $n$ items belong to a cluster even before the first round. Let $J_1, \ldots, J_K$ denote the sizes of the various clusters of items. Hence, we work conditioning on the cluster ids of the items. Now, define $\mathbb{E}^\tau[N_k]$ as the expected number of times an item of cluster $\mathcal{I}_k$ is selected under $\tau$ (of course, $N_k \leq m$) until round $T$. More precisely, denote by $(n_1, n_2, \ldots, n_{J_k})$ the random variables representing the numbers of times the first, the second, the third, etc. items in $\mathcal{I}_k$ are selected under $\tau$. Then by definition: $\mathbb{E}^\tau[N_k] = \mathbb{E}^\tau[\sum_{i=1}^{J_k} n_i]/J_k$. We prove:

**Lemma 7.** *We have for any $m \geq c/\Delta_2^2$ and for any $k \neq 1$:* $\frac{\phi(k,m,p)}{m} \leq \frac{\mathbb{E}^\tau[N_k]}{\mathbb{E}^\tau[N_1]} \leq 1$.

The constant $c$ in the above lemma is the same as that in Theorem 1. It is chosen such that if $m \geq c/\Delta_2^2$, then, for any $k$, $\phi(k,m,p) \leq m$ and thus, the first inequality of Lemma 7 makes sense. Remember that $\phi(k,m,p)$ scales as $1/\Delta_k^2$ (refer to the remark above Theorem 1), and hence such a choice for $c$ is possible.

Lemma 7 is proved at the end of this section. Assume for now that it holds. We complete the proof by deriving a lower bound of the optimal algorithm $\tau$, $R^\tau(T)$. When the item clusters are fixed, the expected conditional regret of $\tau$ is: $R^\tau(T|\mathcal{I}_1, \ldots, \mathcal{I}_K) = \sum_{k \neq 1} J_k \mathbb{E}^\tau[N_k]\Delta_k$. Hence we have:

$$R^\tau(T|\mathcal{I}_1, \ldots, \mathcal{I}_K) = T\frac{\sum_{k \neq 1} J_k \mathbb{E}^\tau[N_k]\Delta_k}{\sum_k J_k \mathbb{E}^\tau[N_k]} \stackrel{(a)}{\geq} \frac{T}{m} \frac{\sum_{k \neq 1} J_k \mathbb{E}^\tau[N_1]\phi(k,m,p)\Delta_k}{\sum_k J_k \mathbb{E}^\tau[N_1]}$$
$$= \frac{T}{m} \sum_{k \neq 1} \frac{J_k}{n} \phi(k,m,p)\Delta_k,$$

where (a) stems from Lemma 7 (in the numerator, we use $\mathbb{E}^\tau[N_k] \geq \mathbb{E}^\tau[N_1]\frac{\phi(k,m,p)}{m}$, and in the denominator $\mathbb{E}^\tau[N_k] \leq \mathbb{E}^\tau[N_1]$). Taking the expectation of the above inequality (noting that $\mathbb{E}[J_k/n] = \alpha_k$), we conclude that: $R^\pi(T) \geq R^\tau(T) \geq \frac{T}{m} \sum_{k \neq 1} \alpha_k \phi(k,m,p)\Delta_k$.

Proof of $R^\pi(T) \geq R_{\mathrm{nr}}(T)$: Observe that under the no-repetition constraint, the number of items that $\pi$ will select is greater than $\max_u N_u(T)$. Now when an item is selected for the first time, by assumption, it belongs to the sub-optimal cluster $\mathcal{I}_k$ with probability $\alpha_k$, in which case this initial selection induces an expected regret $\Delta_k = p_1 - p_k$. Hence $R^\pi(T) \geq \mathbb{E}[\max_u N_u(T)] \sum_{k \neq 1} \alpha_k \Delta_k = \overline{n} \sum_{k \neq 1} \alpha_k \Delta_k$. $\qquad\square$

Proof of $R^\pi(T) \gtrsim R_{\mathrm{sp}}(T)$: To prove this asymptotic lower bound, we consider a simpler problem: the algorithm knows the item clusters $(\mathcal{I}_k)_{k \in [K]}$. Then, the problem reduces to a $K$-armed Bernoulli bandit problem with unknown parameters $(p_k)_{k \in [K]}$.

The proof then proceeds using a classical change-of-measure argument as in [19]. We present this argument for completeness. Assume that the algorithm $\pi$ is uniformly good. Pick any $k \in [K]$ s.t. $k \neq 1$. Let $p = (p_k)_{k \in [K]}$ be original parameters and let $(p'_k)_{k \in [K]}$ be perturbed parameters where $\forall k' \neq k$, $p'_{k'} = p_{k'}$ and $p'_k = p_1 + \varepsilon$ with some constant $\varepsilon > 0$. We denote $N_k(T) := \sum_{t=1}^T \mathbb{1}\{i_t^\pi \in \mathcal{I}_k\}$. We use $\mathbb{E}_p[\cdot]$ (or $\mathbb{E}[\cdot]$) and $\mathbb{E}_{p'}[\cdot]$ to denote the expectation under the original model and under the perturbed model, respectively.

Let $L_T$ be the log-likelihood defined as:

$$L_T := \sum_{t=1}^T \mathbb{1}\{i_t^\pi \in \mathcal{I}_k\}\left(\mathbb{1}\{X_{i_t^\pi u_t} = 1\} \log\left(\frac{p_k}{p_1 + \varepsilon}\right) + \mathbb{1}\{X_{i_t^\pi u_t} = 0\} \log\left(\frac{1 - p_k}{1 - (p_1 + \varepsilon)}\right)\right).$$

Taking the expectation under $p$, we have

$$\mathbb{E}_p[L_T] = \mathbb{E}_p[N_k(T)]\mathrm{kl}(p_k, p_1 + \varepsilon) \overset{(a)}{\geq} \mathrm{kl}\left(\frac{\mathbb{E}_p[N_k(T)]}{T}, \frac{\mathbb{E}_{p'}[N_k(T)]}{T}\right)$$

$$\overset{(b)}{\geq} \left(1 - \frac{\mathbb{E}_p[N_k(T)]}{T}\right)\log\left(\frac{T}{T - \mathbb{E}_{p'}[N_k(T)]}\right) - \log 2.$$

where $(a)$ stems from the *data processing inequality*, see [8], and $(b)$ is from the fact that for all $(x, y) \in [0, 1]^2$, $\mathrm{kl}(x, y) \geq (1 - x)\log\frac{1}{1-y} - \log 2$. By the uniform goodness assumption, with some constant $C > 0$, we have:

$$\mathbb{E}_p[N_k(T)] \lesssim C\sqrt{T} \quad \text{and} \quad T - \mathbb{E}_{p'}[N_k(T)] \lesssim C\sqrt{T}.$$

Hence:

$$\frac{1}{\log(T)}\log\frac{T}{T - \mathbb{E}_{p'}[N_k(T)]} \gtrsim \frac{1}{\log(T)}\log\frac{T}{C\sqrt{T}} \gtrsim \frac{1}{2}.$$

The inequality $\mathbb{E}_p[N_k(T)]\mathrm{kl}(p_k, p_1 + \varepsilon) \gtrsim \frac{1}{2}$ holds for any $\varepsilon > 0$. Therefore, we have:

$$\mathbb{E}_p[N_k(T)]\mathrm{kl}(p_k, p_1) \gtrsim \frac{1}{2}.$$

Thus, we get the regret lower bound:

$$R^\pi(T) = \sum_{k \neq 1} \Delta_k \mathbb{E}_p[N_k(T)] \gtrsim \left(\sum_{k \neq 1}\frac{\Delta_k}{2\mathrm{kl}(p_k, p_1)}\right)\log(T).$$

This concludes the proof of Theorem 1. $\qquad\qquad\qquad\qquad\qquad\qquad\qquad\qquad\qquad \square$

## D.1  Proof of Lemma 7

To establish the lemma, we build, from the optimal algorithm $\tau$, an algorithm $\zeta$ that can be applied to a 2-armed bandit problem with known expected rewards $p_1$ and $p_k$. We then provide a connection between the regret of $\zeta$ in the 2-armed bandit problem and $\mathbb{E}^\tau[N_k]$. We conclude the proof by establishing a regret lower bound for $\zeta$, using similar techniques as in [5].

**2-armed bandit problem with known rewards and the algorithm $\zeta$.** Consider a 2-armed bandit problem with Bernoulli arms 1 and $k$ of means $p_1$ and $p_k$. The means are known but the arm with the highest mean is unknown. That is to say that the expected reward of arm 1 can be either $p_1$ and $p_k$. For this bandit problem, we build an algorithm, denoted by $\zeta$, based on the algorithm $\tau$.

1. Pick $i_1$ and $i_k$ uniformly at random in the clusters $\mathcal{I}_1$ and $\mathcal{I}_k$. We run $\tau$ for $T$ rounds. When $\tau$ selects item $i_1$ (resp. $i_k$), then $\zeta$ also selects arm 1 (resp. $k$).

2. We repeat the above Step 1, to determine the arm selections made by $\zeta$. At the beginning of the successive episodes of $T$ rounds, the items $i_1$ and $i_k$ are again chosen uniformly at random in the clusters $\mathcal{I}_1$ and $\mathcal{I}_k$, independently of the choices made in earlier episodes.

**Regret of $\zeta$ and its connection to $\mathbb{E}^\tau[N_k]$.** Consider an episode of $T$ rounds for $\tau$. Let $i_k$ denote the item selected from $\mathcal{I}_k$ in the design of $\zeta$, the expected regret accumulated by $\zeta$ in this episode is $\Delta_k \mathbb{E}^\tau[n_{i_k}]$, where $n_{i_k}$ is the number of times $\tau$ selects $i_k$ in the episode. Since $i_k$ is chosen uniformly at random, and by definition of $\mathbb{E}^\tau[N_k]$, we actually have $\mathbb{E}^\tau[n_{i_k}] = \mathbb{E}^\tau[N_k]$, which connects the regret of $\zeta$ and $\mathbb{E}^\tau[N_k]$.

Next, assume that we stop the algorithm $\zeta$ after $\kappa$ episodes of $T$ rounds, where $\kappa$ is the first episode where $\zeta$ has made more than $m$ selections. $\kappa$ is a random variable, and Wald's first lemma implies that the expected regret $R^\zeta$ accumulated by $\zeta$ before we stop playing is:

$$R^\zeta = \Delta_k \mathbb{E}[\kappa]\mathbb{E}^\tau[N_k].$$

Since $\max\{\mathbb{E}^\tau[N_1], \mathbb{E}^\tau[N_k]\} \leq m$, we have $\mathbb{E}[\kappa]\mathbb{E}^\tau[N_1] \leq 2m$. Hence:

$$R^\zeta = \Delta_k \mathbb{E}[\kappa]\mathbb{E}^\tau[N_k] \leq 2m\Delta_k\frac{\mathbb{E}^\tau[N_k]}{\mathbb{E}^\tau[N_1]}. \tag{5}$$

By construction, $R^\zeta$ corresponds to the expected regret of our algorithm $\zeta$ for a number of rounds larger than $m$ in the 2-arm bandit problem with known average rewards. The proof of Lemma 7 is completed by establishing a lower bound on $R^\zeta$

**Regret lower bound of $R^\zeta$.** We prove the following lemma, which combined with (5) yields Lemma 7.

**Lemma 8.** *We have:* $R^\zeta \geq 2\Delta_k \phi(k, m, p)$.

**Proof of Lemma 8.** The proof is similar to that of Theorem 6 in [5]. The following lemma by [31, 5] is the essential ingredient of the proof:

**Lemma 9.** *Let $P_0$ and $P_1$ be two probability measures on a measurable space $(\Omega, \mathcal{F})$, with $P_0$ is absolutely continuous with respect to $P_1$. Then, for any $\mathcal{F}$-measurable function $\Psi : \Omega \to \{0, 1\}$, we have:*

$$\mathbb{P}_{P_0}(\Psi(\omega) = 1) + \mathbb{P}_{P_1}(\Psi(\omega) = 0) \geq \frac{1}{2} \exp(-\mathrm{KL}(P_0, P_1)).$$

Consider $a_t \in \{1, k\}$, defined as the $t$-th arm selection made by $\zeta$. This selection happens in the $m_t$-th round of an episode of $T$ rounds for the algorithm $\tau$. At the beginning of this episode, in the design of $\zeta$, items $i_1$ and $i_k$ have been selected, and in this $m_t$-th round, $\tau$ selects either $i_1$ or $i_k$. The decisions made under $\tau$ in this episode depend on the observations made in this episode only, and this remark holds for $\zeta$ as well. We define by $\mathcal{F}$ the $\sigma$-algebra generated by the observations made before the $m_t$-th round in the episode. To build $\mathcal{F}$, we assume that each time $i_1$ or $i_k$ is selected, then a sample of the reward of both items is observed.

With the above definitions, we have $\{a_t = 1\} \in \mathcal{F}$, and $\{a_t = k\} \in \mathcal{F}$. Next consider the the following two probability measures on $\mathcal{F}$: $P_0$ corresponds to the observations made in the original model (with the true item clusters), and $P_1$ to the observations made assuming that items $i_1$ and $i_k$ are swapped: the average reward of $i_1$ is $p_k$ and that of $i_k$ is $p_1$. $P_0$ and $P_1$ differ only when it comes to observations made in rounds where items $i_1$ and $i_k$ are selected. At round $m_t$, we know that we have had at most $t-1$ such rounds. We deduce that:

$$\mathrm{KL}(P_0, P_1) \leq (t-1)(\mathrm{kl}(p_1, p_k) + \mathrm{kl}(p_k, p_1)).$$

Applying Lemma 9, we get:

$$\mathbb{P}_{P_0}(a_t = k) + \mathbb{P}_{P_1}(a_t = 1) \geq \frac{1}{2} \exp\left(-(t-1)(\mathrm{kl}(p_1, p_k) + \mathrm{kl}(p_k, p_1))\right) = \frac{1}{2} e^{-(t-1)\gamma(p_1, p_k)}.$$
(6)

Observe that $\mathbb{P}_{P_0}(a_t = k) + \mathbb{P}_{P_1}(a_t = 1)$ is the expected instantaneous regret of $\zeta$ for its $t$-th arm selection. Hence, we have:

$$\begin{aligned}
R^\zeta &\geq \frac{\Delta_k}{4} \sum_{t=1}^{m} e^{-(t-1)\gamma(p_1, p_k)} \\
&= \frac{\Delta_k}{4} \frac{1 - e^{-m\gamma(p_1, p_k)}}{1 - e^{-\gamma(p_1, p_k)}} \\
&= 2\Delta_k \phi(k, m, p),
\end{aligned}$$

where the first inequality stems from the fact that $R^\zeta$ is the regret accumulated over more than $m$ rounds, and the second inequality is from (6). $\qquad\square$

# E Fundamental limits for Model B: Proof of Theorem 2

We apply the same strategy as in the case of clustered items. Let $\pi$ be an arbitrary algorithm.

Proof of $R^\pi(T) \geq R_{\mathrm{nr}}(T)$: Using the same reasoning as for Model A, the algorithm needs to sample at least $\max_u N_u(T)$, and a new item generated a satisficing regret equal to $\int_0^{\mu_{1-\varepsilon}}(\mu_{1-\varepsilon}-\mu)\zeta(\mu)d\mu$. Hence, we get:

$$R^\pi(T) \geq \mathbb{E}[\max_u N_u(T)]\int_0^{\mu_{1-\varepsilon}}(\mu_{1-\varepsilon}-\mu)\zeta(\mu)d\mu = R_{\mathrm{nr}}(T).$$

Proof of $R^\pi(T) \geq R_{\mathrm{i}}(T)$: For the term $R_{\mathrm{i}}(T)$, assume that $\zeta$ is known. With this knowledge, we denote $\tau$ an optimal algorithm. We denote by $\mathbb{E}^\tau[N_\mu]$ the expected number of rounds an item with success rate $\mu$ is selected under $\tau$. Formally, the algorithm $\tau$ induced the two following random counting measures on the interval $[0,1]$: (i) $\Xi$ counts the number of the items whose parameter is $\mu \in [0,1]$ seen by the algorithm $\tau$ ('seen' means selected at least once), (ii) $\Upsilon$ counts the number of times the algorithm $\tau$ selects items whose parameter is $\mu \in [0,1]$. Now the intensity measures [16] $\gamma$ and $\omega$ of $\Xi$ and $\Upsilon$ are absolutely continuous w.r.t. $\zeta$, and in addition, $\omega$ is absolutely continuous w.r.t. $\gamma$. Denote by $d_\gamma$ and $d_\omega$ the densities of $\gamma$ and $\omega$ w.r.t. $\zeta$. Then, $\mathbb{E}^\tau[N_\mu]$ is defined by $d_\omega(\mu)/d_\gamma(\mu)$. In the remark at the end of this proof, we make these definitions and the expression of the regret of $\tau$ explicit in the case where $\zeta$ is constant over intervals of $[0,1]$. Our proof could actually directly use a sequence of such discretizations, and then concludes by monotone limits.

Now the regret of an algorithm $\pi$ satisfies:

$$R^\pi_\varepsilon(T) \geq T\frac{\int_0^{\mu_{1-\varepsilon}}\mathbb{E}^\tau[N_\mu](\mu_{1-\varepsilon}-\mu)\zeta(\mu)d\mu}{\int_0^1\mathbb{E}^\tau[N_\mu]\zeta(\mu)d\mu}$$

$$\geq T\frac{\int_0^{\mu_{1-\varepsilon}}\mathbb{E}^\tau[N_\mu](\mu_{1-\varepsilon}-\mu)\zeta(\mu)d\mu}{\min\{1,(1+C)\varepsilon\}m+\int_0^{\mu_{1-\varepsilon}-\varepsilon}\mathbb{E}^\tau[N_\mu]\zeta(\mu)d\mu}$$

$$\geq \frac{T}{m}\frac{\frac{(1-\varepsilon)^2}{2C}\left(1-\frac{\varepsilon C}{1-\varepsilon}\right)^2}{\min\{1,(1+C)\varepsilon\}+1/m},$$

where we use the fact that $\varepsilon \geq \frac{1}{\sqrt{m}}$ and $1 \leq \mathbb{E}^\tau[N_\mu] \leq m$ for all $\mu$.

To complete the proof of the theorem, we just establish the following inquality:

$$T\frac{\int_0^{\mu_{1-\varepsilon}}\mathbb{E}^\tau[N_\mu](\mu_{1-\varepsilon}-\mu)\zeta(\mu)d\mu}{\min\{1,(1+C)\varepsilon\}m+\int_0^{\mu_{1-\varepsilon}-\varepsilon}\mathbb{E}^\tau[N_\mu]\zeta(\mu)d\mu} \geq T\frac{\frac{(1-\varepsilon)^2}{2C}\left(1-\frac{\varepsilon C}{1-\varepsilon}\right)^2}{\min\{1,(1+C)\varepsilon\}m+1}.$$

Let $\psi_i = \int_{\mu_{1-\varepsilon}-(i+1)\varepsilon}^{\mu_{1-\varepsilon}-i\varepsilon}\mathbb{E}^\tau[N_\mu]\zeta(\mu)d\mu$. Then,

$$\psi_i \in \left[\int_{\mu_{1-\varepsilon}-(i+1)\varepsilon}^{\mu_{1-\varepsilon}-i\varepsilon}\zeta(\mu)d\mu, \infty\right),$$

since $\mathbb{E}^\tau[N_\mu] \geq 1$ for all $\mu$. We have:

$$T\frac{\int_0^{\mu_{1-\varepsilon}}\mathbb{E}^\tau[N_\mu](\mu_{1-\varepsilon}-\mu)\zeta(\mu)d\mu}{\min\{1,(1+C)\varepsilon\}m+\int_0^{\mu_{1-\varepsilon}-\varepsilon}\mathbb{E}^\tau[N_\mu]\zeta(\mu)d\mu} \geq T\frac{\sum_{i=1}^{\lfloor\mu_{1-\varepsilon}/\varepsilon\rfloor}i\varepsilon\psi_i}{\min\{1,(1+C)\varepsilon\}m+\sum_{i=1}^{\lfloor\mu_{1-\varepsilon}/\varepsilon\rfloor}\psi_i}. \tag{7}$$

As the derivate of $\frac{a+bx}{c+dx}$ is $\frac{bc-da}{(c+dx)^2}$, $\frac{\sum_{i=1}^{\lfloor\mu_{1-\varepsilon}/\varepsilon\rfloor}i\varepsilon\psi_i}{\min\{1,(1+C)\varepsilon\}m+\sum_{i=1}^{\lfloor\mu_{1-\varepsilon}/\varepsilon\rfloor}\psi_i}$ is either an increasing function or a decreasing function of $\psi_i$ when all other $\psi_j$'s are fixed. Therefore, the r.h.s. of (7) can be optimized only when the $\psi_i$'s are at extreme points, either $\int_{\mu_{1-\varepsilon}-(i+1)\varepsilon}^{\mu_{1-\varepsilon}-i\varepsilon}\zeta(\mu)d\mu$ or $\infty$.

When $\psi_i = \int_{\mu_{1-\varepsilon}-(i+1)\varepsilon}^{\mu_{1-\varepsilon}-i\varepsilon} \zeta(\mu)d\mu$ for all $i$,

$$T\frac{\sum_{i=1}^{\lfloor \mu_{1-\varepsilon}/\varepsilon \rfloor} i\varepsilon\psi_i}{\min\{1,(1+C)\varepsilon\}m + \sum_{i=1}^{\lfloor \mu_{1-\varepsilon}/\varepsilon \rfloor}\psi_i} \geq T\frac{\sum_{i=1}^{\lfloor \mu_{1-\varepsilon}/\varepsilon \rfloor} i\varepsilon \int_{\mu_{1-\varepsilon}-(i+1)\varepsilon}^{\mu_{1-\varepsilon}-i\varepsilon}\zeta(\mu)d\mu}{\min\{1,(1+C)\varepsilon\}m + 1}$$

$$\geq T\frac{\frac{(1-\varepsilon)^2}{2C}\left(1-\frac{\varepsilon C}{1-\varepsilon}\right)^2}{\min\{1,(1+C)\varepsilon\}m + 1}.$$

When $\psi_i = \infty$ for some $i \geq 1$, we have

$$T\frac{\sum_{i=1}^{\lfloor \mu_{1-\varepsilon}/\varepsilon \rfloor} i\varepsilon\psi_i}{\min\{1,(1+C)\varepsilon\}m + \sum_{i=1}^{\lfloor \mu_{1-\varepsilon}/\varepsilon \rfloor}\psi_i} \geq \varepsilon T.$$

Thus, we have

$$T\frac{\int_0^{\mu_{1-\varepsilon}} \mathbb{E}^\tau[N_\mu](\mu_{1-\varepsilon}-\mu)\zeta(\mu)d\mu}{\min\{1,(1+C)\varepsilon\}m + \int_0^{\mu_{1-\varepsilon}-\varepsilon} \mathbb{E}^\tau[N_\mu]\zeta(\mu)d\mu} \geq T\min\left\{\frac{\frac{(1-\varepsilon)^2}{2C}\left(1-\frac{\varepsilon C}{1-\varepsilon}\right)^2}{\min\{1,(1+C)\varepsilon\}m + 1}, \varepsilon\right\}$$

$$= T\frac{\frac{(1-\varepsilon)^2}{2C}\left(1-\frac{\varepsilon C}{1-\varepsilon}\right)^2}{\min\{1,(1+C)\varepsilon\}m + 1},$$

where the last equation stems from the assumption $m \geq \frac{c}{\varepsilon^2}$. This concludes the proof. $\square$

**Remark.** Assume that there exists a finite set of non-overlapping intervals of $[0,1]$ and covering $[0,1]$ such that the density of $\zeta$ is constant over each of these intervals. We denote by $\zeta_i$ the probability that when a new item is selected, its parameter lies in the $i$-th interval. Further assume that the satisficing regret of an item with parameter in the $i$-th interval does depend on the parameter, and is equal to $\Delta_i$. Under the algorithm $\tau$, let $\mathbb{E}^\tau[P_i]$ denote the expected number of items seen (selected at least once) by $\tau$ and whose parameter is in the $i$-th interval, and let $\mathbb{E}^\tau[N_i]$ the expected number of times $\tau$ selects an item with parameter in the $i$-th interval. Then, the equivalent of $\mathbb{E}^\tau[N_\mu]$ is $\eta_i$, the expected number of times an item with parameter in the $i$-th interval is selected. $\eta_i$ is defined as $\eta_i = \mathbb{E}^\tau[N_i]/\mathbb{E}^\tau[P_i]$. Observe that since the item parameters of newly selected items are i.i.d. with distribution $\zeta_i$, $\mathbb{E}^\tau[P_i]$ is proportional to $\zeta_i$. This is just a consequence of the general Wald lemma: indeed, if $X_{ki}$ is the binary r.v. indicating whether the $k$-th item seen by the algorithm has a parameter in the $i$-th interval, and if $\kappa$ denotes the random number of items seen by the algorithm within the time horizon $T$, then Wald's equation holds if $\mathbb{E}^\tau[X_{ki}1_{\kappa \geq k}] = \mathbb{E}[X_{ki}]\mathbb{P}^\tau[\kappa \geq k]$. This is true in our case since the event $\{\kappa \geq k\}$ corresponds to the fact that $\tau$ decides to sample the $k$-th item, and this decision is solely based on observations made on the $(k-1)$ first items. Finally, the regret of $\tau$ is:

$$R^\tau(T) = \sum_i \Delta_i \mathbb{E}^\tau[N_i] = T\frac{\sum_i \Delta_i \eta_i \zeta_i}{\sum_i \eta_i \zeta_i}.$$

In the above formula we used the fact that $T = \sum_i \mathbb{E}^\tau[N_i]$. Note that we obtained a discrete version of $T\frac{\int_0^{\mu_{1-\varepsilon}} \mathbb{E}^\tau[N_\mu](\mu_{1-\varepsilon}-\mu)\zeta(\mu)d\mu}{\int_0^1 \mathbb{E}^\tau[N_\mu]\zeta(\mu)d\mu}$.

# F  Fundamental limits for Model C: Proof of Theorem 3

We start this section by illustrating the various terms involved in the regret lower bound in Theorem 3. We then prove the theorem.

## F.1  Examples

Let $\pi$ be a uniformly good algorithm. Then under the conditions of Theorem 3, we have: $R^\pi(T) \geq \max\{R_{\mathrm{nr}}(T), R_{\mathrm{ic}}(T), R_{\mathrm{uc}}(T)\}$ and $R^\pi(T) \gtrsim R'_{\mathrm{uc}}(T)$. We exemplify the scalings of these terms below, with a particular emphasis on those due to the need of learning user clusters, $R_{\mathrm{uc}}(T)$ and $R'_{\mathrm{uc}}(T)$.

**Case 1.** We consider the case $L = 2$ and $K = 2$ with a following parameter set in Table 6.

|         | $k = 1$ | $k = 2$ |
|---------|---------|---------|
| $\ell = 1$ | 0.8     | 0.6     |
| $\ell = 2$ | 0.8     | 0.9     |

Table 6: Values of $(p_{k\ell})$.

For this parameter, $\mathcal{L}_\perp = \{(1,2)\}$ and $\mathcal{L}^\perp(1) = \{2\}$. We have: $R_{\mathrm{nr}} = \Omega(\bar{n})$, $R_{\mathrm{ic}} = \Omega(\frac{T}{m})$ and $R_{\mathrm{uc}} = \Omega(m)$. Furthermore, when $T = \omega(m)$,

$$R^\pi(T) \gtrsim \frac{\beta_1(0.8 - 0.6)}{\mathrm{kl}(0.6, 0.9)} m \log(\frac{T}{m}) = \Omega(m \log(\frac{T}{m})).$$

**Case 2.** We consider the case $L = 2$ and $K = 2$ with a following parameter set in Table 7.

|         | $k = 1$ | $k = 2$ |
|---------|---------|---------|
| $\ell = 1$ | 0.8     | 0.6     |
| $\ell = 2$ | 0.8     | 0.7     |

Table 7: Values of $(p_{k\ell})$.

For this parameter, $\mathcal{L}_\perp = \emptyset$ and $\forall \ell \in [L]$, $\mathcal{L}^\perp(\ell) = \emptyset$. We have: $R_{\mathrm{nr}} = \Omega(\bar{n})$, $R_{\mathrm{ic}} = \Omega(\frac{T}{m})$, $R_{\mathrm{uc}} = 0$ and $c(\beta, p) = 0$.

**Case 3.** We consider the case $L = 2$ and $K = 2$ with a following parameter set in Table 8.

|         | $k = 1$ | $k = 2$ |
|---------|---------|---------|
| $\ell = 1$ | 0.8     | 0.9     |
| $\ell = 2$ | 0.8     | 0.85    |

Table 8: Values of $(p_{k\ell})$.

In this case, $\mathcal{L}_\perp = \emptyset$ and $\forall \ell \in [L]$, $\mathcal{L}^\perp(\ell) = \emptyset$. We have: $R_{\mathrm{nr}} = \Omega(\bar{n})$, $R_{\mathrm{ic}} = \Omega(\frac{T}{m})$, $R_{\mathrm{uc}} = 0$ and $c(\beta, p) = 0$.

**Case 4.** We consider the case $L = 2$ and $K = 3$ with a following parameter set in Table 9.

|         | $k = 1$ | $k = 2$ | $k = 3$ |
|---------|---------|---------|---------|
| $\ell = 1$ | 0.9     | 0.8     | 0.7     |
| $\ell = 2$ | 0.7     | 0.8     | 0.9     |

Table 9: Values of $(p_{k\ell})$.

In this case, $\mathcal{L}_\perp = \{(1,2)\}$ and $\forall \ell \in [L]$, $\mathcal{L}^\perp(\ell) = \emptyset$. We have: $R_{\mathrm{nr}} = \Omega(\bar{n})$, $R_{\mathrm{ic}} = \Omega(\frac{T}{m})$, $R_{\mathrm{uc}} = \Omega(m)$ and $c(\beta, p) = 0$.

## F.2 Proof

Proof of $R^\pi(T) \geq \max\{R_{\mathrm{nr}}(T), R_{\mathrm{ic}}(T)\}$. The proof is the same as in that of Theorem 1.

Proof of $R^\pi(T) \geq R_{\mathrm{uc}}(T)$. We first give a simple proof that learning user clusters induces a regret scaling as $m$, i.e., that $R^\pi(T) = \Omega(m)$. When a user first arrives, we do not know her cluster, and hence we have to recommend an item from a cluster picked randomly. This selection induces an average regret at least equal to $\min_\ell\{\beta_\ell\}\Delta$ when $\mathcal{L}_\perp \neq \emptyset$. Since the number of users that arrive at least once is in expectation larger than $m(1 - (1 - \frac{1}{m})^T) \geq m(1 - e^{-T/m})$, we get that: $R^\pi(T) \geq \min_\ell\{\beta_\ell\}\Delta m(1 - e^{-T/m})$.

To get the right constant in the regret lower bound, we need to develop a more involved argument. Assume that the item clusters $\mathcal{I}_k$'s and the success rates $p$ are known. With this knowledge, we denote by $\tau$ an optimal algorithm. We derive a regret lower bound for $\tau$. Define $N_k^u := \sum_{t=1}^T \mathbb{1}_{\{i_t^\pi \in \mathcal{I}_k, u_t = u\}}$ as the number of times user $u$ is presented items in cluster $\mathcal{I}_k$ (under $\pi$). Fix user clusters $\mathcal{U}_1, \ldots, \mathcal{U}_L$. Similar to the proof of Lemma 7, we will prove that:

**Lemma 10.** *We have for all $T \geq 2m$, for any $\ell \in [L]$, for any $k \in [K]$ such that $\exists r \in \mathcal{R}_\ell$ such that $k = k_r^*$:* $\frac{\psi(\ell, k, T, m, p)}{(T/m)} \leq \frac{\mathbb{E}^\tau[N_k^u | u \in \mathcal{U}_\ell]}{\mathbb{E}^\tau[N_{k_\ell^*}^u | u \in \mathcal{U}_\ell]} \leq 1$, *where* $\psi(\ell, k, T, m, p) = \frac{1 - e^{-\frac{T}{m}\gamma(p_{k_\ell^*}, p_{k\ell})}}{8\left(1 - e^{-\gamma(p_{k_\ell^*}, p_{k\ell})}\right)}$.

For fixed user clusters, the expected conditional regret $R^\tau(T|\mathcal{U}_1, \ldots, \mathcal{U}_L)$ is: $R^\tau(T|\mathcal{U}_1, \ldots, \mathcal{U}_L) = \sum_{\ell \in [L]} |\mathcal{U}_\ell| \sum_{k \in \mathcal{R}_\ell} \mathbb{E}^\tau[N_k^u | u \in \mathcal{U}_\ell]\Delta_{k\ell}$. Therefore, we have:

$$R^\tau(T|\mathcal{U}_1, \ldots, \mathcal{U}_L) = \sum_{\ell \in [L]} |\mathcal{U}_\ell| \frac{\frac{T}{m} \sum_{k \in \mathcal{R}_\ell} \mathbb{E}^\tau[N_k^u | u \in \mathcal{U}_\ell]\Delta_{k\ell}}{\sum_{k \in [K]} \mathbb{E}^\tau[N_k^u | u \in \mathcal{U}_\ell]}$$

$$\overset{(a)}{\geq} \sum_{\ell \in [L]} |\mathcal{U}_\ell| \frac{\frac{T}{m} \sum_{k \in \mathcal{R}_\ell} \mathbb{E}^\tau[N_{k_\ell^*}^u | u \in \mathcal{U}_\ell]\Delta_{k\ell}\psi(\ell, k, T, m, p)}{\sum_{k \in [K]} \mathbb{E}^\tau[N_{k_\ell^*}^u | u \in \mathcal{U}_\ell]\frac{T}{m}}$$

$$= \sum_{\ell \in [L]} |\mathcal{U}_\ell| \frac{\sum_{k \in \mathcal{R}_\ell} \Delta_{k\ell}\psi(\ell, k, T, m, p)}{K},$$

where for $(a)$ we used Lemma 10 ($\psi(\ell, k, T, m, p)\mathbb{E}^\tau[N_{k_\ell^*}^u]/(T/m) \leq \mathbb{E}^\tau[N_k^u | u \in \mathcal{U}_\ell]$ in the numerator, and $\mathbb{E}^\tau[N_k^u | u \in \mathcal{U}_\ell] \leq \mathbb{E}^\tau[N_{k_\ell^*}^u | u \in \mathcal{U}_\ell]$ in the denominator). Taking the expectation over $\mathcal{U}_1, \ldots, \mathcal{U}_L$, we have (since $\mathbb{E}[\mathcal{U}_\ell] = \beta_\ell m$):

$$R^\tau(T) \geq m \sum_{\ell \in [L]} \beta_\ell \frac{\sum_{k \in \mathcal{R}_\ell} \Delta_{k\ell}\psi(\ell, k, T, m, p)}{K}.$$

**Proof of Lemma 10.** Consider a 2-armed bandit problem with Bernoulli arms $k_\ell^*$ and $k$ of means $p_{k_\ell^*}$ and $p_k$. The means are known but the arm with the highest mean is unknown. For this bandit problem, we build an algorithm $\zeta$ based on the decisions by $\tau$.

**A valid algorithm $\zeta$ based on $\tau$.**

1. Pick a user $u$ uniformly at random in the cluster $\mathcal{U}_\ell$. We run $\tau$ for $T$ rounds. When user $u$ comes to the system and $\tau$ selects the item in $\mathcal{I}_{k_\ell^*}$ (resp. $\mathcal{I}_k$), then $\zeta$ also selects the arm $k_\ell^*$ (resp. $k$). We call this procedure as an episode.

2. We repeat the above Step 1, to determine the arm selections made by $\zeta$. At the beginning of the successive episodes of $T$ rounds, the user $u$ are again chosen uniformly at random in the clusters $\mathcal{I}_\ell$, independently of the choices made in earlier episodes.

$\zeta$ is a valid algorithm as the decision by $\tau$ is based on past observations. We stop $\zeta$ after $\kappa$ episodes of $T$ rounds, where $\kappa$ is the first episode where $\zeta$ has made more than $\frac{T}{m}$ selections. By Wald's first lemma, the expected regret accumulated by $\zeta$ before we stop playing is:

$$R^\zeta = \Delta_{k\ell}\mathbb{E}^\tau[\kappa]\mathbb{E}^\tau[N_k^u | u \in \mathcal{U}_\ell].$$

Since $\mathbb{E}^\tau[N_k^u \mid u \in \mathcal{U}_\ell] + \mathbb{E}^\tau[N_{k_\ell^*}^u \mid u \in \mathcal{U}_\ell] \le \frac{T}{m}$, We have:

$$\mathbb{E}[\kappa]\mathbb{E}^\tau[N_{k_\ell^*}^u \mid u \in \mathcal{U}_\ell] \le \underbrace{\frac{T}{m}}_{\text{Stopping criteria of } \zeta} + \underbrace{\mathbb{E}[\sum_k N_k^u \mid u \in \mathcal{U}_\ell]}_{\text{expected number of drawing the user } u \text{ in a single episode}}$$

$$= \frac{2T}{m}$$

We will also prove a lower bound on $R^\zeta$:

**Lemma 11.** *We have for all $T \ge 2m$: $R^\zeta \ge 2\Delta_{k\ell}\psi(\ell, k, T, m, p)$.*

Combining this lemma with Lemma 10 concludes the proof of $R^\pi(T) \ge R_{\text{uc}}(T)$.

**Proof of Lemma 11.** When the algorithm decides which arm to choose at time $t$, we assume that the algorithm has access to the rewards of both arms up to time $t - 1$. This is a simpler problem than the original 2-armed bandit problem. Hence the regret in the original problem is higher than that in the simpler problem. Let $\theta$ denote the distribution of the rewards of both arms, when the average reward of the first arm is $p_{k_\ell^*}$ and that of the second arm is $p_k$. Let $\theta'$ denote the distribution of the rewards of both arms, when arms are swapped: the average reward of the first arm is $p_k$ and that of the second arm is $p_{k_\ell^*}$. Let $\theta^{\otimes(t-1)}$ be a product measure for the reward observations up to time $t - 1$ under the measure $\theta$. Let $k_t$ be the arm selected at time $t$. From Lemma 9, we have, for each $t \ge 2$,

$$\mathbb{P}_\theta(k_t = k) + \mathbb{P}_{\theta'}(k_t = k_\ell^*) \ge \frac{1}{2}\exp(-\text{KL}(\theta^{\otimes(t-1)}, \theta'^{\otimes(t-1)}))$$

$$= \frac{1}{2}\exp\left(-(t-1)(\text{kl}(p_{k_\ell^*\ell}, p_{k\ell}) + \text{kl}(p_{k\ell}, p_{k_\ell^*\ell}))\right) = \frac{1}{2}e^{-(t-1)\gamma(p_{k_\ell^*\ell}, p_{k\ell})}.$$

(8)

Note that (8) also holds for $t = 1$ by the symmetry. We have,

$$R^\zeta \ge \frac{\Delta_{k\ell}}{2}\sum_{t=1}^{T/m}\left(\mathbb{P}_\theta(k_t = k) + \mathbb{P}_{\theta'}(k_t = k_\ell^*)\right)$$

$$\ge \frac{\Delta_{k\ell}}{4}\sum_{t=1}^{T/m}e^{-(t-1)\gamma(p_{k_\ell^*\ell}, p_{k\ell})}$$

$$= \frac{\Delta_{k\ell}}{4}\frac{1 - e^{-\frac{T}{m}\gamma(p_{k_\ell^*\ell}, p_{k\ell})}}{1 - e^{-\gamma(p_{k_\ell^*\ell}, p_{k\ell})}}$$

$$= 2\Delta_{k\ell}\psi(\ell, k, T, m, p),$$

where the first inequality stems from the fact that $R^\zeta$ is the regret accumulated over more than $\frac{T}{m}$ rounds, and the second inequality is from (8). □

Proof of $R^\pi(T) \gtrsim c(\beta, p)m\log(T/m)$. We define:

$$\Theta := \{p \in [0,1]^{K \times L} : \forall \ell \in [L], \exists \Gamma \text{ (permutation of } [K]) \text{ s.t. } p_{\Gamma(1)\ell} > p_{\Gamma(2)\ell} \ge \ldots \ge p_{\Gamma(K)\ell}\}$$

as a set of all possible problems. We denote $k_\ell^* := \arg\max_k p_{k\ell}$ as the index of the best item cluster for users in the cluster $\mathcal{U}_\ell$ and $\Delta_{k\ell} := p_{k_\ell^*\ell} - p_{k\ell} \ \forall k \in [K], \forall \ell \in [L]$. Consider an arbitrary algorithm $\pi$. We define the regret of a single user $u$ as:

$$R_u^\pi(T) := \frac{T}{m}\sum_\ell \beta_\ell p_{k_\ell^*\ell} - \sum_{t=1}^T \mathbb{E}\left[\sum_{k,\ell}\mathbb{1}_{\{u\in\mathcal{U}_\ell, u_t=u, i_t^\pi\in\mathcal{I}_k\}}p_{k\ell}\right]$$

$$= \sum_\ell \beta_\ell \sum_{k\neq k_\ell^*}\Delta_{k\ell}\mathbb{E}[N_k^u|u\in\mathcal{U}_\ell],$$

where $N_k^u := \sum_{t=1}^T \mathbb{1}_{\{i_t^\pi\in\mathcal{I}_k, u_t=u\}}$ is the number of times user $u$ is presented an item of cluster $\mathcal{I}_k$ (under $\pi$). Remember that $u$ is random and belongs to $\mathcal{U}_\ell$ with probability $\beta_\ell$. We further define the

conditional regret of a single user $u \in \mathcal{U}_\ell$ given $N_u(T) = N$ and $u \in \mathcal{U}_\ell$ as:

$$R_u^\pi(T)_{N,\ell} := Np_{k_\ell^* \ell} - \sum_{t=1}^T \mathbb{E}\left[\sum_k \mathbb{1}_{\{u_t=u, i_t^\pi \in \mathcal{I}_k\}} p_{k\ell} \,\middle|\, N_u(T) = N, u \in \mathcal{U}_\ell\right]$$

$$= \sum_{k \neq k_\ell^*} \Delta_{k\ell} \mathbb{E}[N_k^u \mid u \in \mathcal{U}_\ell, N_u(T) = N].$$

Note that we have $R^\pi(T) = \sum_{u \in \mathcal{U}} R_u^\pi(T) = mR_u^\pi(T)$ and $R_u^\pi(T) = \sum_{N \geq 1}^T \mathbb{P}(N_u(T) = N) \sum_{\ell \in [L]} \beta_\ell R_u^\pi(T)_{N,\ell}$.

Assume that $\pi$ is uniformly good. This means that if for all problem $p \in \Theta$, as $N \to \infty$, the conditional regret $R_u^\pi(T)_{N,\ell}$ satisfies:

$$\forall \ell \in [L], \ \forall\, 0 < \alpha < 1, \quad R_u^\pi(T)_{N,\ell} = o\left((N)^\alpha\right). \tag{9}$$

The existence of uniformly good algorithms is guaranteed because applying the classical algorithms (e.g, UCB1) to each user satisfy indeed is uniformly good (this is proved for the ECB algorithm using Theorem 9 presented in Appendix J).

We state our claim in the following theorem, providing a lower bound of the regret of a single user:

**Theorem 8.** *For any uniformly good algorithm $\pi \in \Pi$, for any $p \in \Theta$, when $T = \omega(m)$, we have: for any $u \in \mathcal{U}$,*

$$\liminf_{T \to \infty} \frac{R_u^\pi(T)}{\log(T/m)} \geq c(\beta, p),$$

*where $c(\beta, p)$ is the value of the following optimization problem:*

$$\inf_{n=(n_{k\ell}) \geq 0} \sum_{\ell \in [L]} \beta_\ell \sum_{k \neq k_\ell^*} \Delta_{k\ell} n_{k\ell} \tag{10}$$

$$s.t. \quad \forall \ell \in [L], \quad \forall \ell' \in \mathcal{L}^\perp(\ell), \quad \sum_{k \neq k_\ell^*} \mathrm{kl}(p_{k\ell}, p_{k\ell'}) n_{k\ell} \geq 1.$$

In the above theorem, we can interpret $n_{k\ell}$ as

$$n_{k\ell} \overset{T \to \infty}{\sim} \frac{\mathbb{E}[N_k^u \mid u \in \mathcal{U}_\ell, N_u(T) = T/m]}{\log(T/m)}.$$

**Proof of Theorem 8: Case $K = 2$, $L = 2$.** To illustrate the idea behind the proof, we address the simple case with two item and user clusters. We define the values of $(p_{k\ell})$ as in Table F.2, where $\mu_1 > \mu_2$ and $\mu_1 < \mu_2'$, so that $\mathcal{L}^\perp(1) = \{2\}$.

|          | $k = 1$ | $k = 2$  |
|----------|---------|----------|
| $\ell = 1$ | $\mu_1$ | $\mu_2$  |
| $\ell = 2$ | $\mu_1$ | $\mu_2'$ |

Table 10: The values of $(p_{k\ell})$. $\mu_1 > \mu_2$ and $\mu_1 < \mu_2'$.

The proof is in two steps. In the first step we derive a lower bound of the conditional regret, and in the second step, we de-condition using properties of the user arrival process.

*Step 1.* In this step, we condition on $N_u(T) = N$ and $u \in \mathcal{U}_1$. All expectations and probabilities are conditioned with respect by these events. We apply a classical change-of-measure argument. Let $p$ denote the original model. We build a perturbed model $p'$ obtained from $p$ by just swapping the ides of the user clusters. Let $\mathbb{P}_p$ and $\mathbb{E}_p$ (resp. $\mathbb{P}_{p'}$ and $\mathbb{E}_{p'}$) be the probability measure and the expectation under $p$ (resp. $p'$), respectively. We compute the log-likelihood ratio of the observations for user $u$ generated under $p$ and $p'$ as:

$$L_T := \sum_{t=1}^T \mathbb{1}_{\{u_t=u\}} \left(\mathbb{1}_{\{u \in \mathcal{U}_1, i_t^\pi \in \mathcal{I}_2\}} \left(\mathbb{1}_{\{X_{i_t^\pi u}=+1\}} \log \frac{\mu_2}{\mu_2'} + \mathbb{1}_{\{X_{i_t^\pi u}=0\}} \log \frac{1-\mu_2}{1-\mu_2'}\right)\right.$$

$$\left. + \mathbb{1}_{\{u \in \mathcal{U}_2, i_t^\pi \in \mathcal{I}_2\}} \left(\mathbb{1}_{\{X_{i_t^\pi u}=+1\}} \log \frac{\mu_2'}{\mu_2} + \mathbb{1}_{\{X_{i_t^\pi u}=0\}} \log \frac{1-\mu_2'}{1-\mu_2}\right)\right).$$

For any measurable random variable $Z \in [0, 1]$, we have:

$$\mathbb{E}_p[L_T] = \mathbb{E}_p[N_2^u]\mathrm{kl}(\mu_2, \mu_2') \overset{(a)}{\geq} \mathrm{kl}(\mathbb{E}_p[Z], \mathbb{E}_{p'}[Z]),$$

where $(a)$ stems from the data-processing inequality (cf. [8]). Taking $Z = N_2^u/N$, we have:

$$\mathrm{kl}(\mathbb{E}_p[Z], \mathbb{E}_{p'}[Z]) = \mathrm{kl}\left(\frac{\mathbb{E}_p[N_2^u]}{N}, \frac{\mathbb{E}_{p'}[N_2^u]}{N}\right)$$

$$\geq \left(1 - \frac{\mathbb{E}_p[N_2^u]}{N}\right) \log \left(\frac{N}{N - \mathbb{E}_{p'}[N_2^u]}\right) - \log 2,$$

where for the last inequality, we used that for all $(x, y) \in [0, 1]^2$,

$$\mathrm{kl}(x, y) \geq (1 - x) \log \frac{1}{1 - y} - \log 2.$$

As the algorithm $\pi$ is uniformly good, $N - \mathbb{E}_{p'}(N_2^u) = o(N^\alpha)$ for all $\alpha \in (0, 1)$ as $N \to \infty$. Therefore, for all $\alpha \in (0, 1)$,

$$\liminf_{N \to \infty} \frac{1}{\log N} \log \frac{N}{N - \mathbb{E}_{p'}(N_2^u)} \geq \liminf_{N \to \infty} \frac{1}{\log N} \log \frac{N}{N^\alpha}$$

$$= 1 - \alpha.$$

Furthermore, $\lim_{N \to \infty} \frac{\mathbb{E}_p[N_2^u]}{N} = 0$ as $\pi$ is uniformly good. Therefore, we have:

$$\liminf_{N \to \infty} \frac{\mathbb{E}_p[N_2^u]}{\log N} \geq \frac{1}{\mathrm{kl}(\mu_2, \mu_2')}. \tag{11}$$

*Step 2. De-conditioning.* In view of Lemma 5, we have:

$$\mathbb{P}\left(N_u(T) \leq \frac{T}{2m}\right) \leq \exp\left(-\frac{T}{2m}(1 - \log(2))\right).$$

From the above inequality and (11), we deduce:

$$\liminf_{T \to \infty} \frac{R_u^\pi(T)}{\log(T/m)} \geq \liminf_{T \to \infty} \frac{\sum_{N=1}^T \mathbb{P}(N_u(T) = N)\beta_1 R_u^\pi(T)_{N,1}}{\log(T/m)}$$

$$\geq \liminf_{T \to \infty} \frac{\sum_{N=\frac{T}{2m}}^T \mathbb{P}(N_u(T) = N)\beta_1 R_u^\pi(T)_{\frac{T}{2m},1}}{\log(T/m)}$$

$$\geq \liminf_{T \to \infty} \beta_1(\mu_1 - \mu_2) \sum_{N=\frac{T}{2m}}^T \mathbb{P}(N_u(T) = N)\frac{\mathbb{E}_p[N_2^u | u \in \mathcal{U}_1, N_u(T) = T/(2m)]}{\log(T/m)}$$

$$\overset{(a)}{\geq} \liminf_{T \to \infty} \beta_1(\mu_1 - \mu_2)\left(1 - \exp\left(-\frac{T}{2m}(1 - \log 2)\right)\right)\frac{\log(T/(2m))}{\mathrm{kl}(\mu_2, \mu_2')\log(T/m)}$$

$$\overset{(b)}{=} \frac{\beta_1(\mu_1 - \mu_2)}{\mathrm{kl}(\mu_2, \mu_2')},$$

where for $(a)$ we used Lemma 5 with (11) and for $(b)$ we used $T = \omega(m)$. This concludes the proof of the case $K = 2$ and $L = 2$.

$\square$

**Proof of Theorem 8: General case.** We consider a simpler problem: the algorithm knows the values of $(p_{k\ell})$ and $\beta_\ell$. Take $(\ell, \ell') \in [L]^2$ such that $\ell' \in \mathcal{L}^\perp(\ell)$. As in the case of two user and item clusters, we will prove that (this is done at the end of this proof):

$$\liminf_{N \to \infty} \frac{\sum_{k \neq k_\ell^*} \mathbb{E}_p[N_k^u | u \in \mathcal{U}_\ell, N_u(T) = N]\mathrm{kl}(p_{k\ell}, p_{k\ell'})}{\log N} \geq 1. \tag{12}$$

This inequality holds for any possible $(\ell, \ell')$ such that $\ell' \in \mathcal{L}^\perp(\ell)$. Therefore, for all $\ell \in [L]$, for all $\ell' \in \mathcal{L}^\perp(\ell)$

$$\liminf_{N \to \infty} \frac{\sum_{k \neq k_\ell^*} \mathbb{E}_p[N_k^u \mid u \in \mathcal{U}_\ell, N_u(T) = N] \mathrm{kl}(p_{k\ell}, p_{k\ell'})}{\log N} \geq 1.$$

Then, we have:

$$\liminf_{T \to \infty} \frac{R_u^\pi(T)}{\log(T/m)}$$

$$\geq \liminf_{T \to \infty} \sum_{\ell \in [L]} \beta_\ell \sum_{N=1}^{T} \mathbb{P}(N_u(T) = N) \frac{R_u^\pi(T)_{N,\ell}}{\log(T/m)}$$

$$\geq \liminf_{T \to \infty} \sum_{\ell \in [L]} \beta_\ell \sum_{N=\frac{T}{2m}}^{T} \mathbb{P}(N_u(T) = N) \frac{\sum_{k \neq k_\ell^*} \Delta_{k\ell} \mathbb{E}[N_k^u \mid u \in \mathcal{U}_\ell, N_u(T) = T/(2m)]}{\log(T/m)}$$

$$\overset{(a)}{\geq} \liminf_{T \to \infty} \left(1 - \exp\left(-\frac{T}{2m}(1 - \log 2)\right)\right) \frac{\sum_{\ell \in [L]} \beta_\ell \sum_{k \neq k_\ell^*} \Delta_{k\ell} \mathbb{E}[N_k^u \mid u \in \mathcal{U}_\ell, N_u(T) = T/(2m)]}{\log(T/m)}$$

$$= \liminf_{T \to \infty} \left(1 - \exp\left(-\frac{T}{2m}(1 - \log 2)\right)\right) \frac{\sum_{\ell \in [L]} \beta_\ell \sum_{k \neq k_\ell^*} \Delta_{k\ell} \mathbb{E}[N_k^u \mid u \in \mathcal{U}_\ell, N_u(T) = T/(2m)]}{\log(T/(2m)) + \log 2}$$

$$\overset{(b)}{=} \liminf_{T \to \infty} \frac{\sum_{\ell \in [L]} \beta_\ell \sum_{k \neq k_\ell^*} \Delta_{k\ell} \mathbb{E}[N_k^u \mid u \in \mathcal{U}_\ell, N_u(T) = T/(2m)]}{\log(T/(2m))},$$

where $(a)$ is from Lemma 5 and $(b)$ is from $T = \omega(m)$. Thus, we have:

$$\liminf_{T \to \infty} \frac{R_u^\pi(T)}{\log(T/m)} \geq c(\beta, p),$$

where $c(\beta, p)$ is the value of the following optimization problem:

$$\inf_{n=(n_{k\ell}) \geq 0} \sum_{\ell \in [L]} \beta_\ell \sum_{k \neq k_\ell^*} \Delta_{k\ell} n_{k\ell}$$

$$\text{s.t.} \quad \forall \ell \in [L], \quad \forall \ell' \in \mathcal{L}^\perp(\ell), \quad \sum_{k \neq k_\ell^*} \mathrm{kl}(p_{k\ell}, p_{k\ell'}) n_{k\ell} \geq 1.$$

Proof of the inequality (12). Again, we use a change-of-measure argument. Let $p$ and $p'$ be a original model and a model with the indices of user clusters $(\ell, \ell')$ are swapped from the original model, respectively. Let $\mathbb{P}_p$ and $\mathbb{E}_p$ (resp. $\mathbb{P}_{p'}$ and $\mathbb{E}_{p'}$) be the probability measure and the expectation under $p$ (resp. $p'$), respectively. We define our log-likelihood ratio as:

$$L_T := \sum_{t=1}^{T} \sum_{k \neq k_\ell^*} \mathbb{1}_{\{u_t = u\}} \left( \mathbb{1}_{\{u \in \mathcal{U}_\ell, i_t^\pi \in \mathcal{I}_k\}} \left( \mathbb{1}_{\{X_{i_t^\pi u} = +1\}} \log \frac{p_{k\ell}}{p_{k\ell'}} + \mathbb{1}_{\{X_{i_t^\pi u} = 0\}} \log \frac{1 - p_{k\ell}}{1 - p_{k\ell'}} \right) \right.$$

$$\left. + \mathbb{1}_{\{u \in \mathcal{U}_{\ell'}, i_t^\pi \in \mathcal{I}_k\}} \left( \mathbb{1}_{\{X_{i_t^\pi u} = +1\}} \log \frac{p_{k\ell'}}{p_{k\ell}} + \mathbb{1}_{\{X_{i_t^\pi u} = 0\}} \log \frac{1 - p_{k\ell'}}{1 - p_{k\ell}} \right) \right).$$

Taking the conditional expectation $\mathbb{E}_p[\cdot \mid u \in \mathcal{U}_\ell, N_u(T) = N]$, we have:

$$\mathbb{E}_p[L_T \mid u \in \mathcal{U}_\ell, N_u(T) = N] = \sum_{k \neq k_\ell^*} \mathbb{E}_p[N_k^u \mid u \in \mathcal{U}_\ell, N_u(T) = N] \mathrm{kl}(p_{k\ell}, p_{k\ell'})$$

$$\overset{(a)}{\geq} \mathrm{kl}\left(\frac{\mathbb{E}_p[N_{k_{\ell'}^*}^u]}{N}, \frac{\mathbb{E}_{p'}[N_{k_{\ell'}^*}^u]}{N}\right)$$

$$\overset{(b)}{\geq} \left(1 - \frac{\mathbb{E}_p[N_{k_{\ell'}^*}^u]}{N}\right) \log\left(\frac{N}{N - \mathbb{E}_{p'}[N_{k_{\ell'}^*}^u]}\right) - \log 2,$$

where for $(a)$, we used the data processing inequality by [8] and for $(b)$, we used that for all $(x, y) \in [0, 1]^2$,

$$\mathrm{kl}(x, y) \geq (1 - x) \log \frac{1}{1 - y} - \log 2.$$

As the algorithm $\pi$ is uniformly good, $\mathbb{E}_p[N^u_{k^*_{\ell'}}] = o(N^\alpha)$ and $N - \mathbb{E}_{p'}(N^u_{k^*_{\ell'}}) = o(N^\alpha)$ for all $\alpha \in (0, 1)$ as $N \to \infty$. Therefore, for all $\alpha \in (0, 1)$, we have:

$$\liminf_{N \to \infty} \frac{1}{\log N} \log \frac{N}{N - \mathbb{E}_{p'}[N^u_{k^*_{\ell'}}]} \geq \liminf_{N \to \infty} \frac{1}{\log N} \log \frac{N}{N^\alpha}$$

$$= 1 - \alpha$$

and

$$\liminf_{N \to \infty} \frac{\mathbb{E}_p[N^u_{k^*_{\ell'}}]}{N} = 0.$$

Therefore, we have:

$$\liminf_{N \to \infty} \frac{\sum_{k \neq k^*_\ell} \mathbb{E}_p[N^u_k \mid u \in \mathcal{U}_\ell, N_u(T) = N] \mathrm{kl}(p_{k\ell}, p_{k\ell'})}{\log(N)} \geq 1.$$

This concludes the proof of Theorem 8. □

# G  Performance guarantees of ECT: Proof of Theorem 4

The proof consists in several parts. First we study the initial sampling procedure (at the beginning of the exploration phase). We then upper bound the regret induced by the exploration phase. We analyze the performance of the clustering part of the algorithm, and finally upper bound the regret generated during the test phase.

**Item sampling procedure.** Let $\tilde{\mathcal{I}}_k = \mathcal{S} \cap \mathcal{I}_k$ be the set of items from $\mathcal{I}_k$ that are sampled. Then, for $\epsilon_1 = \sqrt{\frac{\log TK}{2(\min_k \alpha_k)^2 (\log T)^2}}$,

$$\mathbb{P}\left(|\tilde{\mathcal{I}}_k| \leq \alpha_k (1 - \epsilon_1)(\log T)^2\right) \overset{(a)}{\leq} \exp\left(-(\log T)^2 \mathrm{kl}((1 - \epsilon_1)\alpha_k, \alpha_k)\right)$$

$$\overset{(b)}{\leq} \exp\left(-(\log T)^2 2\epsilon_1^2 \alpha_k^2\right)$$

$$\leq \frac{1}{TK}.$$

where (a) is from Chernoff-Hoeffding bound (b) is from Pinsker's inequality.

Hence, the event $\mathcal{A}_1 = \{|\tilde{\mathcal{I}}_k| \geq \alpha_k (1 - \epsilon_1)(\log T)^2 \text{ for } 1 \leq k \leq K\}$ holds with probability at least $1 - \frac{1}{T}$. As a consequence, the expected regret due the event $\mathcal{A}_1^c$ is $O(1)$. Thus, we can assume that the event $\mathcal{A}_1$ holds throughout the remaining of the proof.

**Exploration phase.** In this phase, we wish to recommend each item in $\mathcal{S}$ for $\log T$ times. We prove that this exploration phase takes around $(\log T)^3$ rounds (and this is the regret it generates). Let us consider a user $u$. This user can make the exploration phase longer if it arrives more than $(\log T)^2$ during the $(\log T)^3$ first rounds. We have:

$$\mathbb{P}\left(N_u((\log T)^3) \geq (\log T)^2\right) \leq \exp\left(-(\log T)^3 \mathrm{kl}\left(\frac{1}{\log T}, \frac{1}{m}\right)\right)$$

$$\leq \exp\left(-2(\log T)^3 \left(\frac{1}{\log T} - \frac{1}{m}\right)^2\right)$$

$$\overset{(a)}{\leq} \exp\left(-2\left(\log T - \frac{2}{\log T}\right)\right) \tag{13}$$

where (a) is obtained from $m > (\log T)^3$ (remember that $T = o(m^2)$).

We deduce the probability that the duration of exploration phase $T_{\exp}$ exceeds $(\log T)^3$,

$$\mathbb{P}(T_{\exp} \geq (\log T)^3) = \mathbb{P}(\exists u \text{ s.t. } N_u((\log T)^3) \geq (\log T)^2)$$

$$\overset{(a)}{\leq} \frac{(\log T)^3}{T^2} \exp\left(\frac{4}{\log T}\right)$$

where (a) is obtained from the union bound and (13).

Now the expected time taken in the exploration phase is,

$$\mathbb{E}[T_{\exp}] = (\log T)^3 + \mathbb{E}[T_{\exp}|T_{\exp} \geq (\log T)^3]\mathbb{P}(T_{\exp} \geq (\log T)^3)$$

$$\leq (\log T)^3 + \frac{(\log T)^3}{T} \exp\left(\frac{4}{\log T}\right)$$

$$= \mathcal{O}((\log T)^3).$$

Therefore, we can conclude that the expected regret that occurs in the exploration phase is $\mathcal{O}((\log T)^3)$.

**Clustering phase.** The performance of the clustering phase can be analyzed using the same arguments as in the proof of Theorem 6 in [32]. To simplify the notation, let $\epsilon = (\log T)^{-\frac{1}{4}}$. Recall that $Q_i = \{j \in \mathcal{S} : |\hat{\rho}_i - \hat{\rho}_j| \leq \epsilon\}$ for all $i \in \mathcal{S}$. We also define a set $\mathcal{B}_k$ for $1 \leq k \leq K$ as:

$$\mathcal{B}_k = \{i \in \mathcal{S} : |p_k - \hat{\rho}_i| \leq \frac{1}{2}\epsilon\}.$$

This set has the following properties:

(i) $|\mathcal{B}_k| = \Omega((\log T)^2)$ with probability at least $1 - \frac{1}{T}$. This follows from the following argument.

$$\mathbb{P}(i \in \mathcal{B}_k) \geq \mathbb{P}(i \in \mathcal{B}_k | i \in \tilde{\mathcal{I}}_k)\mathbb{P}(i \in \tilde{\mathcal{I}}_k)$$

$$\overset{(a)}{\geq} \alpha_k(1 - \epsilon_1)\mathbb{P}\left(|p_k - \hat{\rho}_i| \leq \frac{1}{2}\epsilon \Big| i \in \tilde{\mathcal{I}}_k\right)$$

$$\overset{(b)}{\geq} \alpha_k(1 - \epsilon_1)\left(1 - 2\exp\left(-\frac{\sqrt{\log T}}{2}\right)\right),$$

where (a) follows from the assumption that $\mathcal{A}_1$ holds and (b) stems from Chernoff-Hoeffding's bound. Let $r = \alpha_k(1 - \epsilon_1)\left(1 - 2\exp\left(-\frac{\sqrt{\log T}}{2}\right)\right)$. Then,

$$\mathbb{P}\left(|\mathcal{B}_k| < \left(r - \frac{1}{\sqrt{2\log T}}\right)(\log T)^2\right) \leq \exp\left(-(\log T)^2 \mathrm{kl}\left(r - \frac{1}{\sqrt{2\log T}}, r\right)\right)$$

$$\leq \exp\left(-2(\log T)^2\left(\frac{1}{\sqrt{2\log T}}\right)^2\right)$$

$$\leq \frac{1}{T}.$$

Therefore, $|\mathcal{B}_k| = \Omega((\log T)^2)$ with probability at least $1 - \frac{1}{T}$.

(ii) $\left|\mathcal{S} \setminus (\cup_{k=1}^{K}\mathcal{B}_k)\right| = \mathcal{O}(\log T)$ with probability at least $1 - \frac{1}{T}$. To show this, we use a similar argument as in (i):

$$\mathbb{P}(i \in S \setminus (\cup_{k=1}^{K}\mathcal{B}_k)) \leq \sum_{k=1}^{K}\mathbb{P}(i \in \mathcal{I}_k)\mathbb{P}(i \in S \setminus (\cup_{k=1}^{K}\mathcal{B}_k) | i \in \mathcal{I}_k)$$

$$\leq \sum_{k=1}^{K}\mathbb{P}(i \in \mathcal{I}_k)\mathbb{P}\left(|p_k - \hat{\rho}_i| > \frac{1}{2}\epsilon \Big| i \in \mathcal{I}_k\right)$$

$$\leq \sum_{k=1}^{K}\mathbb{P}(i \in \mathcal{I}_k)2\exp\left(-\frac{\sqrt{\log T}}{2}\right)$$

$$= 2\exp\left(-\frac{\sqrt{\log T}}{2}\right)$$

Then, the probability that the size of $\left|\mathcal{S} \setminus (\cup_{k=1}^{K}\mathcal{B}_k)\right|$ is greater than $\log T$ is,

$$\mathbb{P}\left(\left|\mathcal{S} \setminus (\cup_{k=1}^{K}\mathcal{B}_k)\right| \geq \log T\right)$$

$$\leq \exp\left(-(\log T)^2\mathrm{kl}\left(\frac{1}{\log T}, 2\exp\left(-\frac{\sqrt{\log T}}{2}\right)\right)\right)$$

$$\overset{(a)}{\leq} \exp\left(-(\log T)\left(\frac{\sqrt{\log T}}{4} + 2\exp\left(-\frac{\sqrt{\log T}}{2}\right) - \log 2 - 1\right)\right)$$

$$\leq \frac{1}{T},$$

where (a) is obtained from Lemma 4 when $\log T \geq 2^4$.

(iii) If $|\mathcal{B}_k \cap Q_i| \geq 1$, then $|\mathcal{B}_j \cap Q_i| = 0$ for all $j, k$ such that $|p_k - p_j| = \Theta(1)$. Because $|\hat{\rho}_i - \hat{\rho}_l| \geq |p_k - p_j| - |p_k - \hat{\rho}_i| - |p_j - \hat{\rho}_l| \geq |p_k - p_j| - \epsilon$ for $i \in \mathcal{B}_k$ and $l \in \mathcal{B}_j$.

(iv) $\mathcal{B}_k \subset Q_i$ for all $i \in \mathcal{B}_k$, since $|\hat{\rho}_i - \hat{\rho}_j| \leq |\hat{\rho}_i - p_k| + |\hat{\rho}_j - p_k| \leq \epsilon$ for all $j \in \mathcal{B}_k$.

From properties (iii) and (iv), there exists an item $i \in (\cup_{k=1}^{K}\mathcal{B}_k) \setminus (\cup_{\ell=1}^{k-1}Q_{i_\ell})$ such that $|Q_i \setminus (\cup_{\ell=1}^{k-1}Q_{i_\ell})| \geq m_k$ where $m_k$ is the $k$-th largest value among $\{|\mathcal{B}_1|, ..., |\mathcal{B}_{K_0}|\}$ for $K_0 = |\{p_k : 1 \leq k \leq K\}|$. Here, $m_k = \Omega((\log T^2)^2)$ from property (i).

We also have $|Q_v| = \mathcal{O}(\log T)$ for $v$ such that $|Q_v \cap (\cup_{k=1}^{K}\mathcal{B}_k))| = 0$ from property (ii). Thus, the item $v$ cannot be chosen as $i_k$.

We can conclude that $|p_k - \hat{p}_k| \leq \epsilon$ for $k = 1, 2$ with probability $1 - 2/T$, since $|\hat{\rho}_i - p_k| \leq \epsilon$ when $|Q_i \cap \mathcal{B}_k| \geq 1$. Hence as before for event $\mathcal{A}_1$, we can assume that $\mathcal{A}_2 = \{|p_k - \hat{p}_k| \leq \epsilon$ for $k = 1, 2\}$ holds in the remaining of the proof.

**Test phase.** After $n$ recommendations of an item $i$ from $\mathcal{I}_k \neq \mathcal{I}_1$, the probability that $i$ passes the test is,

$$
\begin{aligned}
\mathbb{P}\left(\hat{\rho}_i > \hat{p}_1 - \Delta_0/2\right) &= \mathbb{P}\left(\hat{\rho}_i > \frac{1}{2}(\hat{p}_1 + \hat{p}_2)\right) \\
&\overset{(a)}{\leq} \mathbb{P}\left(\hat{\rho}_i > \frac{1}{2}(p_1 + p_2) - \epsilon\right) \\
&\leq \exp\left(-n\mathrm{kl}\left(\frac{1}{2}(p_1 + p_2) - \epsilon, p_k\right)\right) \\
&\leq \exp\left(-2n\left(\frac{1}{2}(p_1 + p_2) - p_k - \epsilon\right)^2\right)
\end{aligned}
$$

where (a) is obtained from the assumption that $\mathcal{A}_2$ holds.

To simplify the notation, let $x = \lfloor \frac{2\log 3}{\Delta_0^2} \rfloor$. Since we test the item after every $x$ recommendations, we have at most $m/x$ tests for each item. Therefore, the expected number of times a sub-optimal item $j$ is recommended is:

$$
\begin{aligned}
\mathbb{E}\left[N(j)\right] &= x + \sum_{i=1}^{m/x} x\mathbb{P}\left(\hat{\rho}_k > \hat{p}_1 - \Delta_0/2 \text{ for } i\text{-th test}\right) \\
&\leq x + \sum_{i=1}^{m/x} x\exp\left(-2ix\left(\frac{1}{2}(p_1 + p_2) - p_k - \epsilon\right)^2\right) \\
&\leq x + \frac{2}{(p_1 + p_2 - 2p_k - 2\epsilon)^2} \\
&\leq \frac{4\log 3}{(p_1 - p_2)^2} + \frac{2}{(p_1 + p_2 - 2p_k - 2\epsilon)^2}.
\end{aligned}
\tag{14}
$$

Furthermore, the probability that item $i \in I_1$ is not removed until the last test is,

$$
\begin{aligned}
\mathbb{P}\left(\bigcap_{t=1}^{m/x} \{\hat{\rho}_i > \hat{p}_1 - \Delta_0/2 \text{ for } t\text{-th test}\}\right) &\geq 1 - \sum_{t=1}^{m/x} \mathbb{P}\left(\hat{\rho}_i \leq \hat{p}_1 - \Delta_0/2 \text{ for } t\text{-th test}\right) \\
&= 1 - \sum_{t=1}^{m/x} \mathbb{P}\left(\hat{\rho}_i < \frac{1}{2}(p_1 + p_2) - \epsilon\right) \\
&\geq 1 - \sum_{t=1}^{m/x} \exp\left(-tx\mathrm{kl}\left(\frac{1}{2}(p_1 + p_2) - \epsilon, p_1\right)\right) \\
&\geq 1 - \sum_{t=1}^{m/x} \exp\left(-\frac{1}{2}tx\left(p_1 - p_2 + 2\epsilon\right)^2\right) \\
&\geq 1 - \frac{\exp\left(-\frac{1}{2}x\left(p_1 - p_2 + 2\epsilon\right)^2\right)}{1 - \exp\left(-\frac{1}{2}x\left(p_1 - p_2 + 2\epsilon\right)^2\right)} \\
&\geq \frac{1}{2}.
\end{aligned}
\tag{15}
$$

By (15), if we assume that user arrives for $\overline{N}$ times at most, we need at most $2\overline{N}$ optimal items in $\mathcal{V}$ in expectation. Thus, the required number of new samples from $\mathcal{I} \setminus \mathcal{V}_0$ is less than $\frac{2\overline{N}}{\alpha_1}$. Therefore,

from (14), the expected regret that occurs in the test phase under the assumption that every user arrives for less than $\overline{N}$ times is,

$$\mathcal{O}\left(\frac{2\overline{N}}{\alpha_1} \sum_{k=2}^{K} \alpha_k (p_1 - p_k) \left(\frac{4 \log 3}{(p_1 - p_2)^2} + \frac{2}{(p_1 + p_2 - 2p_k)^2}\right)\right).$$

On the other hand, the regret due to more than $\overline{N}$ arrivals of users is $\frac{m}{m(e-1)} = O(1)$ by Lemma 1.

Finally, the expected regret of ECT satisfies:

$$R^{ECT}(T) = \mathcal{O}\left((\log T)^3 + \frac{2\overline{N}}{\alpha_1} \sum_{k=2}^{K} \left[\alpha_k (p_1 - p_k) \left(\frac{4 \log 3}{(p_1 - p_2)^2} + \frac{2}{(p_1 + p_2 - 2p_k)^2}\right)\right]\right).$$

$\square$

# H   Performance guarantees of ET: Proof of Theorem 5

Recall that $\mu_x$ is the expected reward such that $\mathbb{P}(\rho_i \leq \mu_x) = x$, and that we are interested in the satisficing regret defined by:

$$R_\varepsilon^\pi(T) = \mathbb{E}^\pi \left( \sum_{t=1}^T \max\{0, \mu_{1-\varepsilon} - \rho_{i_t^\pi}\} \right).$$

We consider the case where $\varepsilon \geq C\sqrt{\frac{\pi}{2\log T}}$. Further recall that we assume that $\zeta(x) \leq C$ for all $x \in [0, 1]$.

To prove Theorem 5, we first analyze the performance of the exploration phase, and in particular show that $\hat{\mu}_{1-\frac{\varepsilon}{2}}$ is very close to $\mu_{1-\varepsilon}$. We then study the regret generated during the test phase.

**Exploration Phase.** We first derive an upper and a lower bound of $\hat{\mu}_{1-\frac{\varepsilon}{2}}$. Here, we use the fact that for all $i \in \mathcal{S}$, the $\hat{\rho}_i$'s are i.i.d. random variables.

From Chernoff bound and Pinsker's inequality,

$$
\begin{aligned}
\mathbb{P}(\hat{\rho}_i \geq \mu_{1-\frac{\varepsilon}{4}}) \leq & \frac{\varepsilon}{4} + \int_0^{\mu_{1-\frac{\varepsilon}{4}}} \exp\left(-2^4 \log(T)\mathrm{kl}(\mu_{1-\frac{\varepsilon}{4}}, \mu)\right) \zeta(\mu)d\mu \\
\leq & \frac{\varepsilon}{4} + \int_0^{\mu_{1-\frac{\varepsilon}{4}}} \exp\left(-2^5 (\mu_{1-\frac{\varepsilon}{4}} - \mu)^2 \log(T)\right) \zeta(\mu)d\mu \\
\leq & \frac{\varepsilon}{4} + \int_0^\infty C \exp\left(-2^5 x^2 \log(T)\right) dx \\
\leq & \frac{\varepsilon}{4} + \frac{\varepsilon}{8} = \frac{3\varepsilon}{8},
\end{aligned}
\tag{16}
$$

where for the last inequality, we use the Gaussian integral $\int_{-\infty}^\infty e^{-x^2} dx = \sqrt{\pi}$. When $\varepsilon \geq C\sqrt{\frac{\pi}{2\log T}}$,

$$
\begin{aligned}
\int_0^\infty C \exp\left(-2^5 x^2 \log(T)\right) dx = & \frac{1}{2} \int_{-\infty}^\infty \frac{C}{\sqrt{2^5 \log(T)}} \exp\left(-x^2\right) dx \\
= & \frac{1}{8} C \sqrt{\frac{\pi}{2\log T}} \leq \frac{\varepsilon}{8}.
\end{aligned}
$$

Similarly,

$$
\begin{aligned}
\mathbb{P}(\hat{\rho}_i \leq \mu_{1-\frac{3\varepsilon}{4}}) \leq & 1 - \frac{3\varepsilon}{4} + \int_{\mu_{1-\frac{3\varepsilon}{4}}}^1 e^{-2^4 \log(T)\mathrm{kl}(\mu_{1-\frac{3\varepsilon}{4}}, \mu)} \zeta(\mu)d\mu \\
\leq & 1 - \frac{3\varepsilon}{4} + \int_{\mu_{1-\frac{3\varepsilon}{4}}}^1 e^{-2^5 (\mu_{1-\frac{3\varepsilon}{4}} - \mu)^2 \log(T)} \zeta(\mu)d\mu \\
\leq & 1 - \frac{3\varepsilon}{4} + \int_0^\infty C \exp\left(-2^5 \mu^2 \log(T)\right) d\mu \\
< & 1 - \frac{3\varepsilon}{4} + \frac{\varepsilon}{8} \leq 1 - \frac{5\varepsilon}{8}.
\end{aligned}
\tag{17}
$$

From the Chernoff-Hoeffding and (16),

$$
\begin{aligned}
\mathbb{P}\left(\left|\{i \in \mathcal{S} : \hat{\rho}_i \geq \mu_{1-\frac{\varepsilon}{4}}\}\right| \geq \frac{\varepsilon}{2}|\mathcal{S}|\right) \leq & \exp\left(-|\mathcal{S}|\mathrm{kl}(\frac{\varepsilon}{2}, \frac{3\varepsilon}{8})\right) \\
\leq & \exp\left(-2|\mathcal{S}|\frac{\varepsilon^2}{8^2}\right) \leq \frac{1}{T^2}.
\end{aligned}
\tag{18}
$$

From the Chernoff-Hoeffding and (17),

$$
\begin{aligned}
\mathbb{P}\left(\left|\{i \in \mathcal{S} : \hat{\rho}_i \leq \mu_{1-\frac{3\varepsilon}{4}}\}\right| \leq (1 - \frac{\varepsilon}{2})|\mathcal{S}|\right) \leq & \exp\left(-|\mathcal{S}|\mathrm{kl}(1 - \frac{\varepsilon}{2}, 1 - \frac{5\varepsilon}{8})\right) \\
\leq & \exp\left(-2|\mathcal{S}|\frac{\varepsilon^2}{8^2}\right) \leq \frac{1}{T^2}.
\end{aligned}
\tag{19}
$$

We conclude from (18) and (19) that with probability $1 - \frac{2}{T^2}$, we have

$$\mu_{1-\frac{3\varepsilon}{4}} \le \hat{\mu}_{1-\frac{\varepsilon}{2}} \le \mu_{1-\frac{\varepsilon}{4}}. \tag{20}$$

Further observe that using the same arguments as those used to upper bound the duration of the exploration phase of the ECT algorithm, the expected duration, and hence the expected regret, of the exploration phase in ET is $O(\frac{(\log T)^2}{\varepsilon^2})$.

**Test Phase.** For convenience, let $\Delta = \log \log_2(2^e m^2)$. Then, ET runs at most $\tau = \lfloor \log_2(\frac{m}{\Delta}) \rfloor$ tests for each item. We define the distance $\mathcal{D}$ between two Bernoulli distributions as follows:

$$\mathcal{D}(p,q) = \mathrm{kl}(s,p) = \mathrm{kl}(s,q) \quad \text{with} \quad s = \frac{\log \frac{1-p}{1-q}}{\log \frac{q(1-p)}{p(1-q)}} \quad \text{for} \quad p \ne q.$$

Let $m^\pi(\mu)$ be the expected number of users to whom a randomly selected item with parameter $\mu$ is recommended. Let $\rho^{(\ell)}(\mu)$ be the random value $\hat{\rho}_i$ of item $i$ having $\mu$ after $\lfloor 2^\ell \Delta \rfloor$ observations.

Consider items having $\mu$ such that $\mu \le \hat{\mu}_{1-\frac{\varepsilon}{2}}$ and $2^{-\ell} \le \mathcal{D}(\mu, \hat{\mu}_{1-\frac{\varepsilon}{2}})$. Then, $\mathrm{kl}(\bar{\rho}^{(\ell)}, \mu) \ge 2^{-\ell}$ and we have

$$\begin{aligned}
m^\pi(\mu) &\le 2^\ell \Delta + \sum_{r=\ell}^\tau 2^{r+1}\Delta \mathbb{P}\left(\hat{\rho}^{(r)}(\mu) \ge \bar{\rho}^{(r)}\right) \\
&\le 2^\ell \Delta + \sum_{r=\ell}^\tau 2^{r+1}\Delta \mathbb{P}\left(\hat{\rho}^{(r)}(\mu) \ge \bar{\rho}^{(\ell)}\right) \\
&\le 2^\ell \Delta + \sum_{r=\ell}^\tau 2^{r+1}\Delta \exp\left(-(2^r \Delta)\mathrm{kl}(\bar{\rho}^{(\ell)}, \mu)\right) \\
&\le 2^\ell \Delta + \sum_{r=\ell}^\tau 2^{r+1}\Delta \exp\left(-2^{r-\ell}\Delta\right) \\
&\le 2^{\ell+2}\Delta. \tag{21}
\end{aligned}$$

From (21),

$$m^\pi(\mu) \le \begin{cases} 2^3\Delta & \text{for } 2^{-1} \le \mathcal{D}(\mu, \hat{\mu}_{1-\frac{\varepsilon}{2}}) \\ \frac{2^3\Delta}{\mathcal{D}(\mu,\hat{\mu}_{1-\frac{\varepsilon}{2}})} & \text{for } 2^{-\ell} \le \mathcal{D}(\mu, \hat{\mu}_{1-\frac{\varepsilon}{2}}) \le 2^{-\ell+1} \\ m & \text{for } \mathcal{D}(\mu, \hat{\mu}_{1-\frac{\varepsilon}{2}}) \le 2^{-\tau} \end{cases} \tag{22}$$

Next we study the expected regret generated by recommending a newly sampled item. From the regret definition and (21),

$$\begin{aligned}
\frac{1}{8\Delta} \int_0^{\mu_{1-\varepsilon}} (\mu_{1-\varepsilon} - \mu)m^\pi(\mu)\zeta(\mu)d\mu &\le \int_0^{\mu_{1-\varepsilon}} (\mu_{1-\varepsilon} - \mu)\left(1 + \frac{1}{\mathcal{D}(\mu, \mu_{1-\frac{3\varepsilon}{4}})}\right)\zeta(\mu)d\mu \\
&\le \int_0^{\mu_{1-\varepsilon}} C(\mu_{1-\varepsilon} - \mu)\left(1 + \frac{2}{(\mu - \mu_{1-\frac{3\varepsilon}{4}})^2}\right)d\mu \\
&\le \int_0^{\mu_{1-\varepsilon}} C(\mu_{1-\varepsilon} - \mu)\left(1 + \frac{2}{(\frac{\varepsilon}{4C} + \mu_{1-\varepsilon} - \mu)^2}\right)d\mu \\
&\le \frac{C}{2} + \int_0^{\mu_{1-\varepsilon}} \frac{2C}{\frac{\varepsilon}{4C} + \mu_{1-\varepsilon} - \mu}d\mu \\
&\le \frac{C}{2} + \log(4C/\varepsilon), \tag{23}
\end{aligned}$$

where the second inequality stems from Pinsker's inequality $2(p-q)^2 \le \mathrm{kl}(p,q)$ and the definition of $\mathcal{D}$, and the third inequality uses the assumption $\zeta(\mu) \le C$.

If an item has a parameter $\mu \geq \hat{\mu}_{1-\frac{\varepsilon}{2}}$, we do not remove it from $\mathcal{V}$ with probability at least

$$\mathbb{P}\left(\bigcap_{\ell=1}^{\tau}\left\{\hat{\rho}^{(\ell)}(\mu) > \bar{\rho}^{(\ell)}\right\}\right) \geq 1 - \sum_{\ell=1}^{\tau}\mathbb{P}\left(\hat{\rho}^{(\ell)}(\mu) \leq \bar{\rho}^{(\ell)}\right)$$

$$\geq 1 - \sum_{i=1}^{\tau} e^{-\Delta}$$

$$= 1 - \frac{1}{\log_2(2^e m^2)}\frac{1 - e^{-\tau\Delta}}{1 - \frac{1}{\log_2(2^e m^2)}}$$

$$\geq 1 - \frac{1}{\log_2(2^e m^2) - 1}$$

$$\geq \frac{1}{2}.$$

To recommend items $i$ with parameters $\mu_i \geq \mu_{1-\varepsilon}$ to the $\overline{N}$ arrivals of every user, we then need, on average, $\frac{2\overline{N}}{\varepsilon}$ sampled items. From (23), we conclude that the satisficing regret of ET satisfies:

$$R_\varepsilon^\pi(T) = \mathcal{O}\left(\frac{\overline{N}\log(e/\varepsilon)\log\log(m)}{\varepsilon} + \frac{(\log T)^2}{\varepsilon^2}\right).$$

$\square$

# I Performance guarantees of EC-UCS: Proof of Theorem 6

These two last sections I and J of the appendix are devoted to the analysis of the regret of EC-UCS and ECB in systems with clustered items and users. The two algorithms share the same initial phase to cluster items. The next subsection is hence devoted to the analysis of this item clustering phase. Then, we present an analysis of the performance of the other phases of EC-UCS, and conclude this section with the statement and proof of lemmas used in the analysis of EC-UCS.

## I.1 Clustering items in EC-UCS and ECB

The exploration phase for item clustering is of duration $10m$, and hence induces a regret upper bounded by $10m$. In what follows, we just investigate the quality of the item clusters that result from this phase.

Recall that the algorithm randomly selects a set $\mathcal{S}$ of items to cluster. We denote by $V_1, \ldots, V_K$ the true cluster $\mathcal{S} \cap \mathcal{I}_1, \ldots, \mathcal{S} \cap \mathcal{I}_K$, respectively, and assume that $m^2 \geq T(\log T)^3$ and $n = \omega(\log T)$. We let $n_0 := \min\{n, \frac{m}{(\log T)^2}\}$ be the number of sampled items. For each $k$, the size of $V_k$ concentrates around $\alpha_k n_0$. Indeed, from the Chernoff-Hoeffding's inequality,

$$\mathbb{P}\left( ||V_k| - \alpha_k n_0| \geq \sqrt{n_0 \log T} \right) \leq \frac{2}{T^2}. \tag{24}$$

Since $n_0 = \omega(\log T)$, we have

$$|V_k| = \alpha_k n_0 (1 + o(1)) \quad \text{for all} \quad k \in [K]. \tag{25}$$

Then, $|V_k| \geq \overline{N}$ for all $1 \leq k \leq K$ since $n_0 = \omega(\log(m) + \frac{T}{m})$. Therefore, all users arriving after the exploration phase could be potentially recommended by items from a single cluster $V_k$ without repetition.

Recall the procedure used by EC-UCS to cluster items in $\mathcal{S}$. For the $10m$ first user arrivals, it recommends items from $\mathcal{S}$ uniformly at random. These $10m$ recommendations and the corresponding user responses are recorded in the dataset $\mathcal{D}$. From the dataset $\mathcal{D}$, the item clusters are extracted using a spectral algorithm (see Algorithm 4). This algorithm is taken from [33], and considers the *indirect edges* between items created by users. Specifically, when a user appears more than twice in $\mathcal{D}$, she creates an indirect edge between the items recommended to her for which she provided the same answer (1 or 0). Items with indirect edges are more likely to belong to the same cluster.

Algorithm 4 builds an adjacency matrix $A$ from indirect edges. From Lemma 6 (presented in Appendix B), we know that at least $m/2$ users arrive twice in the first $10m$ arrivals with probability at least $1 - \frac{1}{T}$. We conclude that the construction of $A$ is equivalent to a stochastic block model with random sampling where the number of vertices is $n_0$, the sampling budget is $s \geq m/2$. We establish in the next theorem that this budget is enough to reconstruct the clusters $V_1, \ldots, V_K$ exactly using Algorithm 4. Theorem 9 is proved in Appendix J.2.

**Theorem 9.** *Let $\hat{I}_1, \ldots, \hat{I}_K$ be the output of Algorithm 4. With probability $1 - \frac{1}{T}$, there exists permutation $\Gamma$ such that*

$$\left| \bigcup_{k=1}^{K} (\hat{I}_{\Gamma(k)} \setminus V_k) \right| = 0.$$

## I.2 Regret of EC-UCS: Proof of Theorem 6

The first component of the regret of EC-UCS is generated during the exploration phase for item clustering. This component is $\mathcal{O}(m)$. Then in view of Theorem 9, errors in item clustering cannot generate more than a $\mathcal{O}(1)$ regret. Hence, in what follows, we always assume that after the item clustering phase, we have:

$$\left| \bigcup_{k=1}^{K} (\hat{I}_{\Gamma(k)} \setminus V_k) \right| = 0.$$

Without loss of generality, we assume that $\Gamma(k) = k$ in the remaining of this section. After the item clustering phase, there are four sources of regret referred to as: 1. Exploration for user clustering, 2. Arrival of reference users, 3. User clustering, and 4. Optimistic assignments.

**1. Exploration for user clustering.** The regret induced by exploration of the users in $\mathcal{U}_0$ until $t \leq (10 + \log T)m$ is $\frac{m \log T}{\log T} = m$. Hence, the regret due to this step is:

$$\mathcal{O}\left( m \sum_\ell \beta_\ell (p_{\sigma_\ell(1)\ell} - p_{\sigma_\ell(K)\ell}) \right).$$

**2. Arrival of reference users.** If the users in $\mathcal{U}^*$ have not arrived enough times, the algorithm cannot cluster them as intended, and this generates regret. Let $n_u$ denote the number of times user $u$ has arrived (until a time that will always be specified).

We define the event $\mathcal{E}_{top}^{(i)} = \{\exists u \in \mathcal{U}^*$ such that $n_u \leq \frac{(9+2^i)}{2}$ at $t = (9 + 2^i)m\}$ for $0 \leq i \leq \log_2 \log T$. Then, by Lemma 14, the regret due to $\mathcal{E}_{top}^{(i)}$ until $t = \lfloor (10 + \log T)m \rfloor$ is,

$R_{ref}(\lfloor (10 + \log T)m \rfloor)$

$$\leq \sum_\ell \beta_\ell(p_{\sigma_\ell(1)\ell} - p_{\sigma_\ell(K)\ell}) \sum_{i=0}^{\lfloor \log_2 \log T \rfloor} 2^{i+1} m \left( \frac{me}{(\log T)^3} \right)^{(\log T)^2} \exp\left( -\frac{(9+2^i)m}{16 \log T} \right)$$

$$\leq \sum_\ell \beta_\ell(p_{\sigma_\ell(1)\ell} - p_{\sigma_\ell(K)\ell}) m \left( \frac{me}{(\log T)^3} \right)^{(\log T)^2} \int_{-1}^{\log_2 \log T} 2^{x+1} \exp\left( -\frac{(9+2^x)m}{16 \log T} \right) dx$$

$$= \sum_\ell \beta_\ell(p_{\sigma_\ell(1)\ell} - p_{\sigma_\ell(K)\ell}) m \left( \frac{me}{(\log T)^3} \right)^{(\log T)^2} \int_{1/2}^{\log T} \frac{2}{\log 2} \exp\left( -\frac{(9+y)m}{16 \log T} \right) dy$$

$$= \sum_\ell \beta_\ell(p_{\sigma_\ell(1)\ell} - p_{\sigma_\ell(K)\ell}) m \left( \frac{me}{(\log T)^3} \right)^{(\log T)^2} \left[ -\frac{32 \log T}{m \log 2} \exp\left( -\frac{(9+y)m}{16 \log T} \right) \right]_{1/2}^{\log T}$$

$$= \sum_\ell \beta_\ell(p_{\sigma_\ell(1)\ell} - p_{\sigma_\ell(K)\ell}) \exp\left( -\Theta\left( \frac{m}{\log T} \right) \right). \tag{26}$$

where we have used the assumption $m^2 \geq T(\log T)^3$.

Also, $\mathbb{P}(\mathcal{E}_{top}^{(\lfloor \log_2 \log T \rfloor)}) \leq 2\left( \frac{me}{(\log T)^3} \right)^{(\log T)^2} \exp\left( -\frac{(9+\frac{1}{2}\log T)m}{4 \log T} \right) = \exp(-\Theta(m))$. Hence, in view of (26), the regret due to $\mathcal{E}_{top}^{(i)}$ satisfies

$$R_{ref}(T) \leq \sum_\ell \beta_\ell(p_{\sigma_\ell(1)\ell} - p_{\sigma_\ell(K)\ell}) \exp\left( -\Theta\left( \frac{m}{\log T} \right) \right). \tag{27}$$

Let $\mathcal{B}_1 = \left( \bigcup_{i=0}^{\lfloor \log_2 \log T \rfloor} \mathcal{E}_{top}^{(i)} \right)^c$, i.e., $\mathcal{B}_1$ correspond to the event where every $u \in \mathcal{U}^*$ has arrived $n_u > \frac{(9+2^i)}{2}$ times at $t = (9 + 2^i)m$ for all $i$. Then, from (27), the regret because of $\mathcal{B}_1^c$ is

$$R_{ref}(T) \leq \exp\left( -\Theta\left( \frac{m}{\log T} \right) \right). \tag{28}$$

**3. User clustering.** The size of $\mathcal{U}^*$ is sufficiently large, so that $\mathcal{U}^*$ consists of users from all clusters. More precisely, for a well-chosen $\epsilon_1 > 0$, Lemma 15 states that with probability $1 - 1/T$, the size of $\tilde{\mathcal{U}}_\ell = \mathcal{U}^* \cap \mathcal{U}_\ell$ is greater than $\beta_\ell(1 - \epsilon_1)(\log T)^2$ for all $\ell$. Let $\mathcal{B}_2 = \{|\tilde{\mathcal{U}}_\ell| \geq \beta_\ell(1 - \epsilon_1)(\log T)^2$ for $1 \leq \ell \leq L\}$. By Lemma 15, the expected regret due to the event $\mathcal{B}_2^c$ is $\mathcal{O}(1)$.

We now assume that both $\mathcal{B}_1$ and $\mathcal{B}_2$ holds throughout the remaining of the proof.

Under $\mathcal{B}_1$, we have numerous observations for users in $\mathcal{U}^*$. Hence, most of users in $\mathcal{U}^*$ have their empirical average success rate vector concentrated around the true parameter vector when $t$ is large. Therefore, under $\mathcal{B}_2$, the clustering step can learn the hidden parameters very accurately (since there are clear user clusters). We formalize this observation below. Consider $t \geq T_0 = \lceil Cm \rceil$ where

$$C = \max\left( \frac{512K^3}{\min(y_{\ell r}, \delta)^2} \log\left( \frac{16K^{\frac{3}{2}}}{\min(y_{\ell r}, \delta)} \right), \frac{2\sqrt{K}}{\min_\ell \beta_\ell} \right).$$

Then, we have

*(C1)* $K\sqrt{\frac{8Km}{T_0}\log\frac{T_0}{m}} < \frac{1}{4}\min(y_{\ell r},\delta)$,

*(C2)* $(1-\epsilon_1)\left(1-2K\left(\frac{m}{T_0}\right)^2\right)\min_\ell \beta_\ell > \frac{m}{T_0}$.

Recall that in EC-UCS (see the pseudo-code), we use a parameter $\epsilon > 0$ when clustering users. This parameter is fixed and equal to $\epsilon = K\sqrt{\frac{8Km}{t}\log\frac{t}{m}}$. From Lemma 16, under *(C1)* and *(C2)*, we have

$$\|p_\ell - \hat{p}_\ell\| < \epsilon < \frac{1}{4}\min(y_{\ell r},\delta) \quad \forall t \geq T_0 \quad \text{with probability} \quad 1-\frac{2}{T}.$$

Hence after $T_0$ rounds, the algorithm has accurate estimates of the parameters. We include $T_0$ in the regret upper bound, but can then assume that $\|p_\ell - \hat{p}_\ell\| < \epsilon < \frac{1}{4}\min(y_{\ell r},\delta)$ for all $t \geq T_0$ in the remaining of the proof. The expected regret generated by the complement of this event is $\mathcal{O}(1)$.

**4. Optimistic assignments.** When $\|p_\ell - \hat{p}_\ell\| < \epsilon$, the algorithm can exploit the learned parameters. Suppose $u_t \in \mathcal{U}_\ell$. Recall the notation: $\mathcal{L}(u_t) \leftarrow \{\ell \in [L_0] : \sum_{k=1}^K n_{ku_t}x_{k\ell}^2 < 2K\log n_{u_t}\}$ used in the pseudo-code of EC-UCS.

Since the probability of the event $\{\ell \notin \mathcal{L}(u_t)\}$ decreases rapidly with the number of arrivals of $u_t$, the regret induced by this event is $\mathcal{O}(1)$. Since $\ell \in \mathcal{L}(u_t)$ holds most of the time, the algorithm recommend optimal items in item cluster $k_\ell^*$ at least $\frac{n_{u_t}}{2K}$ times. If $r \notin \mathcal{L}^\perp(\ell)$, we can distinguish $\mathcal{U}_\ell$ from $\mathcal{U}_r$ with the constant number of recommendations of optimal item $k_\ell^*$, since $p_{k_\ell^*\ell} \neq p_{k_\ell^*r}$. On the other hand, if $r \in \mathcal{L}^\perp(\ell)$, the algorithm cannot distinguish them unless it plays suboptimal items. Actually, suboptimal items should be played at most $\mathcal{O}(\log \overline{N})$ times in expectation. We make the above observations precise in Lemma 17, from which we conclude that the regret generated in this phase is:

$$\mathcal{O}\left(m\sum_\ell \beta_\ell(p_{\sigma_\ell(1)\ell} - p_{\sigma_\ell(K)\ell})\left(\sum_{r\in\mathcal{R}_\ell\backslash\mathcal{L}^\perp(\ell)}\frac{K^2\log K}{\phi(|p_{k_\ell^*r}-p_{k_\ell^*\ell}|^2)} + \sum_{k\in\mathcal{S}_{\ell r}}\sum_{r\in\mathcal{L}^\perp(\ell)}\frac{K\log\overline{N}}{|\mathcal{S}_{\ell r}||p_{k\ell}-p_{kr}|^2}\right)\right)$$

Overall, accounting for all the regret sources, we have established that the regret of EC-UCS is:

$$\mathcal{O}\left(m\sum_\ell \beta_\ell(p_{\sigma_\ell(1)\ell} - p_{\sigma_\ell(K)\ell})\left(\max\left(\frac{K^3\log K}{\phi(\min(y_{\ell r},\delta)^2)},\frac{\sqrt{K}}{\min_\ell \beta_\ell}\right)\right.\right.$$

$$\left.\left.+ \sum_{r\in\mathcal{R}_\ell\backslash\mathcal{L}^\perp(\ell)}\frac{K^2\log K}{\phi(|p_{k_\ell^*r}-p_{k_\ell^*\ell}|^2)} + \sum_{k\in\mathcal{S}_{\ell r}}\sum_{r\in\mathcal{L}^\perp(\ell)}\frac{K\log\overline{N}}{|\mathcal{S}_{\ell r}||p_{k\ell}-p_{kr}|^2}\right)\right)$$

### I.3 Technical lemmas for the proof of Theorem 6

**Lemma 12** (Cramer's theorem)**.** *For any i.i.d. sequence* $(X_i)_{i\geq1}$ *of real r.v. and any closed set* $F \subseteq \mathbb{R}$,

$$\mathbb{P}\left(\frac{1}{n}\sum_{i=1}^n X_i \in F\right) \leq 2\exp\left(-n\inf_{x\in F}I(x)\right),$$

*where* $I(a) = \sup_{\theta\in\mathbb{R}}(\theta a - \log E[e^{\theta X}])$.

**Lemma 13.** *In the 'Exploitation' step (see EC-UCS pseudo-code), the expected regret generated by the exploration of* $u_t \in \mathcal{U}_0$ *until* $t = \lfloor(10+m)\log T\rfloor$ *is* $\mathcal{O}(m)$.

*Proof.* Since the probability that a user from $\mathcal{U}_0$ arrives for each time $t$ is $\frac{1}{\log T}$, the regret induced when exploring for users in $\mathcal{U}_0$ is $\mathcal{O}(\frac{m\log T}{\log T}) = \mathcal{O}(m)$. $\square$

**Lemma 14.** *In the 'Exploitation' step, with probability* $1 - 2\left(\frac{me}{(\log T)^3}\right)^{(\log T)^2}\exp\left(-\frac{t}{16\log T}\right)$, *at least* $\lfloor(\log T)^2\rfloor$ *users in* $\mathcal{U}_0$ *arrive at least* $\lfloor\frac{t}{2m}\rfloor$ *times within the first* $t$ *arrivals.*

*Proof.* We denote by $N_u(t)$ the number of times a user $u$ has arrived in the first $t$ arrivals. For any set $A \subset \mathcal{U}_0$, let $N_A(t)$ denote the total number of arrivals of users in $A$ among the first $t$ arrivals.

We write the probability that less than $(\log T)^2$ users in $\mathcal{U}_0$ arrive for $t/2m$ times in the $t$ first arrivals as:

$$\mathbb{P}[\sum_{u \in \mathcal{U}_0} \mathbb{1}\{N_u(t) \geq \frac{t}{2m}\} < (\log T)^2] = \mathbb{P}[\sum_{u \in \mathcal{U}_0} \mathbb{1}\{N_u(t) < \frac{t}{2m}\} \geq |\mathcal{U}_0| - (\log T)^2]$$

$$\leq \mathbb{P}[\exists A \subset \mathcal{U}_0 : |A| = |\mathcal{U}_0| - (\log T)^2, \forall u \in A, N_u(t) \leq \frac{t}{2m}]$$

$$\leq \mathbb{P}[\exists A \subset \mathcal{U}_0 : |A| = |\mathcal{U}_0| - (\log T)^2, N_A(t) \leq \frac{t(|\mathcal{U}_0| - (\log T)^2)}{2m}] \qquad (29)$$

$$\overset{(a)}{\leq} 2\binom{|\mathcal{U}_0|}{(\log T)^2} \exp\left(-\frac{t}{4}\log\left(1 + \frac{p}{2-2p}\right)\right)$$

$$\overset{(b)}{\leq} 2\left(\frac{me}{(\log T)^3}\right)^{(\log T)^2} \exp\left(-\frac{t}{16 \log T}\right),$$

where (b) follows from $\log(1+x) > \frac{x}{2}$ for $0 < x < 1$ and (a) can be proved using lemma 12. For simplicity, we define i.i.d. random variables $X_i \sim \text{Bern}(p)$ where $p = \frac{|\mathcal{U}_0| - (\log T)^2}{m}$. Then, $I(a) = a \log \frac{a(1-p)}{p(1-a)} - \log \frac{1-p}{1-a}$. Since $I(a)$ is a decreasing function in $(-\infty, p]$, $\inf_{a \leq \frac{p}{2}} I(a) = I\left(\frac{p}{2}\right)$. Therefore, (29) can be rewritten as:

$$\mathbb{P}\left(\frac{1}{t}\sum_{i=1}^{t} X_i \leq \frac{p}{2}\right) \leq 2 \exp\left(-tI\left(\frac{p}{2}\right)\right)$$

$$= 2 \exp\left(-t\left(\log \frac{2-p}{2-2p} + \frac{p}{2}\log \frac{1-p}{2-p}\right)\right)$$

$$\overset{(a)}{\leq} 2 \exp\left(-\frac{t}{4}\log \frac{2-p}{2-2p}\right).$$

where (a) holds since $\log(1+x) \geq x - \frac{x^2}{2}$ for $0 < x < 1$. $\qquad \square$

**Lemma 15.** *Fix* $\epsilon_1 = \sqrt{\frac{\log TK}{2(\min_\ell \beta_\ell)^2(\log T)^2}}$. *Let* $\tilde{\mathcal{U}}_\ell = \mathcal{U}^* \cap \mathcal{U}_\ell$ *and* $\mathcal{B}_2 = \{|\tilde{\mathcal{U}}_\ell| \geq \beta_\ell(1 - \epsilon_1)(\log T)^2 \text{ for } 1 \leq \ell \leq L\}$. *Then,* $\mathbb{P}(\mathcal{B}_2) \geq 1 - \frac{1}{T}$.

*Proof.* Since $\mathbb{P}(u \in \tilde{\mathcal{U}}_\ell | u \in \mathcal{U}^*) = \mathbb{P}(u \in \mathcal{U}_\ell | u \in \mathcal{U}^*) = \mathbb{P}(u \in \mathcal{U}_\ell) = \beta_\ell$,

$$\mathbb{P}\left(|\tilde{\mathcal{U}}_\ell| \leq \beta_\ell(1 - \epsilon_1)(\log T)^2\right) \overset{(a)}{\leq} \exp\left(-(\log T)^2 \text{kl}((1-\epsilon_1)\beta_\ell, \beta_\ell)\right)$$

$$\overset{(b)}{\leq} \exp\left(-(\log T)^2 2\epsilon_1^2 \beta_\ell^2\right)$$

$$\leq \frac{1}{TK}.$$

where (a) is from Chernoff-Hoeffding bound (b) is from Pinsker's inequality.

Hence, the event $\mathcal{B}_2 = \{|\tilde{\mathcal{U}}_\ell| \geq \beta_\ell(1 - \epsilon_1)(\log T)^2 \text{ for } 1 \leq \ell \leq L\}$ holds with probability at least $1 - \frac{1}{T}$. $\qquad \square$

In the remaining of this section, we fix $\epsilon_1 = \sqrt{\frac{\log TK}{2(\min_\ell \beta_\ell)^2(\log T)^2}}$ as chosen in the previous lemma.

**Lemma 16.** *In the 'Exploitation' step, under* $\mathcal{B}_1$ *and* $\mathcal{B}_2$, *if $t$ is large enough to satisfy the conditions* $\epsilon < \frac{1}{2}\min_{\ell \neq \ell'} \|p_\ell - p_{\ell'}\|$ *and* $(1 - \epsilon_1)\left(1 - 2K\left(\frac{m}{t}\right)^2\right)\min_\ell \beta_\ell > \frac{m}{t}$, *then* $\|p_\ell - \hat{p}_\ell\| < \epsilon$ *with probability at least* $1 - \frac{2}{T}$.

*Proof.* Recall that $\epsilon = K\sqrt{\frac{8Km}{t}\log\frac{t}{m}}$ and $Q_u = \{v \in \mathcal{U}^* : \|\hat{\rho}_u - \hat{\rho}_v\| \leq \epsilon\}$ for all $u \in \mathcal{U}^*$. We define a set $\mathcal{C}_\ell$ for $1 \leq \ell \leq L$ as: $\mathcal{C}_\ell = \{u \in \mathcal{U}^* : \|p_\ell - \hat{\rho}_u\| \leq \frac{\epsilon}{2}\}$. This set has the following properties:

(i) $|\mathcal{C}_\ell| = \Omega((\log T)^2)$ with probability at least $1 - \frac{1}{T}$. This follows from the following argument.

$$\mathbb{P}(u \in \mathcal{C}_\ell) \geq \mathbb{P}(u \in \mathcal{C}_\ell | u \in \mathcal{U}_\ell)\mathbb{P}(u \in \mathcal{U}_\ell)$$

$$\overset{(a)}{\geq} \beta_\ell(1-\epsilon_1)\mathbb{P}\left(\|p_\ell - \hat{\rho}_u\| \leq \frac{\epsilon}{2}\Big| u \in \mathcal{U}_\ell\right)$$

$$\overset{(b)}{\geq} \beta_\ell(1-\epsilon_1)\left(1 - 2\exp\left(-2\frac{t}{2Km}\left(\frac{\epsilon}{2K}\right)^2\right)\right)^K$$

$$\geq \beta_\ell(1-\epsilon_1)\left(1 - 2K\exp\left(-\frac{t\epsilon^2}{4K^3m}\right)\right)$$

$$\geq \beta_\ell(1-\epsilon_1)\left(1 - 2K\left(\frac{m}{t}\right)^2\right),$$

where (a) follows from the assumption that $\mathcal{B}_2$ holds and (b) stems from $\mathcal{B}_1$ and Chernoff-Hoeffding's bound.

Let $r = \beta_\ell(1-\epsilon_1)\left(1 - 2K\left(\frac{m}{t}\right)^2\right)$. Then,

$$\mathbb{P}\left(|\mathcal{C}_\ell| < \left(r - \frac{1}{\sqrt{2\log T}}\right)(\log T)^2\right) \leq \exp\left(-(\log T)^2\text{kl}\left(r - \frac{1}{\sqrt{2\log T}}, r\right)\right)$$

$$\leq \exp\left(-2(\log T)^2\left(\frac{1}{\sqrt{2\log T}}\right)^2\right)$$

$$\leq \frac{1}{T}.$$

Therefore, $|\mathcal{C}_\ell| = \Omega((\log T)^2)$ with probability at least $1 - \frac{1}{T}$.

(ii) $|\mathcal{U}^* \setminus (\cup_{\ell=1}^L \mathcal{C}_\ell)| = \mathcal{O}(\frac{m(\log T)^2}{t})$ with probability at least $1 - \frac{1}{T}$. To show this, we use a similar argument as in (i):

$$\mathbb{P}(u \in \mathcal{U}^* \setminus (\cup_{\ell=1}^L \mathcal{C}_\ell)) \leq \sum_{\ell=1}^L \mathbb{P}(u \in \mathcal{U}_\ell)\mathbb{P}(u \in \mathcal{U}^* \setminus (\cup_{\ell=1}^L \mathcal{C}_\ell) | u \in \mathcal{U}_\ell)$$

$$\leq \sum_{\ell=1}^L \mathbb{P}(u \in \mathcal{U}_\ell)\mathbb{P}\left(\|p_\ell - \hat{\rho}_u\| > \frac{\epsilon}{2}\Big| u \in \mathcal{U}_\ell\right)$$

$$\leq \sum_{\ell=1}^L \mathbb{P}(u \in \mathcal{U}_\ell)2K\left(\frac{m}{t}\right)^2$$

$$= 2K\left(\frac{m}{t}\right)^2.$$

Then, the probability that the size of $|\mathcal{U}^* \setminus (\cup_{\ell=1}^L \mathcal{C}_\ell)|$ is greater than $\frac{m(\log T)^2}{t}$ is,

$$\mathbb{P}\left(|\mathcal{U}^* \setminus (\cup_{\ell=1}^L \mathcal{C}_\ell)| \geq \frac{m(\log T)^2}{t}\right) \leq \exp\left(-(\log T)^2\text{kl}\left(\frac{m}{t}, 2K\left(\frac{m}{t}\right)^2\right)\right)$$

$$\overset{(a)}{\leq} \exp\left(-(\log T)^2\frac{m}{t}\log\frac{t}{m}\right)$$

$$\overset{(b)}{\leq} \frac{1}{T},$$

where (a) is obtained from Lemma 4 and $t \geq 2Km$ and (b) is from $t \leq m\log T$.

(iii) If $|\mathcal{C}_\ell \cap Q_u| \geq 1$, then $|\mathcal{C}_m \cap Q_u| = 0$ for all $\ell \neq m$. Because $\|\hat{\rho}_u - \hat{\rho}_v\| \geq \|p_\ell - p_m\| - \|p_\ell - \hat{\rho}_u\| - \|p_m - \hat{\rho}_j\| \geq \|p_\ell - p_m\| - \epsilon > \epsilon$ for $u \in \mathcal{C}_\ell$ and $j \in \mathcal{C}_m$, where the last inequality follows from $2\epsilon < \min_{\ell \neq \ell'} \|p_\ell - p_{\ell'}\|$.

(iv) $\mathcal{C}_\ell \subset Q_u$ for all $u \in \mathcal{C}_\ell$, since $\|\hat{\rho}_u - \hat{\rho}_v\| \leq \|\hat{\rho}_u - p_\ell\| + \|\hat{\rho}_v - p_\ell\| \leq \epsilon$ for all $v \in \mathcal{C}_\ell$.

From the properties (iii) and (iv), there exists an item $u \in (\cup_{\ell=1}^L \mathcal{C}_\ell) \setminus (\cup_{r=1}^{\ell-1} Q_{i_r})$ such that $|Q_u \setminus (\cup_{r=1}^{\ell-1} Q_{i_r})| \geq m_\ell$. Here, $m_\ell = \Omega((\log T)^2)$ from property (i).

We also have $|Q_v| = \mathcal{O}(\frac{m(\log T)^2}{t})$ for $v$ such that $|Q_v \cap (\cup_{k=1}^K \mathcal{C}_k))| = 0$ from property (ii). Since we assume $(1 - \epsilon_1) \left(1 - 2K \left(\frac{m}{t}\right)^2\right) \min_\ell \beta_\ell > \frac{m}{t}$, the item $v$ cannot be chosen as $i_k$.

We can conclude that $\|p_\ell - \hat{p}_\ell\| \leq \epsilon$ with probability $1 - 2/T$, since $\|\hat{\rho}_u - p_\ell\| \leq \epsilon$ when $|Q_u \cap \mathcal{C}_\ell| \geq 1$. $\qquad \square$

**Lemma 17.** *If $\|p_\ell - \hat{p}_\ell\| < \frac{1}{4}\min(y_{\ell r}, \delta)$ for all $r \neq \ell$, the regret due to recommendations based on optimistic user assignments is,*

$$\mathcal{O}\left(m \sum_\ell \beta_\ell (p_{\sigma_\ell(1)\ell} - p_{\sigma_\ell(K)\ell}) \left(\sum_{r \in \mathcal{R}_\ell \setminus \mathcal{L}^\perp(\ell)} \frac{K^2 \log K}{\phi(|p_{k_\ell^* r} - p_{k_\ell^* \ell}|^2)} + \sum_{k \in \mathcal{S}_{\ell r}} \sum_{r \in \mathcal{L}^\perp(\ell)} \frac{K \log \overline{N}}{|\mathcal{S}_{\ell r}| |p_{k\ell} - p_{kr}|^2}\right)\right).$$

*Proof.* Recall that $x_{k\ell} = \max\{|\hat{p}_{k\ell} - \hat{\rho}_{ku_t}| - \epsilon, 0\}$ and $\mathcal{L}^\perp(\ell) = \{\ell' \neq \ell : k_\ell^* \neq k_{\ell'}^*, p_{k_\ell^* \ell} = p_{k_\ell^* \ell'}\}$. We take $\epsilon < \frac{1}{4}\min(y_{\ell r}, \delta)$ to satisfy the condition $\|p_\ell - \hat{p}_\ell\| < \frac{1}{4}\min(y_{\ell r}, \delta)$. When we make a recommendation to $u_t \in \mathcal{U}_\ell$ by referring to their neighbors from $\mathcal{U}^*$, if regret is generated, the following event holds.

$$\mathcal{E}_N = \left\{\sum_{k=1}^K n_{ku_t} x_{k\ell}^2 > 2K \log n_{u_t}\right\}$$

$$\cup \left(\bigcup_{r \neq \ell} \left\{\sum_{k=1}^K n_{ku_t} x_{kr}^2 < 2K \log n_{u_t}\right\} \cap \left\{\arg\max_k p_{k\ell} \neq \arg\max_k p_{kr}\right\}\right)$$

$$= \left\{\sum_{k=1}^K n_{ku_t} x_{k\ell}^2 > 2K \log n_{u_t}\right\} \cup \left(\bigcup_{r \in \mathcal{R}_\ell} \left\{\sum_{k=1}^K n_{ku_t} x_{kr}^2 < 2K \log n_{u_t}\right\}\right)$$

$$:= \mathcal{E}_{N_1} \cup \left(\bigcup_{r \in \mathcal{R}_\ell} \mathcal{E}_{N_2}^{(r)}\right)$$

First, an upper bound of the probability of the event $\mathcal{E}_{N_1}$ is

$$\mathbb{P}(\mathcal{E}_{N_1}) \leq \mathbb{P}\left(\bigcup_{k=1}^{K}\{n_{ku_t}x_{k\ell}^2 > 2\log n_{u_t}\}\right)$$

$$\leq \sum_{k=1}^{K}\mathbb{P}\left(n_{ku_t}x_{k\ell}^2 > 2\log n_{u_t}\right)$$

$$\leq \sum_{k=1}^{K}\mathbb{P}\left(|\hat{p}_{k\ell} - \hat{\rho}_{ku_t}| - \epsilon > \sqrt{\frac{2\log n_{u_t}}{n_{ku_t}}}\right)$$

$$\leq \sum_{k=1}^{K}\mathbb{P}\left(|\hat{p}_{k\ell} - \hat{\rho}_{ku_t}| - |p_{k\ell} - \hat{p}_{k\ell}| > \sqrt{\frac{2\log n_{u_t}}{n_{ku_t}}}\right)$$

$$\leq \sum_{k=1}^{K}\mathbb{P}\left(|p_{k\ell} - \hat{\rho}_{ku_t}| > \sqrt{\frac{2\log n_{u_t}}{n_{u_t}}}\right)$$

By Lemma 1, we know that $u_t$ arrives at most $\overline{N}$ times in expectation. Therefore, the regret induced by the event $\mathcal{E}_{N_1}$ is

$$R_{\mathcal{E}_{N_1}}(\overline{N}) \leq \sum_{k=1}^{K}\sum_{s=1}^{\overline{N}}\mathbb{P}\left(|p_{k\ell} - \hat{\rho}_{ku_t}| > \sqrt{\frac{2\log s}{s}}\ \middle|\ n_{u_t} = s\right)$$

$$\leq \sum_{k=1}^{K}\sum_{s=1}^{\overline{N}}2\exp\left(-4\log s\right)$$

$$\leq 2K\left(1 + \int_{1}^{\overline{N}}\frac{1}{x^4}dx\right)$$

$$\leq 2K\left(1 + \left[-\frac{1}{3x^3}\right]_{1}^{\overline{N}}\right) = \frac{2K}{3}\left(4 - \frac{1}{\overline{N}^3}\right) \tag{30}$$

Next, we evaluate the probability of the event $\mathcal{E}_{N_2}^{(r)}$. First, we assume that $r \notin \mathcal{L}^{\perp}(\ell)$. Then,

$$\mathbb{P}\left(\max_{\{1 \leq n_{ku_t} \leq n_{u_t}\}} n_{ku_t}x_{k\ell}^2 > 2\log\left(\frac{n_{u_t}}{2}\right)\right) \leq \sum_{s=1}^{n_{u_t}}\mathbb{P}\left(|\hat{p}_{k\ell} - \hat{\rho}_{ku_t}| - \epsilon > \sqrt{\frac{2\log(\frac{n_{u_t}}{2})}{n_{ku_t}}}\ \middle|\ n_{ku_t} = s\right)$$

$$\leq \sum_{s=1}^{n_{u_t}}\mathbb{P}\left(|p_{k\ell} - \hat{\rho}_{ku_t}| > \sqrt{\frac{2\log(\frac{n_{u_t}}{2})}{n_{ku_t}}}\ \middle|\ n_{ku_t} = s\right)$$

$$\leq \sum_{s=1}^{n_{u_t}}2\exp\left(-4\log\left(\frac{n_{u_t}}{2}\right)\right) = \frac{32}{n_{u_t}^3}$$

$$\tag{31}$$

If the event $\{\forall k, \max_{\{1 \leq n_{ku_t} \leq n_{u_t}\}} n_{ku_t}x_{k\ell}^2 < 2\log\left(\frac{n_{u_t}}{2}\right)\}$ occurs, then the event $\{n_{k_\ell^* u_t} \geq \frac{n_{u_t}}{2K}\}$ occurs as well. Hence, we can deduce that $\mathbb{P}\left(n_{k_\ell^* u_t} < \frac{n_{u_t}}{2K}\right) \leq \frac{32K}{n_{u_t}^3}$ by (31). The probability of the event $\mathcal{E}_{N_2}^{(r)}$ satisfies:

$$\mathbb{P}(\mathcal{E}_{N_2}^{(r)}) \leq \mathbb{P}\left(n_{k_\ell^* u_t} x_{k_\ell^* r}^2 < 2K \log n_{u_t}\right)$$

$$\leq \mathbb{P}\left(x_{k_\ell^* r} < \sqrt{\frac{2K \log n_{u_t}}{n_{k_\ell^* u_t}}}\right)$$

$$\leq \mathbb{P}\left(|\hat{p}_{k_\ell^* r} - \hat{\rho}_{k_\ell^* u_t}| < \sqrt{\frac{2K \log n_{u_t}}{n_{k_\ell^* u_t}}} + \epsilon\right)$$

$$\leq \mathbb{P}\left(|p_{k_\ell^* r} - p_{k_\ell^* \ell}| - |p_{k_\ell^* \ell} - \hat{\rho}_{k_\ell^* u_t}| - |p_{k_\ell^* r} - \hat{p}_{k_\ell^* r}| < \sqrt{\frac{2K \log n_{u_t}}{n_{k_\ell^* u_t}}} + \epsilon\right)$$

$$\leq \mathbb{P}\left(|p_{k_\ell^* \ell} - \hat{\rho}_{k_\ell^* u_t}| > |p_{k_\ell^* r} - p_{k_\ell^* \ell}| - \sqrt{\frac{2K \log n_{u_t}}{n_{k_\ell^* u_t}}} - 2\epsilon\right)$$

$$\leq \mathbb{P}\left(|p_{k_\ell^* \ell} - \hat{\rho}_{k_\ell^* u_t}| > \frac{1}{2}|p_{k_\ell^* r} - p_{k_\ell^* \ell}| - \sqrt{\frac{2K \log n_{u_t}}{n_{k_\ell^* u_t}}}\right)$$

$$\leq \mathbb{P}\left(|p_{k_\ell^* \ell} - \hat{\rho}_{k_\ell^* u_t}| > \frac{1}{2}|p_{k_\ell^* r} - p_{k_\ell^* \ell}| - \sqrt{\frac{4K^2 \log n_{u_t}}{n_{u_t}}}\right) + \frac{32K}{n_{u_t}^3}.$$

Therefore, the regret induced by event $\mathcal{E}_{N_2}^{(r)}$ is

$$R_{\{\mathcal{E}_{N_2}^{(r)} \backslash \mathcal{E}_{N_1}\}}(\overline{N})$$

$$\leq \sum_{s=1}^{\overline{N}} \mathbb{P}\left(|p_{k_\ell^* \ell} - \hat{\rho}_{k_\ell^* u_t}| > \frac{1}{2}|p_{k_\ell^* r} - p_{k_\ell^* \ell}| - \sqrt{\frac{4K^2 \log s}{s}}\right) + \frac{32K}{s^3}$$

$$\leq \frac{64K^2}{|p_{k_\ell^* r} - p_{k_\ell^* \ell}|^2} \log\left(\frac{6K}{|p_{k_\ell^* r} - p_{k_\ell^* \ell}|}\right)$$

$$+ \sum_{s=\lceil \frac{64K^2}{|p_{k_\ell^* r} - p_{k_\ell^* \ell}|^2} \log\left(\frac{6K}{|p_{k_\ell^* r} - p_{k_\ell^* \ell}|}\right)\rceil}^{\overline{N}} 2\exp\left(-\frac{s|p_{k_\ell^* \ell} - p_{k_\ell^* r}|^2}{8}\right) + 48K$$

$$\leq \frac{64K^2}{|p_{k_\ell^* r} - p_{k_\ell^* \ell}|^2} \log\left(\frac{6K}{|p_{k_\ell^* r} - p_{k_\ell^* \ell}|}\right) + \frac{16}{|p_{k_\ell^* \ell} - p_{k_\ell^* r}|^2} + 48K$$

$$\leq \frac{128K^2}{|p_{k_\ell^* r} - p_{k_\ell^* \ell}|^2} \log\left(\frac{6K}{|p_{k_\ell^* r} - p_{k_\ell^* \ell}|}\right). \tag{32}$$

Next assume that $r \in \mathcal{L}^{\perp}(\ell)$. Then, $k_{\ell}^* \neq k_r^*$ and $p_{k_{\ell}^* \ell} = p_{k_{\ell}^* r}$, and

$$
\begin{aligned}
\mathbb{P}(\mathcal{E}_{N_2}^{(r)}) &\leq \mathbb{P}\left( \sum_{k \in S_{\ell r}} n_{k u_t} x_{kr}^2 < 2K \log n_{u_t} \right) \\
&\leq \mathbb{P}\left( \bigcup_{k \in S_{\ell r}} \{ n_{k u_t} x_{kr}^2 < \frac{2K \log n_{u_t}}{|S_{\ell r}|} \} \right) \\
&\leq \sum_{k \in S_{\ell r}} \mathbb{P}\left( x_{kr} < \sqrt{\frac{2K \log n_{u_t}}{|S_{\ell r}| n_{k u_t}}} \right) \\
&\leq \sum_{k \in S_{\ell r}} \mathbb{P}\left( |p_{k\ell} - \hat{\rho}_{k u_t}| > \frac{1}{2}|p_{kr} - p_{k\ell}| - \sqrt{\frac{2K \log n_{u_t}}{|S_{\ell r}| n_{k u_t}}} \right) \\
&\leq \sum_{k \in S_{\ell r}} \mathbb{P}\left( |p_{k\ell} - \hat{\rho}_{k u_t}| > \frac{1}{2}|p_{kr} - p_{k\ell}| - \sqrt{\frac{2K \log \overline{N}}{|S_{\ell r}| n_{k u_t}}} \right).
\end{aligned}
$$

$$
\begin{aligned}
R_{\mathcal{E}_{N_2}^{(r)}}(\overline{N}) &\leq \sum_{k \in S_{\ell r}} \sum_{s=1}^{\overline{N}} \mathbb{P}\left( |p_{k\ell} - \hat{\rho}_{k u_t}| > \frac{1}{2}|p_{kr} - p_{k\ell}| - \sqrt{\frac{2K \log \overline{N}}{|S_{\ell r}| s}} \Big| n_{k u_t} = s \right) \\
&\overset{(a)}{\leq} \sum_{k \in S_{\ell r}} \left( \frac{32K \log \overline{N}}{|S_{\ell r}||p_{k\ell} - p_{kr}|^2} + \sum_{s=\lceil \frac{32K \log \overline{N}}{|S_{\ell r}||p_{k\ell} - p_{kr}|^2} \rceil}^{\overline{N}} 2\exp\left( -\frac{s|p_{k\ell} - p_{kr}|^2}{8} \right) \right) \\
&\leq \sum_{k \in S_{\ell r}} \frac{32}{|p_{k\ell} - p_{kr}|^2} \left( \frac{K \log \overline{N}}{|S_{\ell r}|} + 1 \right).
\end{aligned} \tag{33}
$$

Combining (30), (32) and (33), the expected regret due to recommendations made by referring to the nearest neighbors in $\mathcal{U}^*$ is,

$$
\begin{aligned}
R_N(T) &\leq m \sum_{\ell} \beta_{\ell}(p_{\sigma_{\ell}(1)\ell} - p_{\sigma_{\ell}(K)\ell}) \left( \frac{2K}{3}\left( 4 - \frac{1}{\overline{N}^3} \right) \right) \\
&+ \sum_{r \in \mathcal{R}_{\ell} \setminus \mathcal{L}^{\perp}(\ell)} \frac{128K^2}{|p_{k_{\ell}^* r} - p_{k_{\ell}^* \ell}|^2} \log\left( \frac{6K}{|p_{k_{\ell}^* r} - p_{k_{\ell}^* \ell}|} \right) + \sum_{r \in \mathcal{L}^{\perp}(\ell)} \sum_{k \in S_{\ell r}} \frac{32}{|p_{k\ell} - p_{kr}|^2}\left( \frac{K \log \overline{N}}{|S_{\ell r}|} + 1 \right) \Bigg) \\
&= \mathcal{O}\left( m \sum_{\ell} \beta_{\ell}(p_{\sigma_{\ell}(1)\ell} - p_{\sigma_{\ell}(K)\ell}) \left( \sum_{r \in \mathcal{R}_{\ell} \setminus \mathcal{L}^{\perp}(\ell)} \frac{K^2 \log K}{\phi(|p_{k_{\ell}^* r} - p_{k_{\ell}^* \ell}|^2)} + \sum_{k \in S_{\ell r}} \sum_{r \in \mathcal{L}^{\perp}(\ell)} \frac{K \log \overline{N}}{|S_{\ell r}||p_{k\ell} - p_{kr}|^2} \right) \right).
\end{aligned}
$$

$\square$

# J Performance guarantees of ECB and Item Clustering: Proof of Theorems 7 and 9

## J.1 Performance guarantees of ECB: Proof of Theorem 7

The proof is straightforward from the results of Theorem 9. Indeed, the latter implies that we can assume that the item clusters estimated from the first phase of the algorithm are exact (the complement of this event happens with probability $1/T$, and hence generates an expected regret $\mathcal{O}(1)$).

Hence we can assume that we know the exact clusters of items in $\mathcal{S}$. Now observe that for each cluster $V_k$ (a subset of $\mathcal{S}$), we have $|V_k| \geq \overline{N}$ (refer to Appendix I for a precise statement). As a consequence, all users can be served using items from $V_1, \ldots, V_K$, except for a few users arriving more than $\overline{N}$. Actually, from Lemma 1, these exceptional arrivals induce an average regret $\mathcal{O}(1)$. ECB applies, for each user, a UCB1 algorithm [1] to select the cluster from which an item is recommended. For this exploitation period, each user in $\mathcal{U}_\ell$ will then induce a regret $\mathcal{O}(\sum_{k \neq k_\ell^*} \frac{\log(\overline{N})}{p_{k_\ell^* \ell} - p_{k\ell}})$. This completes the proof. $\qquad\square$

## J.2 Item Clustering Phase: Proof of Theorem 9

We let $e(v, S) := \sum_{x \in S} A_{vx}$ and $e(A, B) := \sum_{v \in A} \sum_{w \in B} A_{vw}$.

Let

$$p(a,b) := (|V_b| - \mathbb{1}\{a = b\}) \frac{2}{n_0(n_0 - 1)} \sum_{\ell=1}^{L} \beta_\ell p_{a\ell} p_{b\ell}$$

and $p(a, 0) := 1 - \sum_{k=1}^{K} p(a, k)$.

We also define $e(v, V_0)$ and $e(v, S_0^{(t)})$ as follows:

$$e(v, V_0) := s - \sum_{k=1}^{K} e(v, V_k) \quad \text{and}$$

$$e(v, S_0^{(t)}) := s - \sum_{k=1}^{K} e(v, S_k^{(t)}),$$

where $s$ is the number of users who have received recommendations at least twice until $t = 10m$.

**Proof of the theorem.** The proof of Theorem 9 relies on the following random matrix concentration inequality. Specifically, from the matrix Bernstein inequality, we can bound $\|A - \mathbb{E}[A]\|$ as follows.

**Lemma 18.** *Assume $T > 10m$. Let $A$ be the adjacency matrix obtained in Algorithm 4. Let $\|\cdot\|$ denote the spectral norm. Then,*

$$\mathbb{P}\left(\|A - \mathbb{E}[A]\| > 5\sqrt{\frac{m \log m}{n_0}}\right) \leq \frac{1}{m^2}.$$

From Lemma 18, we deduce that $\hat{A}$ (the rank-$K$ approximation of $A$) is approximately the same as $\mathbb{E}[A]$. Indeed, since both $\hat{A}$ and $\mathbb{E}[A]$ are of rank $K$,

$$\begin{aligned}
\|\hat{A} - \mathbb{E}[A]\|_F^2 &\leq 2K\|\hat{A} - \mathbb{E}[A]\|^2 \\
&\leq 4K(\|\hat{A} - A\|^2 + \|A - \mathbb{E}[A]\|^2) \\
&\leq 8K\|A - \mathbb{E}[A]\|^2,
\end{aligned}$$

where $\|A - \mathbb{E}[A]\|$ is negligible compared to $\|\mathbb{E}[A]\| = \Omega(\frac{m}{n_0})$.

From the columns of $\hat{A}$, we can classify items. Here, $\|\mathbb{E}[A]_v - \mathbb{E}[A]_w\| = \Omega(\frac{m}{n_0^{3/2}})$ when $v$ and $w$ belong to different clusters and $\mathbb{E}[A]_v = \mathbb{E}[A]_w$ when $v$ and $w$ belong to the same cluster. Therefore,

the columns of $\hat{A}$ are concentrated around the correct cluster column unless $\|\hat{A}_v - \mathbb{E}[A]_v\| = \Omega(\frac{m}{n_0^{3/2}})$.
From this argument, the spectral decomposition used in the algorithms satisfies

$$\left| \cup_{k=1}^K \left( S_k^{(0)} \setminus V_k \right) \right| = O\left( \frac{n_0^2 \log m}{m} \right), \tag{34}$$

since $\sum_{v \in \mathcal{S}} \|\hat{A}_v - \mathbb{E}[A]_v\|^2 = \|A - \mathbb{E}[A]\|_F^2 = O(\frac{m \log m}{n_0})$ from Lemma 18 (cf. [26]).

In the improvement step of Algorithm 5, the algorithm refines the result of Spectral Decomposition iteratively. We denote the set of misclassified items after $t$-th iteration by $\mathcal{E}^{(t)}$. We also introduce $\mathcal{E}_{k\ell}^{(t)} = S_k^{(t)} \cap V_\ell$ so that $\mathcal{E}^{(t)} = \bigcup_{k=1}^K \bigcup_{\ell:\ell \neq k} \mathcal{E}_{k\ell}^{(t)}$.

Since the items move to more likely cluster with respect to $\hat{p}(i,j)$ at each step,

$$0 \leq \sum_{k,\ell:k\neq\ell} \sum_{v\in\mathcal{E}_{k\ell}^{(t+1)}} \sum_{j=0}^K e\left(v, S_j^{(t)}\right) \log \frac{\hat{p}(k,j)}{\hat{p}(\ell,j)}$$

$$\overset{(a)}{\leq} \sum_{k,\ell:k\neq\ell} \sum_{v\in\mathcal{E}_{k\ell}^{(t+1)}} \sum_{j=0}^K e\left(v, S_j^{(t)}\right) \log \frac{p(k,j)}{p(\ell,j)}$$

$$+ C_1 |\mathcal{E}^{(t+1)}| \sqrt{\frac{m \log m}{n_0}}$$

$$\overset{(b)}{\leq} \sum_{k,\ell:k\neq\ell} \sum_{v\in\mathcal{E}_{k\ell}^{(t+1)}} \sum_{j=0}^K e\left(v, V_j\right) \log \frac{p(k,j)}{p(\ell,j)}$$

$$+ C_1 |\mathcal{E}^{(t+1)}| \sqrt{\frac{m \log m}{n_0}} + C_2 e(\mathcal{E}^{(t)}, \mathcal{E}^{(t+1)})$$

$$\overset{(c)}{\leq} \sum_{k,\ell:k\neq\ell} \sum_{v\in\mathcal{E}_{k\ell}^{(t+1)}} \sum_{j=0}^K e\left(v, V_j\right) \log \frac{p(k,j)}{p(\ell,j)}$$

$$+ C_1 |\mathcal{E}^{(t+1)}| \sqrt{\frac{m \log m}{n_0}} + C_3 \sqrt{|\mathcal{E}^{(t)}||\mathcal{E}^{(t+1)}| \frac{m \log m}{n_0}}$$

$$\overset{(d)}{\leq} -C_4 \frac{m}{n_0} |\mathcal{E}^{(t+1)}| + C_3 \sqrt{|\mathcal{E}^{(t)}||\mathcal{E}^{(t+1)}| \frac{m \log m}{n_0}} \tag{35}$$

where (a) is obtained from Lemma 19; (b) stems from the fact that $\frac{p(k,j)}{p(\ell,j)}$ is a positive constant for all $1 \leq j \leq K$; (c) follows from Lemma 20; and (d) is obtained from Lemma 21.

From (35), we can conclude that:

$$\frac{|\mathcal{E}^{(t+1)}|}{|\mathcal{E}^{(t)}|} \leq \frac{C_3^2}{C_4^2} \frac{n_0 \log m}{m}$$

$$\overset{(a)}{\leq} C_5 \frac{1}{\log T},$$

where $(a)$ is from $n_0 \leq m/(\log T)^2$ and $10m < T$.

Therefore, after $\log(n_0)$ iterations, we have recovered the perfect clusters. $\qquad\square$

Nest we state the lemmas used in the proof above.

**Lemma 19.** *When* $|\mathcal{E}^{(0)}| = O\left( \frac{n_0^2 \log m}{m} \right)$*, with probability* $1 - \frac{2}{m^2}$,

$$\sum_{i=0}^K e\left(v, S_i^{(t)}\right) \left| \log \frac{p(k,i)}{\hat{p}(k,i)} \right| = O\left( \sqrt{\frac{m \log m}{n_0}} \right).$$

**Lemma 20.** *When* $\|A - \mathbb{E}[A]\| = O\left(\sqrt{\frac{m \log m}{n_0}}\right)$, $|\mathcal{E}^{(t)}| = O\left(\frac{n_0^2 \log m}{m}\right)$, *and* $|\mathcal{E}^{(t+1)}| = O\left(\frac{n_0^2 \log m}{m}\right)$,

$$\sum_{v \in \mathcal{E}^{(t+1)}} e(v, \mathcal{E}^{(t)}) = O\left(\sqrt{|\mathcal{E}^{(t)}||\mathcal{E}^{(t+1)}|\frac{m \log m}{n_0}}\right).$$

**Lemma 21.** *With probability at least* $1 - \frac{1}{m^2}$, *for all k, for all* $v \in V_k$,

$$\sum_{a=0}^{K} e(v, V_a) \log\left(\frac{p(k, a)}{p(k', a)}\right) = \Omega\left(\frac{m}{n_0}\right) \quad \text{for all} \quad k' \neq k.$$

### J.3 Proof of the lemmas

**Proof of Lemma 18.**

The adjacency matrix $A$ can be considered as the sum of $s$ samples of connected pairs for some $s \geq \frac{m}{2}$. We denote such samples as $X_\ell$ for $1 \leq \ell \leq s$. Then, $A = \sum_{\ell=1}^{s} X_\ell$.

By the matrix Bernstein inequality (cf. [30]), we have:

$$\mathbb{P}\left(\|A - \mathbb{E}[A]\| > t\right) = \mathbb{P}\left(\|\sum_{l=1}^{s}(X_\ell - \mathbb{E}[X_\ell])\| > t\right)$$

$$\leq n_0 \exp\left(-\frac{t^2}{2\left(\sigma^2(A) + Dt/3\right)}\right), \tag{36}$$

where $\sigma^2(A) = \|\mathbb{E}[(A - \mathbb{E}[A])^2]\|$ and $D$ is an upper bound of $\|X_\ell - \mathbb{E}[X_\ell]\|$ for all $\ell$.

The expectation of the elements $x_{ij}$ of matrix $X_\ell$ is

$$\mathbb{E}\left[x_{ij} | i \in V_k, j \in V_{k'}\right] = \frac{2}{n_0(n_0 - 1)} \sum_{\ell=1}^{L} \beta_\ell p_{k\ell} p_{k'\ell}.$$

Since every elements in $\mathbb{E}[X_\ell]$ is less than $\frac{2}{n_0(n_0-1)}$, we have:

$$\|X_\ell - \mathbb{E}[X_\ell]\| \overset{(a)}{\leq} \sqrt{\frac{4n_0^2}{n_0^2(n_0 - 1)^2} + 2}$$

$$\leq \frac{2}{n_0 - 1} + \sqrt{2}, \tag{37}$$

where (a) is from the fact that Frobenius norm of the matrix is greater than its spectral norm.

From (37), we deduce that we can choose $D = \frac{3}{2}$.
Moreover, the variance of the matrix A is

$$\sigma^2(A) = \|\mathbb{E}[(A - \mathbb{E}[A])^2]\|$$

$$\overset{(a)}{=} \|\sum_{\ell=1}^{s} \mathbb{E}[(X_\ell - \mathbb{E}[X_\ell])^2]\|$$

$$\leq \sum_{\ell=1}^{s} \|\mathbb{E}[(X_\ell - \mathbb{E}[X_\ell])^2]\|$$

$$\leq \sum_{\ell=1}^{s} \left(\|\mathbb{E}[X_\ell^2]\| + \|\mathbb{E}[X_\ell]^2\|\right), \tag{38}$$

where (a) is obtained from the independence of the $X_\ell$'s.

To get an upper bound of (38), observe that the expectation of the $(i, j)$-th element of matrix $X_\ell^2$ is

$$\mathbb{E}[(X_\ell^2)_{ij}] = \mathbb{E}\left[\sum_{k=1}^{n_0} x_{ik}x_{kj}\right] = \sum_{k=1}^{n_0} \mathbb{E}[x_{ik}x_{kj}] = 0.$$

In addition, the expectation of the $(i, i)$-th elements of matrix $X_\ell^2$ is

$$\mathbb{E}[(X_\ell^2)_{ii}] = \sum_{k=1}^{n_0} \mathbb{E}[x_{ik}^2] = \sum_{k=1}^{n_0} \mathbb{E}[x_{ik}] \le \frac{2}{n_0}.$$

On the other hand, the elements of $\mathbb{E}[X_\ell]^2$ are $\mathcal{O}\left(\frac{1}{n_0^3}\right)$, which implies $\|\mathbb{E}[X_\ell]^2\| = \mathcal{O}\left(\frac{1}{n_0^2}\right)$. Hence, using (38), we deduce that $\sigma^2(A) \le \frac{2s}{n_0} + \mathcal{O}\left(\frac{s}{n_0^2}\right) \le \frac{2m}{n_0}$.

Now, an upper bound of (36) is

$$n_0 \exp\left(-\frac{t^2}{\frac{4m}{n_0} + t}\right). \tag{39}$$

To conclude the proof of this lemma, we need to consider two cases: $n_0 = \frac{m}{(\log T)^2}$ and $n_0 = n$.

(i) When $n_0 = \frac{m}{(\log T)^2}$, $t = 5\sqrt{\log m} \log T$. So, (39) becomes:

$$\frac{m}{(\log T)^2} \exp\left(-\frac{25(\log T)^2 \log m}{4(\log T)^2 + 5\sqrt{\log m} \log T}\right) \overset{(a)}{\le} \exp\left(-3\log m + \log\left(\frac{m}{(\log T)^2}\right)\right)$$

$$\le \frac{1}{m^2}$$

where (a) is obtained from the assumption $T > 10m$.

(ii) If $n_0 = n$, $t = 5\sqrt{\frac{m \log m}{n}}$. Then, (39) becomes:

$$n \exp\left(-\frac{\frac{25m \log m}{n}}{\frac{4m}{n} + 5\sqrt{\frac{m \log m}{n}}}\right) \le \exp\left(-3\log m + \log n\right)$$

$$\overset{(a)}{\le} \frac{1}{m^2}$$

where (a) holds since $n \le \frac{m}{(\log T)^2}$. $\qquad\square$

**Proof of Lemma 19.**

Recall that

$$p(a, b) = (|V_b| - \mathbb{1}\{a = b\})\frac{2}{n_0(n_0 - 1)}\sum_{\ell=1}^{L} \beta_\ell p_{a\ell} p_{b\ell}$$

and $p(a, 0) = 1 - \sum_{k=1}^{K} p(a, k)$. The estimations are $\hat{p}(i, j) = \frac{\sum_{v \in \mathcal{S}_i} \sum_{v' \in \mathcal{S}_j} A_{v,v'}}{s|S_i^{(0)}|}$ for all $1 \le i, j \le K$ and $\hat{p}(i, 0) = 1 - \sum_{k=1}^{K} \hat{p}(i, k)$.

An upper bound of $|p(i, j) - \hat{p}(i, j)|$ for $1 \le i, j \le K$ is

$$|\hat{p}(i, j) - p(i, j)| \le \frac{1}{s|S_i^{(0)}|}\left|e(S_i^{(0)}, S_j^{(0)}) - \mathbb{E}[e(S_i^{(0)}, S_j^{(0)})]\right|$$

$$+ \frac{1}{s|S_i^{(0)}|}\left|\mathbb{E}[e(S_i^{(0)}, S_j^{(0)})] - s|S_i^{(0)}|p(i, j)\right|. \tag{40}$$

Let $\mathcal{A}$ be the set of partitions $\{S_k\}_{1 \le k \le K}$ of the set $\mathcal{S}$. Recall that $\mathcal{S}$ is of cardinality $n_0$. Then,

$$|\mathcal{A}| \le K^{n_0}. \tag{41}$$

Now, we fix one partition $\{S_k\} \in \mathcal{A}$. Then, by Chernoff-Hoeffding bound,

$$\mathbb{P}\left(\left|e(S_i, S_j) - \mathbb{E}[e(S_i, S_j)]\right| < \sqrt{mn_0 \log m} \text{ for all } i, j\right) \ge 1 - \exp\left(-\Theta\left(n_0 \log m\right)\right). \tag{42}$$

Combining (41) and (42), we deduce that the following event holds:

$$\left|e(S_i, S_j) - \mathbb{E}[e(S_i, S_j)]\right| < \sqrt{mn_0 \log m}$$

for all $i, j$ and $\{S_k\} \in \mathcal{A}$ with probability $1 - \exp\left(-\Theta\left(n_0 \log m\right)\right)$ (just applying a union bound).

Since $\{S_k^{(0)}\} \in \mathcal{A}$, with probability $1 - \exp\left(-\Theta\left(n_0 \log m\right)\right)$,

$$\left|e(S_i^{(0)}, S_j^{(0)}) - \mathbb{E}[e(S_i^{(0)}, S_j^{(0)})]\right| < \sqrt{mn_0 \log m} \tag{43}$$

for all $i, j$.

On the other hand, since $|\mathcal{E}^{(0)}| = \mathcal{O}\left(\frac{n_0^2 \log m}{m}\right)$ from the assumption, with probability $1 - \exp\left(-\Theta\left(n_0 \log m\right)\right)$,

$$\frac{1}{s|S_i^{(0)}|}\left|\mathbb{E}[e(S_i^{(0)}, S_j^{(0)})] - s|S_i^{(0)}|p(i,j)\right| = \mathcal{O}\left(\frac{n_0 \log m}{m}p(i,j)\right), \tag{44}$$

for all $i, j$.

Then, conditioned on (44) for all $1 \le i, j \le K$, we can derive an upper bound of $|p(i,j) - \hat{p}(i,j)|$ for $1 \le i, j \le K$, using (40), (43) and (44):

$$|p(i,j) - \hat{p}(i,j)| = \mathcal{O}\left(\sqrt{\frac{n_0 \log m}{m}}p(i,j)\right),$$

which implies that for all $1 \le i, j \le K$

$$\left|\log\frac{\hat{p}(i,j)}{p(i,j)}\right| \le \frac{|p(i,j) - \hat{p}(i,j)|}{p(i,j)}$$
$$= \mathcal{O}\left(\sqrt{\frac{n_0 \log m}{m}}\right). \tag{45}$$

Furthermore, an upper bound of $\left|\log\frac{\hat{p}(i,0)}{p(i,0)}\right|$ is

$$\left|\log\frac{\hat{p}(i,0)}{p(i,0)}\right| = \mathcal{O}\left(\frac{1}{n_0}\sqrt{\frac{n_0 \log m}{m}}\right), \tag{46}$$

with probability at least $1 - \exp\left(-\Theta\left(n_0 \log m\right)\right)$.

We also have $\mathbb{E}[e(v, \mathcal{S})] \le \frac{m}{n_0}$ and from Chernoff inequality,

$$\mathbb{P}\left(|e(v, \mathcal{S}) - \mathbb{E}[e(v, \mathcal{S})]| > \sqrt{4m \log m}\right) \le \frac{1}{m^2}. \tag{47}$$

Finally, we obtain the following from (45), (46) and (47):

$$\sum_{i=0}^{K} e\left(v, S_i^{(t)}\right)\left|\log\frac{p(k,i)}{\hat{p}(k,i)}\right| = \mathcal{O}\left(\sqrt{\frac{m \log m}{n_0}}\right),$$

with probability at least $1 - \frac{2}{m^2}$. This concludes the proof. $\qquad\square$

**Proof of Lemma 20.** We have:

$$\sum_{v \in \mathcal{E}^{(t+1)}} \left( e(v, \mathcal{E}^{(t)}) - \mathbb{E}[e(v, \mathcal{E}^{(t)})] \right) \le \mathbb{1}_{\mathcal{E}^{(t)}}^T (A - \mathbb{E}[A]) \mathbb{1}_{\mathcal{E}^{(t+1)}},$$

where $\mathbb{1}_S$ is the vector whose $i$-th component is equal to 1 if $i \in S$ and to 0 otherwise. Since $\mathbb{E}[e(v, \mathcal{E}^{(t)})] \le \frac{2m}{n_0^2} |\mathcal{E}^{(t)}| \le 2 \log m$,

$$
\begin{aligned}
\sum_{v \in \mathcal{E}^{(t+1)}} e(v, \mathcal{E}^{(t)}) &\le \sum_{v \in \mathcal{E}^{(t+1)}} \left( e(v, \mathcal{E}^{(t)}) - \mathbb{E}[e(v, \mathcal{E}^{(t)})] \right) + 2|\mathcal{E}^{(t+1)}| \log m \\
&\le \|\mathbb{1}_{\mathcal{E}^{(t)}}^T (A - \mathbb{E}[A]) \mathbb{1}_{\mathcal{E}^{(t+1)}}\| + 2|\mathcal{E}^{(t+1)}| \log m \\
&\le \|\mathbb{1}_{\mathcal{E}^{(t)}}^T\| \|(A - \mathbb{E}[A])\| \|\mathbb{1}_{\mathcal{E}^{(t+1)}}\| + 2|\mathcal{E}^{(t+1)}| \log m \\
&\overset{(a)}{=} \mathcal{O}\left( \sqrt{|\mathcal{E}^{(t)}||\mathcal{E}^{(t+1)}| \frac{m \log m}{n_0}} \right),
\end{aligned}
$$

where for $(a)$, we used the assumption that $\|A - \mathbb{E}[A]\| = \mathcal{O}\left( \sqrt{\frac{m \log m}{n_0}} \right)$ and the definition of $\ell_2$ norm. $\qquad\square$

**Proof of Lemma 21.**

From Chernoff-Hoeffding bound, there exists $C > 0$ such that

$$\mathbb{P}\left( |e(v, V_a) - \mathbb{E}[e(v, V_a)]| > C\sqrt{\frac{m \log m}{n_0}} \right) \le \frac{1}{m^3}. \tag{48}$$

We also have for all $v \in V_k$ and all $k$,

$$\sum_{a=0}^{K} \mathbb{E}[e(v, V_a)] \log\left( \frac{p(k, a)}{p(k', a)} \right) = \Omega\left( \frac{m}{n_0} \right) \quad \text{for all} \quad k' \ne k. \tag{49}$$

Since $\log\left( \frac{p(k,a)}{p(k',a)} \right) = \Theta(1)$ and $\frac{m}{n_0} = \Omega((\log T)^2)$, from (48) and (49), we have Lemma 21. $\qquad\square$