[Reviews · NeurIPS 2020]

Review 1

Summary and Contributions: This paper formulates the recommendation problem as non-repeated item recommendation to users who arrive randomly from a user population. The value of items is learned purely from rewards observed from these users, without any side information being available. The paper presents algorithmic solutions for three different settings with extensive theoretical analysis.

Strengths: - The extensive theoretical analysis of regret in this problem is well presented. While this reviewer did not study the proofs in detail, they appear generally clear and convincing. - (Almost) optimal algorithms for this setting are presented and evaluated.

Weaknesses: - The problem modelled is essentially the cold-start recommendation problem, where items are shown to a new user to learn the user’s preferences. Similarly to this work, each item for which preferences are obtained is only presented once. A key difference is the ‘purity’ of the problem here (no side information) and deep theoretical analysis. It would be helpful for the related work section to make this connection. - The the first two settings studied, recommendation utility does not depend on the user (as all users are statistically indistinguishable). This is a highly simplified setting. While the simplification still requires complex analysis, it does not represent a classic recommendation application (where personalization plays a key role). - The authors argue that the techniques developed to allow the analysis are a key contribution. However, almost all of this is in the supplemental work, rather than in the core of the paper. Similarly, the authors present all “the pseudo-codes of our algorithms, all proofs, numerical experiments, as well as some insightful discussions” in the appendix, suggesting there is just too much attempted content here: The paper cannot really be appreciated without the appendix, so the appendix is not really “supplemental”.

Correctness: This reviewer did not check proofs for correctness, especially as the bulk of the results are in the appendix.

Clarity: The writing of the main paper is clear and easy to follow. That's great to see for a largely theoretical paper!

Relation to Prior Work: Prior work is discussed, although connected to cold start recommendation (especially where bandit algorithms are used) is not made.

Reproducibility: Yes

Additional Feedback: In terms of reproducibility, the experimental results (presented in the appendix) use simulated datasets. It is a little unclear how to reproduce such an experiment. Perhaps code for this simulation could be shared? Overall, there is clearly a huge amount of analysis in here, and the techniques may well be useful. I would encourage the authors to make these insights clearer to the reader in the main body of the submission.


Review 2

Summary and Contributions: Added after rebuttal ------------------------ I would actually like to raise my score to 6 and I am actually inclined to accept this paper now. 1. Question about lower bound: I think the authors have convinced me that R_{ic}(T) term in the lower bound is actually non-trivial to arrive at and the reduction is actually interesting. 2. 1/ \Delta dependence: The Delta term does appear in the lower bound. I am still confused as to how an explore and then commit style of algorithm is getting the optimal 1/\Delta dependence in the regret upper bound. Authors have provided some discussion on the relation with related work as well, which is satisfactory and I do not think it is fair to push them to compare with Bressler et al as the settings are different and I am not sure what hypothesis would we be testing. Overall I now think that the paper is interesting theoretically. ---------------------------------------------- The paper studies three problem settings for online recommendation of items to users under the lens of regret in a bandit setting. The common condition in all three settings is that one item cannot be recommended to a user more than once. The first setting is when users are identical (statistically) and items are clustered. The second settings considers identical users and all items have a different prob. of success. The probabilities themselves come from an initial distribution. The third setting is when both users and items are clustered. The paper provides regret lowers bound for these settings and and also algorithms that achieve regret close to those lower bounds.

Strengths: 1. The criterion of not recommending the same item twice to a user is an important one. It is rarely considered in the bandit literature except. I only knew of one other work (https://arxiv.org/pdf/1411.6591.pdf) that considers this criterion. 2. The paper presents the regret lower bounds and upper bounds in an interpretable way. The regret components are separated into regret due to the recommending new items constraint, and the regret due to the learning task, in all three settings.

Weaknesses: My main observation is that the paper does not clearly compares the regret bounds it obtains with existing literature. I find the presentation of the regret bounds to be fairly non-standard and hard to interpret. These are some of my concerns. 1. Can the authors explain better the technical ideas in the analysis of Theorem 1 and why is such an analysis novel? It seems to me that R_sp(T) is just a standard K-armed bandit lower bound which can be applied here by the reduction to the case where the cluster identity of each item is known, but {p_1, ..., p_K} needs to be learned. On the other hand, R_{ic} just seems to be something coming from running out of items to recommend from the top cluster and a bound on the size of such a cluster because of the sampling from {\alpha}'s initially. In my opinion this is a result of the regret definition being defined in such a way that the oracle algorithm can always recommend items from the top cluster. This makes the problem less interesting. You will always incur linear regret when T > n * size of the top cluster. The regret analysis will be more nuanced if the same restrictions are imposed on the oracle. Also, when the regret is discussed in the introduction, K which is an important parameter is not mentioned. 2. Similarly, in Theorem 2, the problem seems almost identical to the case of satisficing bandits (https://arxiv.org/pdf/1512.07638.pdf). You just need to account for the case when you run out of items in the set with mean >= mu_{1 - \epsilon}. I am not sure if this is an interesting setting to study, when compared to to already studied satisficing bandit problem. 3. All the algorithms studied are explore then commit kind of algorithms. These algorithms will always have 1/ \Delta^2 kind of dependence (in front of log T terms), where \Delta is the gap between the best and the second best items. I do need see this dependence in Theorem 4 for instance. Is this hidden in the Log ^3 T term? 4. There is no discussion or comparison to https://arxiv.org/pdf/1411.6591.pdf in terms of regret guarantees wrt to Model C, which is in a similar setting. An empirical comparison would also be good to see, as model C seems to be the only practically relevant recommendation setting. The paper misses several references for low-ranked bandit models and comparisons (Empirical) with them such as these papers: 1. http://proceedings.mlr.press/v54/katariya17a/katariya17a.pdf 2. http://papers.nips.cc/paper/5985-efficient-thompson-sampling-for-online-matrix-factorization-recommendation.pdf 3. http://proceedings.mlr.press/v54/sen17a.html

Correctness: I checked the analysis of theorem 1 and theorem 2 and partially theorem 4. They seems to be correct. The results are not presented in standard way that makes comparisons harder. For instance, check my previous comment on the gap dependence of the bounds.

Clarity: The paper is written moderately well. The results are presented in non-standard ways, though.

Relation to Prior Work: I find this lacking in the paper overall. See my earlier comments.

Reproducibility: Yes

Additional Feedback:


Review 3

Summary and Contributions: This paper studies 3 models in the online recommendation setting, stressing the constraint that no items can be recommended twice to a user. They design ETC-type algorithms and provide both upper and lower regret bounds for each of the three models.

Strengths: They meticulously study the effect of the no-repetition constraint on the regret. The results are complete, including both upper and lower bounds.

Weaknesses: The main weakness in my opinion is the algorithms are based on the ETC idea, which means the algorithm can not adaptively update when the environment changes. The current algorithms run with a fixed (T,m,n) and fails to deal with the case that when there are a little more rounds and several new users/items are added. If the algorithms can't solve this case, I don't see the motivation to assume the asymptotical property among T,m,n. Also if m changes with T, the assumption that all users come in a uniform way should change. Model B seems a little far away from A&C, also from the regret definition and results. Perhaps the authors could just present A&C with more adaptive algorithms. Where is the dependence on n for the regret upper and lower bounds? Since there are gaps between upper and lower bounds, I don't get 'minimal regret' in the title. Also I would suggest the authors change the title to a more specific one. --- I have read the rebuttal.

Correctness: Yes.

Clarity: It is good, but it would be much better if the authors can simplify the notations and reorganise materials in a clearer way.

Relation to Prior Work: Yes.

Reproducibility: Yes

Additional Feedback:


Review 4

Summary and Contributions: The paper provides a comprehensive analysis of three setups for recommender systems in an online setting. The paper considers a setup with n items and m items, and at each time step, a user is chosen uniformly at random, and a recommendation has to be provided to that user. An important constraint is that items cannot be recommended twice to a user. Performance is measured in terms of the regret relative to an oracle. The closest related work is [3,4,13] which consider similar setups, but the paper goes significantly beyond the results presented in those papers, which focus on user and item clusters. The paper considers three different structural assumptions: A: Clustered items and statistically identical users. Here it is key to identify the items from the best cluster, and then recommend those to users. B: Unclustered items and statistically identical users. Here the items are not clustered, but have the same reward probability for each user. C: Clustered items and clustered users. Here, both items and users are clustered. This is the most complicated and perhaps most relevant setup studied in the paper. Those structural assumptions dictate the probability of a user item pair returning reward 1 or 0, and the relations of those probabilities relative to other users and items. The setups also, to some extent, dictate natural algorithms for exploiting the structure. The paper states those algorithms in section 5. The paper provides regret lower bounds in Section 4. The results show that the algorithms and analysis for setups A and B are near optimal (up to logarithmic factors, in an interesting regime). The analysis for setup C is more intricate, but the results are again order wise optimal.

Strengths: The paper provides lower and upper bounds for three models of online recommender systems. The results appear correct (albeit I did not go through the proofs in the appendix), and are highly relevant to the NeurIPS community, as recommender systems are a classical learning problem, with relatively few theoretical results. It is interesting how the results reflect particular aspects of the model, such as the constraint that items can only be recommended once.

Weaknesses: The modeling assumptions are relatively far from practice, for example the point in many recommender systems is that users do not behave the same, yet Model A and B assume they are statistically equivalent. Model C accounts for different users. Moreover, the paper considers the regime of log(m) = o(n), so the paper considers a regime where the number of users is enormous relative to the users. The assumption that the number of users is larger is not restrictive, but the assumption that it is exponentially large is rather restrictive. ---- Post Rebuttal: Thanks for clarifying that m can be polynomial in n.

Correctness: I have no reason to think that the results are incorrect, but I did not go through the proofs in the appendix, to check for correctness.

Clarity: The main body of the paper is well written.

Relation to Prior Work: To the best of my knowlege, related work is adequately cited. Specifically, related works are [3,4,13], and the paper referes to those papers early on.

Reproducibility: Yes

Additional Feedback:

[Author Response · NeurIPS 2020]

We would like to thank the reviewers for carefully reading our paper, and for their insightful and constructive comments.
We will address these comments and update our draft accordingly. Please find our answers below.

**Reviewer 1:**
**1) Comparison to other work and the no-repetition constraint.** The main novelty of our paper lies in the fact that
we consider various settings with the no-repetition constraint and yet we are able to derive tight regret lower bounds
and to devise algorithms achieving these limits.
Note that the two papers, Zhou and Brunskill, IJCAI 2016, and Kawale et al, NeurIPS 2015) you suggested, *do not*
consider the no-repetition constraint. To the best of our knowledge, Bresler et al. paper [3] and [4] are the only papers
who theoretically analyze the no-repetition constraint (but with different models and an analysis that is not as exhaustive
as ours, i.e., providing regret lower bounds and achieving algorithms).
**2) Statistically identical users.** In the first two models, we consider statistically identical users. Indeed this simplifies
the analysis, but the latter is still challenging when accounting for the no-repetition constraint. Statistically identical
users can be a good assumption when the budget $T$ is rather small (if a user is not seen many times within this time
budget). In addition, the first model also serves as a 'warm-up' to understand the last model, where we consider
clustered users. Comparing the results of Models A (identical users) and C (clustered users) is insightful: it can tell us
in which regimes exploiting the user clusters can be beneficial. For example, we remark that for small budgets (e.g.
$T = o(\frac{m}{\Delta})$), we cannot leverage the user clusters to reduce regret.
**3) Appendices.** The proofs and pseudo-codes are in appendix due to space constraints. However, the main ideas of the
algorithms are thoroughly discussed in the paper. Also note that when published on the NeurIPS website, both the main
paper and the supplementary material are available.

**Reviewer 2:** For generic remarks on the novelty of our framework, please refer to answer 1) to Reviewer 1.
**1) Novelty of the analysis.** In Theorem 1, the lower bounds $R_{nr}(T)$ and $R_{sp}(T)$ can be easily derived using existing
techniques in the bandit literature. The lower bound $R_{ic}(T)$, however, is extremely challenging to derive due to the
no-repetition constraint, which makes the problem non-asymptotic by nature (new items must be tested continuously).
This term captures the fact that the decision-maker does not know the item clusters. We are unaware of any paper
deriving regret lower bound due to such a lack of knowledge. Our main proof ingredient is a mapping of this lack of
knowledge to a 2-arm bandit problem (see Lemma 7 Appendix D); and we find this argument elegant. We then use
finite-time regret lower bound for this type of bandits. Importantly, $R_{ic}(T)$ *does not* arise because of our definition of
regret. In Appendix C, we prove that the difference between our definition of regret and the true regret is negligible
compared to $R_{ic}(T)$. Hence this term is not artificial at all.
**2) About Model B.** Our results are not simple extensions of those for the satisficing bandit problem, again due to the
no-repetition constraint. Also, remember that users arrive randomly. Deriving the regret lower bound in Theorem 2 is
again technical.
**3) Dependence in $\frac{1}{\Delta}$.** This dependence is present in our results (see the summary in the introduction). For example, in
Theorem 1 (lower bound), this term is present in $R_{ic}(T)$ through $\phi$ and in $R_{sp}(T)$ explicitly. In Theorem 4, it is present
in the first term of the regret upper bound, because in the sum, the term for $k = 2$ scales as $1/\Delta$. In Theorem 5, the
term $\varepsilon$ corresponds to the gap, and in Theorem 6, various gaps are also present.
**4) Related work.** Thanks for pointing out the two papers on low-rank bandits (again, these papers do not account for
the no-repetition constraint). Regarding Bresler et al. 2014 paper, the setting is a bit different, even from our Model C
(they do not have clustered items). Another important difference is that they consider that users arrive simultaneously,
whereas, in our setting, users arrive sequentially, which is more realistic. Bresler et al. uses a cosine similarity approach
that requires an $\varepsilon$-greedy exploration approach. In turn, this would lead to higher regret. Overall, we found it very
difficult to compare our results to theirs.

**Reviewer 3:**
**1) Changes in environment.** Deriving efficient ETC algorithms was very already challenging. Devising algorithms
that could adapt to non-stationary settings is left for future work.
**2) About Model B.** We wished to include this model to account for the possibility of un-clustered items.
**3) Optimal regrets are independent of $n$.** As part of our results, we figured out that under optimal algorithms,
the regret does not scale in $n$ for most regimes of $(T, n, m)$. Note that in our regret expressions, $\frac{T}{m}(\ll n)$ roughly
corresponds to the number of items that are recommended.
**4) Choice of the title.** The gaps between the upper/lower regret bounds are small. But we can modify the title to
remove 'minimal'.

**Reviewer 4**: For the relevance of our Models A and B, please refer to answer 2) to Reviewer 1.
**Assumption** $\log(m) = o(n)$. Note that $m$ can be polynomial in $n$. This is the same assumption as that made and
justified in Bresler et al., NeurIPS 2014. Refer to the latter paper for an explanation of why the assumption is without
loss of generality.

[Meta-Review · NeurIPS 2020]

The reviewers warmed up to this paper during the discussion. Its scores increased from (5, 5, 6, 7) to (6, 6, 6, 7). Although we agreed that there are shortcomings, there are also new results. So the paper is worth accepting. My additional comments are below: 1) The title is indeed confusing. The authors study a very specific problem and it is unclear how this addresses the bigger picture of regret minimization when no arm can be pulled twice. My opinion is that the learning agent has to learn to sort the arms. If you cannot pull the best arm twice, the second best thing is to pull the best two arms. In general, you want to pull the best K arms in the descending order. 2) Sections 5.1 and 5.2 study the above problem. I find the solutions unconvincing though. For instance, in Theorem 1, the regret seems to be driven by the hardness of sorting the best and second best arms, in term 1 / (p_1 - p_2)^2. What happened to the other arms? This is due to the algorithm design, which only learns to identify the best cluster. 3) The proposed algorithms are impractical. The authors essentially try to redo online low-rank factorization without going into the latent space.